# Recovering true FRET efficiencies from smFRET investigations requires triplet state mitigation

Avik K. Pati [1,4,5], Zeliha Kilic [1,5], Maxwell I. Martin[1,2], Daniel S. Terry[1], Alessandro Borgia [1], Sukanta Bar [1,2], Steffen Jockusch [3], Roman Kiselev [1], Roger B. Altman[1,2] & Scott C. Blanchard [1,2] ✉

Single-molecule fluorescence resonance energy transfer (smFRET) methods employed to quantify time-dependent compositional and conformational changes within biomolecules require elevated illumination intensities to recover robust photon emission streams from individual fluorophores. Here we show that outside the weak-excitation limit, and in regimes where fluorophores must undergo many rapid cycles of excitation and relaxation, non-fluorescing, excitation-induced triplet states with lifetimes orders of magnitude longer lived than photon-emitting singlet states degrade photon emission streams from both donor and acceptor fluorophores resulting in illumination-intensity-dependent changes in FRET efficiency. These changes are not commonly taken into consideration; therefore, robust strategies to suppress excited state accumulations are required to recover accurate and precise FRET efficiency, and thus distance, estimates. We propose both robust triplet state suppression and data correction strategies that enable the recovery of FRET efficiencies more closely approximating true values, thereby extending the spatial and temporal resolution of smFRET.

Time-dependent changes in conformation and composition underpin the function of diverse biochemical systems[1–10]. Single-molecule fluorescence (Förster) resonance energy transfer (smFRET) methods access this information by enabling measurements of nanometer scale changes in fluorescence energy transfer efficiency (FRET efficiency, $E$) between individual donor and acceptor fluorophores strategically attached to biologically informative sites on a given biomolecule (Fig. 1a). In low-abundance regimes, and at the single-molecule scale in particular, accurate and precise FRET measurements are aided by elevated illumination intensities that ensure photon emission streams sufficient to quantify rapid and transient biological processes via FRET and to reliably translate FRET efficiency into distance information (Fig. 1a)[10–21]. Elevated excitation rates are also required to maintain adequate photon

emission rates for FRET efficiency measurements at high time resolution. Advances that facilitate imaging at elevated illumination intensities are therefore expected to broaden the breadth of biological systems that can be examined and the scope of dynamic regimes that can be interrogated[4,9,20,22–25].

Robust extraction of inter-fluorophore distances ($R$) from FRET efficiency ($E$) depends on estimations of the photophysical parameters of donor and acceptor fluorophores at their sites of attachment[11–21]. Such parameters include the donor and acceptor spectral overlap integral ($J$), the fluorescence quantum yield of the donor fluorophore ($\varphi_D$) and the time-averaged donor and acceptor fluorophore mobilities, described by the orientation factor ($\kappa^2$). FRET efficiency calculations further require consideration of correction terms that account for

[1]Department of Structural Biology, St. Jude Children's Research Hospital, Memphis, TN, USA. [2]Department of Chemical Biology & Therapeutics, St. Jude Children's Research Hospital, Memphis, TN, USA. [3]Center for Photochemical Sciences and Department of Chemistry, Bowling Green State University, Bowling Green, OH, USA. [4]Present address: Department of Chemistry, Birla Institute of Technology and Science, Pilani, Rajasthan, India. [5]These authors contributed equally: Avik K. Pati, Zeliha Kilic. ✉e-mail: scott.blanchard@stjude.org

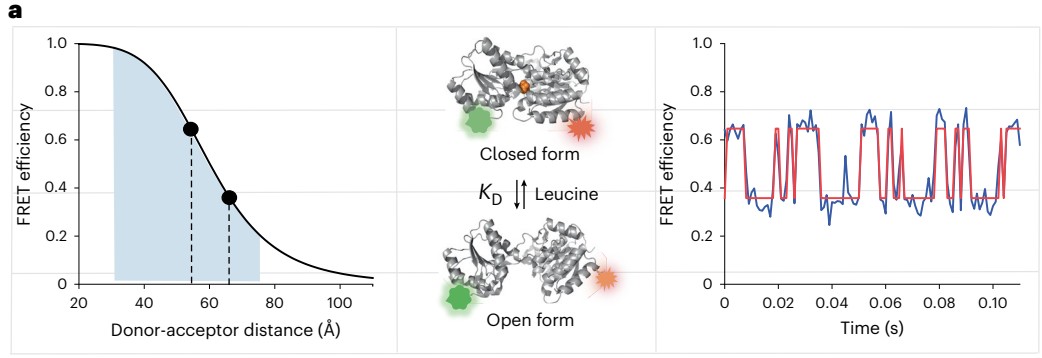

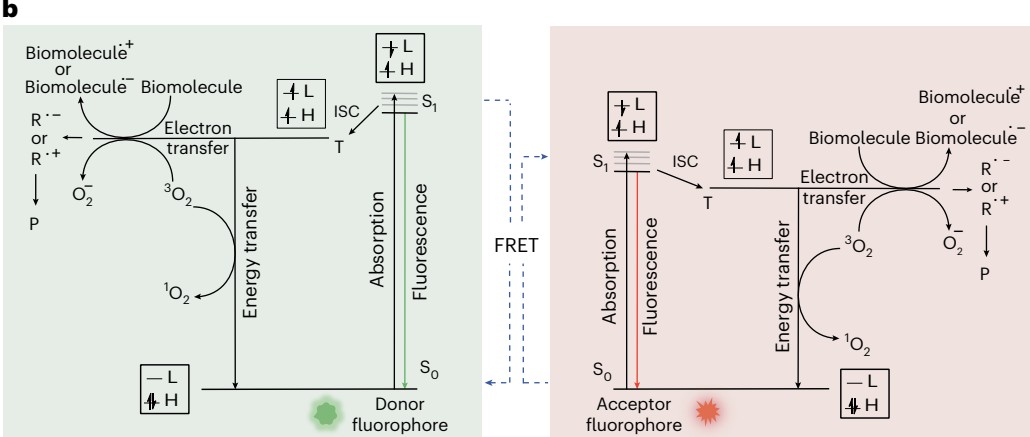

**Fig. 1 | High time resolution smFRET imaging. a**, smFRET measures distance within or between biomolecules that are site-specifically labeled with a donor and an acceptor fluorophore. When using Cy3 and Cy5 as donor and acceptor, FRET efficiency is most sensitive in the range ~30–70 Å (left). An example of functionally important structural dynamics of an individual biomolecule that can be interrogated by smFRET imaging, illustrated by crystal structures of open and closed forms of the clam-shell-type amino acid sensor protein (LIV-BP) that interconvert upon leucine binding (middle)[10]. Green and red stars represent donor and acceptor fluorophores, respectively. Experimental smFRET trace of LIV-BP dynamics captured at high time resolution (1 ms) using wide-field TIRF imaging (right)[10]. **b**, Jablonski diagram of donor (left) and acceptor (right) fluorophores. $S_0$, $S_1$ and T represent singlet ground state, first singlet excited state and triplet state of the fluorophores, respectively. ISC, $R^{.+}$ and $R^{.-}$ indicate intersystem crossing from $S_1 \rightarrow T$, radical cation and radical anion of the fluorophore species, respectively. P indicates a photobleached species. H and L indicate the highest occupied molecular orbital (HOMO) and lowest unoccupied molecular orbital (LUMO) of the fluorophores, respectively.

---

spectral crosstalk, relative fluorophore brightness ($\gamma$) and direct excitation of the acceptor fluorophore by the laser. While the importance of these quantities and correction terms are universally acknowledged[12,18,21], it is less often appreciated that illumination-intensity-dependent fluorophore photophysics are inherent to single-molecule FRET measurements[25–29] and that such phenomena should be taken into consideration when calculating FRET efficiencies[30,31].

Ideal organic fluorophores predominantly occupy ground states, while undergoing millions of excitation ($k_{ex}$) and emission cycles per second between singlet ground and excited states ($k_{S_1}^D$, $k_{S_1}^A$) to yield uninterrupted photon emissions for prolonged periods of time. In FRET-based investigations, the excited donor fluorophore can also relax to the ground state via FRET pathways ($k_{ET}$) when an acceptor fluorophore is close in space. In this regime, distances are extracted from FRET efficiency based on Förster theory[11–21,32] (equation (1)):

$$E_{theory} = \frac{k_{ET}}{k_{S_1}^D + k_{ET}} = \frac{1}{1 + \left(\frac{R}{R_0}\right)^6}, \tag{1}$$

where $R_0$ is the Förster radius, the inter-fluorophore distance at which 50% of donor relaxation from singlet excited states occurs via FRET to the acceptor fluorophore (Fig. 1b). $R_0$ includes terms for $\kappa^2$, $\varphi_D$, $J$ and the refractive index of the medium in which measurements are made ($n$) (equation (2)).

$$R_0 = 0.0211 \left(\frac{\kappa^2 \varphi_D}{n^4} J\right)^{\frac{1}{6}}, \tag{2}$$

In practice, approximations of FRET efficiency based on Förster theory ($E_{exp}$) are often achieved through experimental measures of fluorophore brightness (equation (3)):

$$E_{exp} = \frac{I_A}{I_A + \gamma^{exp} I_D}, \tag{3}$$

where $I_A$ and $I_D$ or $I_A^{exp}$ and $I_D^{exp}$ are the average number of photons experimentally detected in acceptor and donor channels, respectively after correcting for spectral crosstalk and acceptor direct excitation, where $\gamma^{exp}$ is the empirical correction factor that accounts for differences in quantum yield and detection efficiencies between donor and acceptor fluorophores (Methods)[12,18].

Unfortunately, common imaging probes (for example, cyanine- and rhodamine-class) can also relax from singlet excited states through intersystem crossing (ISC) into non-fluorescing triplet states that can be orders of magnitude longer lived (Fig. 1b). The probability of ISC scales with excitation rate. Previous studies demonstrate that

excited state accumulations, particularly triplet states, must be considered when making FRET efficiency calculations in elevated illumination intensity regimes, where triplet state accumulation tends to reduce FRET efficiency[29,30]. At elevated excitation rates and short inter-fluorophore distances, alternative energy transfer pathways between excited states tend to counter these effects[31]. These potentially complex excited state accumulation processes are not currently considered in standard-practice FRET efficiency estimation procedures[18–21].

Triplet state accumulations can be experimentally suppressed by imaging biological systems in oxygen-depleted solutions that contain millimolar concentrations of photostabilizing agents, typically referred to as photoprotective agents or triplet state quenchers (TSQs)[33–39]. Common TSQs include β-mercaptoethanol (BME), nickel ($Ni^{2+}$) complexes or cocktails of Trolox, cyclooctatetraene (COT) and 4-nitrobenzyl alcohol (NBA) or ascorbic acid (AA) and methyl viologen (MV)[23,35,37,38,40]. The addition of high concentrations of TSQs in solution can, however, exhibit irregularities linked to their varied and limited aqueous solubilities[41], and exert drawbacks related to their reactivities with fluorophore excited states[42], as well as their toxicities to the biological systems under investigation, particularly living cells[43]. To bypass such issues, intramolecular photostabilization methods referred to as 'self-healing' strategies have been advanced that link one or more TSQs proximal to the fluorophore[44–48] to efficiently mitigate triplet states. Self-healing fluorophores specifically leveraging the Baird aromatic properties of COT can reduce triplet state lifetimes by orders of magnitude, into sub-microsecond time regimes[43,46].

Here, we demonstrate using widely employed fluorophore pairs that experimentally determined FRET efficiencies vary as a function of illumination intensity. We find that the observed deviations principally arise from increased donor and acceptor fluorophore triplet state occupancy. We further show that self-healing strategies that robustly suppress triplet states together with judicious application of data correction strategies that account for triplet states enable the recovery of FRET efficiencies more closely approximating true values and substantially extend the spatial and temporal resolution of FRET measurements.

## Results

In the present investigations we examine the apparent FRET efficiencies of distinct cyanine- and rhodamine-class dye pairs widely employed in biological research[49,50], including the self-healing cyanine-class fluorophores LD555 and LD655 (Supplementary Fig. 1), as a function of illumination intensity[43,46]. As the present investigations seek only to examine the specific behavioral properties of the fluorophores themselves, not the underlying biological system, we examined well-established double-stranded DNA oligonucleotides site-specifically labeled with donor and acceptor fluorophores as model systems[18,38,43]. To image thousands of individual FRET dye pair-labeled biomolecules simultaneously over extended periods of time, we utilized wide-field, total internal reflection fluorescence (TIRF) microscopy equipped with single-frequency lasers and camera-based detection (Methods)[51]. To analogously examine freely diffusing FRET dye pair-labeled biomolecules at illumination intensities elevated beyond those that can be readily achieved by TIRF imaging, we employed continuous illumination confocal microscopy (Methods)[6]. In both cases, FRET efficiency calculations were made based on experimental measures of donor and acceptor fluorophore emission intensities.

### Illumination-intensity-dependent changes in FRET efficiency

We first imaged the cyanine fluorophores, Cy3 and Cy5, attached to a 21-nucleotide DNA duplex at positions yielding intermediate FRET efficiency by TIRF microscopy (Fig. 2a,b) using low illumination intensity (0.04 kW cm$^{-2}$) in enzymatically deoxygenated[52] imaging buffers (Methods). This condition yielded a total of ~260 detected photons from donor and acceptor fluorophores per 100 ms, more than ten-fold above shot noise, enabling reliable estimates of FRET efficiency.

Using standard correction procedures (equation (3))[18], we observed a corrected FRET efficiency value of ~0.39 (0.394 ± 0.003, where the uncertainty is the s.d. of mean FRET efficiency values from five experimental repeats) (Fig. 2c, left contour plot).

While FRET efficiency is typically held to be independent of the excitation rate[32], we observe that experimental FRET efficiency values decrease by more than 40% (~0.39 to ~0.23) over a 16-fold increase (0.04 to 0.64 kW cm$^{-2}$) in illumination intensity. Increasing illumination intensity an additional fivefold to ~3.60 kW cm$^{-2}$ further broadened and decreased the mean FRET efficiency distribution to ~51% of the value evidenced at low power (Fig. 2c,d). Similar trends, albeit to lesser extents, were also observed in ambient oxygenated imaging buffers (Supplementary Fig. 2). Illumination-intensity-dependent decreases in FRET efficiency were also observed using confocal illumination strategies (Supplementary Fig. 3). Consideration of these data in the context of extant literature led us to hypothesize that the experimentally observed deviations in FRET efficiency arise from donor and/or acceptor fluorophore triplet state occupancies[28–30,43,46,47,53].

### Exogenous TSQs ineffectively suppress triplet states

To test the hypothesis that triplet states are the main source of the illumination-intensity-dependent changes in FRET efficiency, we repeated our experiments in the presence of solution additives known to quench fluorophore triplet states (Supplementary Figs. 4 and 5). Consistent with established collisional quenching mechanisms[36–38], solution additives such as 143 mM BME; a cocktail of 1 mM COT, NBA and Trolox; or a cocktail of 1 mM AA and MV; and 0.4 mM $Ni^{2+}$ generally suppressed illumination-intensity-dependent changes in Cy3–Cy5 FRET efficiency. The cocktail of 1 mM COT, NBA and Trolox or 1 mM AA and MV performed best[38], reducing FRET efficiency changes from ~40% to ~11% at the highest illumination intensity examined (0.64 kW cm$^{-2}$; ~1,450 total photons per 100 ms; Supplementary Fig. 4); however, the estimated FRET efficiencies varied in both deoxygenated and oxygenated conditions depending on the TSQ cocktail employed (Supplementary Figs. 4 and 6). Similar variations were also evidenced for the rhodamine-class of FRET pair, ATTO550–ATTO647N (Supplementary Fig. 7). These data support the hypothesis that illumination-intensity-dependent changes in FRET efficiency arise principally from increasing donor and acceptor triplet state occupancy[30]. The data also demonstrate that solution additives fail to fully suppress triplet states, particularly at elevated illumination intensities, where the rates of ISC exceed triplet state quenching rates[43].

### Robust mitigation of triplet states via self-healing

The triplet states of self-healing LD555 and LD655 organic fluorophores are efficiently intramolecularly quenched through covalent linkage of a single COT molecule[43,46–48]. LD555 and LD655 exhibit nearly 100-fold shorter triplet state lifetimes, up to 25-fold higher photon count rates, as well as reduced rates of reactive oxygen species generation and photobleaching compared to Cy3 and Cy5 dyes[43,46]. Fluorescence correlation spectroscopy (FCS) measurements of rigidified Cy3 and Cy5 (Supplementary Fig. 8) suggest that Cy3 and Cy5 triplet state lifetimes in deoxygenated buffers (~31 ± 5 and ~51 ± 8 μs, respectively) are ~2–3-fold shorter-lived than observed for laser flash photolysis investigations in deoxygenated organic solvent[43,46,53] and ~30–200-fold longer lived than LD555 (~1.1 ± 0.1 μs) and LD655 (~0.20 ± 0.01 μs) (Supplementary Table 1)[43,46,53].

Congruent with triplet states being the principal determinant of illumination-intensity-dependent changes in FRET efficiency, the FRET distributions of LD555–LD655 labeled duplexes narrowed and remained largely unaltered in value as a function of illumination intensity in both deoxygenated (Fig. 2e,f and Supplementary Fig. 4) and oxygenated imaging buffers (Supplementary Fig. 9). At the highest illumination intensity examined in deoxygenated conditions (~3.6 kW cm$^{-2}$), where ~315 photons were detected per millisecond we observed only a relatively modest, ~10% reduction in FRET (Fig. 2e,f).

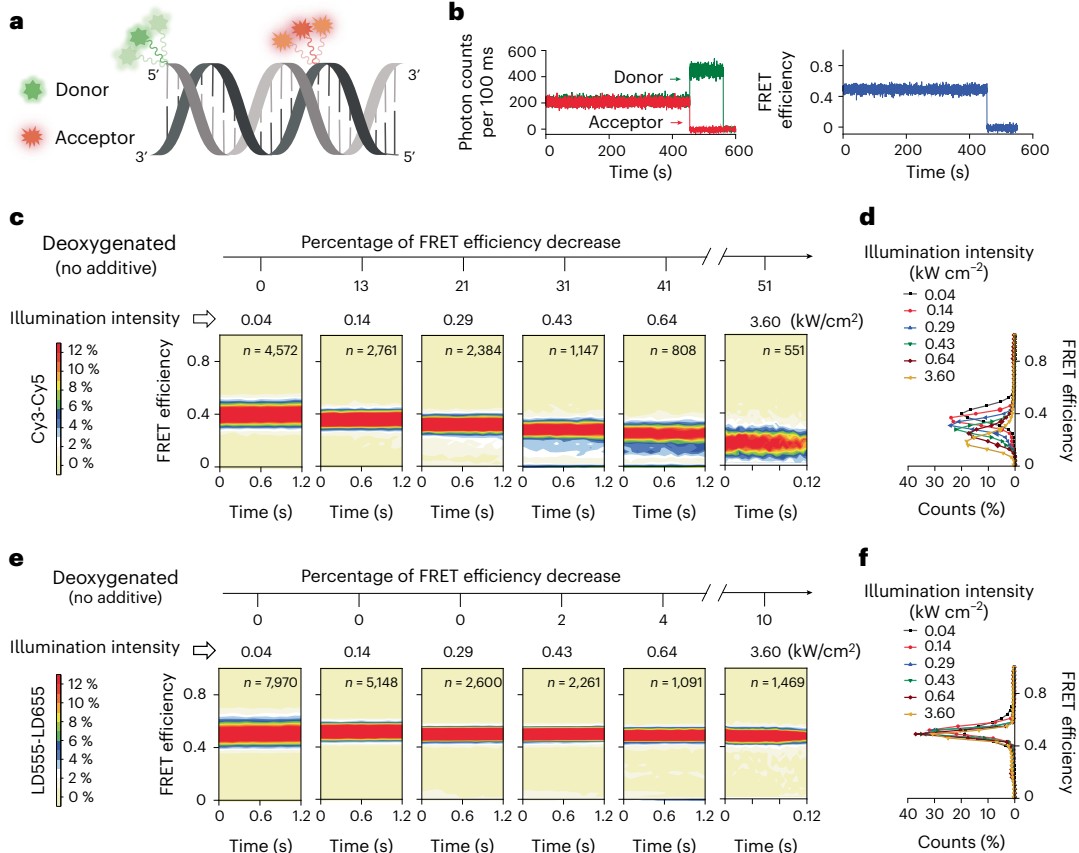

**Fig. 2 | Illumination-intensity-dependent changes in FRET efficiency.**
**a**, Schematic of the 21-nucleotide DNA duplex labeled with a donor dye at the 5′ terminus of one strand and an acceptor dye labeled at an internal position of the complementary strand separated by 14 nucleotides. **b**, Representative single-molecule fluorescence and smFRET trace of a donor- and acceptor-labeled DNA duplex. **c–f**, Three-dimensional population FRET histograms (contour plots) (**c,e**) and corresponding population FRET histograms (**d,f**) of Cy3–Cy5 and LD555–LD655 FRET pairs attached to the DNA duplex at various illumination intensities, respectively. All the data were collected in deoxygenated imaging buffers in the absence of any exogenous solution additives (photoprotective agents) at 100 ms time resolution using a custom-built TIRF imaging platform[51]. The data at 3.60 kW cm$^{-2}$ were collected at 5 ms time resolution.

## Unequal fluorophore saturation

To better understand the photophysical basis of illumination-intensity-dependent changes in FRET efficiency, we quantified the brightness of donor and acceptor dyes during FRET ($I_D^{FRET}$ and $I_A^{FRET}$, respectively) and the donor dye after acceptor photobleaching ($I_D^{No\ FRET}$) (Fig. 3a,b). These measurements were performed in deoxygenated imaging buffers, in the absence of solution additives. After crosstalk and acceptor direct excitation correction and before γ correction, this analysis revealed that both Cy3 and Cy5 brightness plateaued as a function of excitation rate and that Cy5 plateaued more rapidly than Cy3 (Fig. 3d,e). By contrast, over the same excitation range the brightness of both LD555 and LD655 increased nearly linearly as a function of illumination intensity (Fig. 3d,e). These findings suggest that illumination-intensity-dependent changes in FRET efficiency for the Cy3–Cy5 FRET pair likely arise from increasing disparities in donor and acceptor fluorophore brightness, which consequently impact γ-correction inputs into FRET efficiency calculations (equation (3)).

Consistent with this notion, we observed that the empirical γ-correction parameter[12] increased as a function of illumination intensity for the Cy3–Cy5 FRET pair, while remaining relatively constant for the LD555–LD655 FRET pair (Fig. 3c,f). Control studies of Cy3–LD655- and LD555–Cy5-labeled DNA oligonucleotides, where donor and acceptor triplet lifetimes are mismatched by orders of magnitude (~200 and ~40-fold, respectively), revealed that illumination-intensity-dependent γ-correction parameter impacts are predominantly attributed to the Cy3 donor fluorophore (Fig. 3f and Supplementary Fig. 10).

## Triplet states accumulation alters FRET efficiency

Motivated by these findings, we sought to establish a quantitative framework for understanding the relationship between the triplet state occupancies of donor and acceptor fluorophores and illumination-dependent changes in FRET efficiency. To do so, as described by Camley et al.[30] and Nettels et al.[31] (and references therein), we constructed single-fluorophore and smFRET stochastic simulations based on theoretical principles[54,55]. We based these simulations on the experimental construct exhibiting intermediate FRET efficiency and an excitation and emission framework for donor and acceptor fluorophores that each includes just singlet ground and excited states as well as a single-triplet excited state ($S_0^D$, $S_1^D$, $T^D$ and $S_0^A$, $S_1^A$, $T^A$, respectively) (Fig. 1b and Methods). We seeded our simulations with donor and acceptor excitation ($k_{ex}$) and singlet excited state relaxation rate constants ($k_{S_1}$) derived from experiment, triplet state relaxation ($k_{TR}$) and ISC rate constants ($k_{ISC}$) derived from literature[26,56–58] and boundary conditions obtained from FCS data for ($k_{TR}$), as well as energy transfer rate constants ($k_{ET}$) consistent with estimated inter-dye distance (Methods and Supplementary Tables 1 and 2). Photoisomerization processes were not included because illumination-intensity-dependent plateaus in brightness were also evidenced for Cy3B, a chemically rigidified Cy3 derivative that does not photoisomerize[49] (Supplementary Figs. 1, 8 and 11). This framework, which stipulates joint states for each combination of singlet ground, singlet excited and triplet states of both donor and acceptor fluorophores ($S_0^D S_0^A$, $S_0^D S_1^A$, $S_0^D T^A$; $S_1^D S_0^A$, $S_1^D S_1^A$, $S_1^D T^A$, $T^D S_0^A$, $T^D S_1^A$, and $T^D T^A$), yields a nine-state photophysical model for simulating the FRET efficiency of the donor and acceptor fluorophore pair (Fig. 4a and Methods).

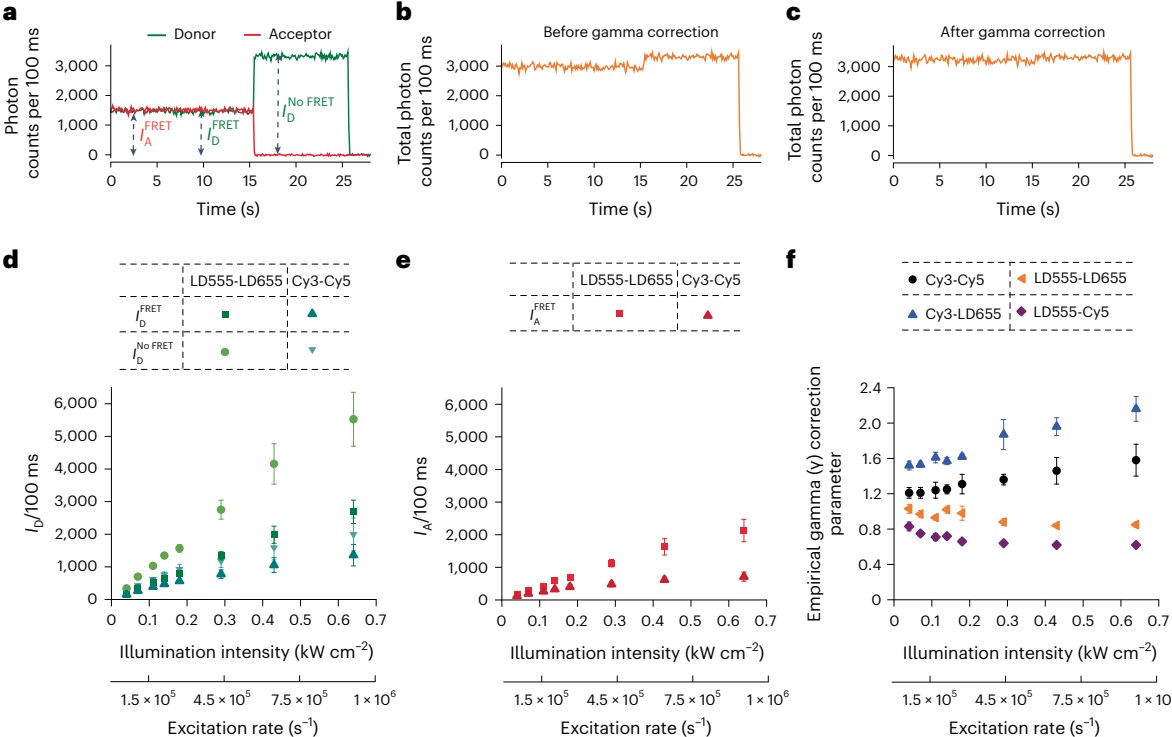

**Fig. 3 | Illumination-intensity-dependent changes in donor and acceptor brightness. a**, Representative single-molecule donor and acceptor fluorescence traces used for brightness calculations, which are crosstalk and acceptor direct excitation corrected, but not $\gamma$-corrected. $I_D^{FRET}$ and $I_A^{FRET}$ indicate brightness of the donor and acceptor fluorophore, respectively, during a FRET process. $I_D^{No\ FRET}$ indicates brightness of donor fluorophore after acceptor photobleaching. **b,c**, Total (donor + acceptor) intensity of the single-molecule trace before (**b**) and after (**c**) $\gamma$ correction. **d,e**, Changes of $\gamma$-uncorrected (**d**) donor and (**e**) acceptor brightness (photon counts per 100 ms frames) for Cy3–Cy5 and LD555–LD655 FRET pairs attached to the DNA duplex (Fig. 2a) with increasing illumination intensity. Data were collected in deoxygenated imaging buffers in the absence of

any exogenous solution additives (photoprotective agents) at 100 ms time resolution using a custom-built TIRF imaging platform[51]. Error bars represent the s.d. of mean intensity values from five experimental repeats. **f**, Variations of empirical $\gamma$ correction parameter of Cy3–Cy655, Cy3–Cy5, LD555–LD655 and LD555–Cy5-labeled DNA oligonucleotides imaged in deoxygenated imaging buffers in the absence of any exogenous solution additives at 100 ms time resolution at increasing illumination intensity. Error bars for Cy3–Cy5 and LD555–LD655 data represent the s.d. of mean $\gamma$ values from five experimental repeats. Error bars for Cy3–LD655 and LD555–Cy5 data represent the s.d. of mean $\gamma$ values from three experimental repeats.

We ran stochastic simulations of smFRET emission trajectories for Cy3–Cy5 and LD555–LD655 in the absence and presence of triplet states to examine whether this was sufficient to recapitulate experimental data. Control simulations of $E_{theory}$ (equation (1)), in which triplet states were not explicitly considered, showed that FRET efficiencies for both Cy3–Cy5 and LD555–LD655 FRET pairs remained constant as a function of illumination intensity (Fig. 4b,c). In contrast, and in line with experimental observations (Fig. 2c,d), the nine-state photophysical model including triplet states closely matched the illumination-intensity-dependent decreases in FRET efficiency for the Cy3–Cy5 FRET pair (Fig. 4b and Supplementary Fig. 12a). The extent of the observed decrease scaled with triplet state lifetime, where simulation closely matched experiment near the established triplet state lifetimes (Supplementary Fig. 12b). As observed experimentally (Fig. 4c), simulated FRET efficiency values for the LD555–LD655 FRET pair remained nearly independent of excitation rate (Supplementary Fig. 12c). Analogous simulations on Cy3–LD655- and LD555–Cy5-labeled DNA oligonucleotides confirmed the contributions of both donor and acceptor triplet states to FRET efficiency calculations (Methods and Supplementary Fig. 10). We therefore conclude that donor triplet state occupancy can degrade emission from both donor and acceptor fluorophores while also lowering FRET efficiencies using standard-practice correction procedures[12,18–21]. Increased donor triplet state occupancy also alters FRET efficiency by giving rise to alterations in Cy3 brightness after acceptor photobleaching compared to expectation that affect $\gamma$-correction procedures. Increased acceptor

triplet state occupancy lowers FRET efficiency by being increasingly unavailable for FRET[30]. Consideration of both donor and acceptor triplet states is thus critical to recapitulate experimentally observed illumination-intensity-induced changes in FRET efficiencies and empirical $\gamma$-correction parameters (Supplementary Fig. 12d). In addition, the distinct contributions of donor and acceptor triplet states also implies that the extent of FRET efficiency deviation should exhibit non-linear dependencies on the distance between donor and acceptor fluorophores.

**Correcting triplet state-mediated changes in FRET efficiency**

High spatiotemporal-resolution confocal and TIRF measurements, where acquisition rates of 100–1,000 s⁻¹ (1–10 ms integration times) are employed, require elevated illumination intensities to sustain photon emission rates sufficient for FRET measurements (>250 total detected photons per image frame)[10,43,59]. In this regime, experimental FRET efficiency calculations ($E_{exp}$) can fall short of recapitulating theory ($E_{theory}$), particularly when donor and acceptor fluorophores increasingly accumulate in dark excited states in the absence of robust triplet state mitigation strategies (Fig. 4d,e)[30].

We therefore considered whether inclusion of an additional correction parameter ($\zeta$) can be employed to account for triplet state-mediated deviations in FRET efficiency equation (3). Here, the $\zeta$-correction parameter is defined as $1 - \Lambda$; (in other words $\zeta = 1 - \Lambda$), where $\Lambda$ is the theoretical parameter described by Camley et al. that accounts for triplet state relaxation pathways[30]. For this analysis, we

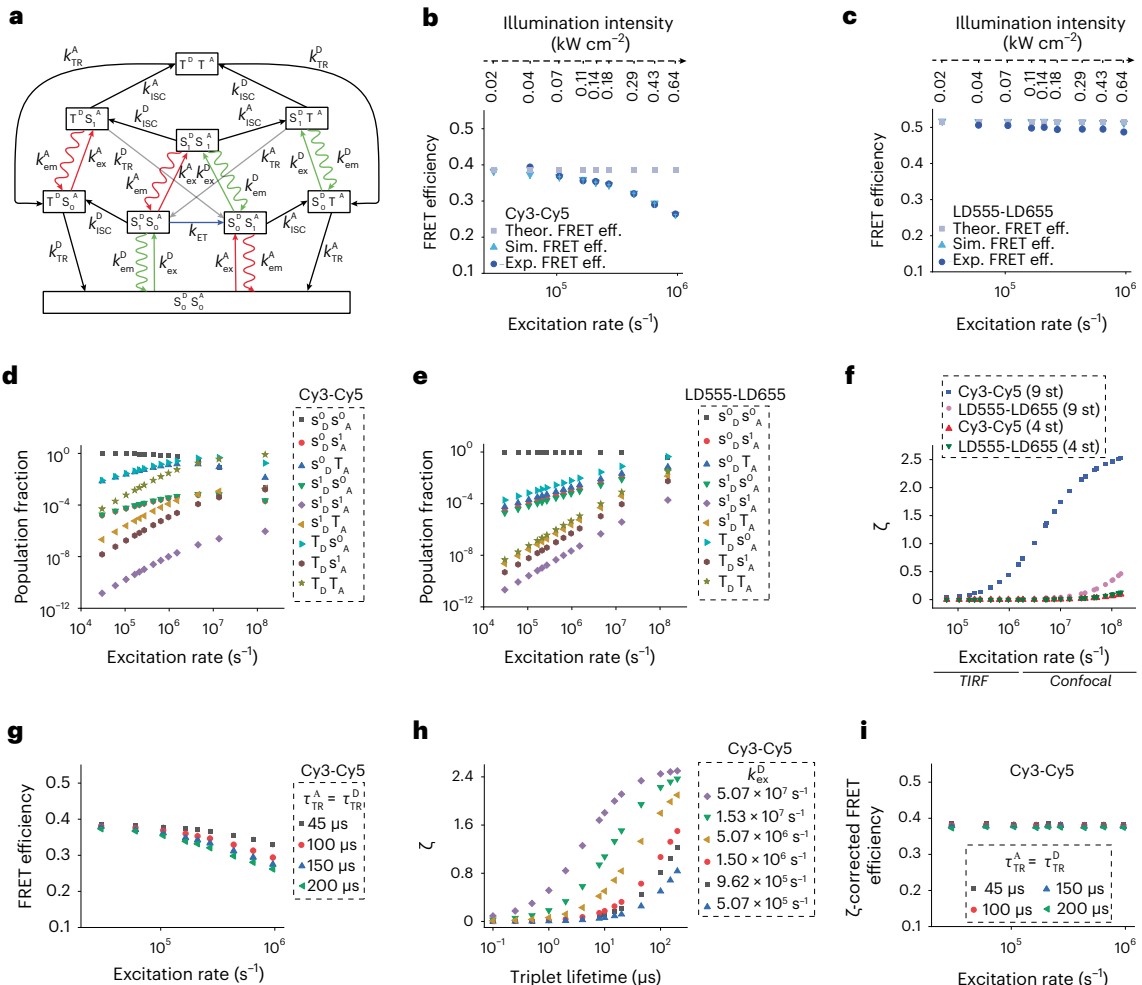

**Fig. 4 | Stochastic simulations of smFRET efficiency. a**, Schematic of a nine-state photophysical model used for stochastic simulations of smFRET efficiency. For simplicity in representation, energetic relations between excited states are not maintained in the vertical dimension. **b,c**, Simulations of smFRET efficiency for Cy3–Cy5 (**b**) and LD555–LD655 (**c**) FRET pairs. Theoretical FRET efficiencies were calculated based on Förster theory without consideration of triplet states (Methods). Error bars in experimental FRET data represent s.d. in mean FRET efficiency values from five experimental repeats. The experimental data are crosstalk, direct excitation and $\gamma$-corrected. **d,e**, Simulations of population fractions of nine joint states at different excitation rates for Cy3–Cy5 (**d**) and LD555–LD655 (**e**) pairs. **f**, Power-dependent variations in the $\zeta$-correction parameter based on the nine-state photophysical model for Cy3–Cy5 and LD555–LD655 FRET pairs over a broad range of illumination intensities spanning those commonly used for TIRF and confocal imaging. **g**, Simulation of smFRET efficiency for the Cy3–Cy5 FRET pair across donor-acceptor triplet state lifetimes and excitation rates. **h**, Variations of $\zeta$ for the Cy3–Cy5 FRET pair with varied donor and acceptor triplet state lifetimes and excitation rates. **i**, $\zeta$-corrected smFRET efficiencies for simulated Cy3–Cy5 FRET pairs at varied excitation rates.

again employed a nine-state photophysical framework[30] (Fig. 4a) to express FRET efficiency (denoted as $E_{9st}$; Methods) in explicit terms from first principles as shown in equation (4):

$$E_{9st} = \frac{\rho_A}{\rho_A + \gamma \rho_D} \qquad (4)$$

Here, $\rho_A$ and $\rho_D$ represent the mean rate of acceptor and donor photon detection during a given exposure period and $\gamma$ represents the $\gamma$-correction factor ($\gamma = \frac{\eta_A \varphi_A}{\eta_D \varphi_D}$) that accounts for intrinsic differences in donor and acceptor detection efficiencies ($\eta_A$, $\eta_D$) as well as quantum yields ($\varphi_A$, $\varphi_D$). For simplicity, this initial model does not include spectral crosstalk, direct acceptor fluorophore excitation or higher-order excited states. With this foundation, we can establish the relationship between theoretical FRET efficiency ($E_{theory}$) and the first-principle nine-state framework ($E_{9st}$) (equation (5) and Methods):

$$E_{theory} = \frac{1}{\frac{1}{E_{9st}} - \zeta} = \frac{\rho_A}{\rho_A(1-\zeta) + \gamma \rho_D}, \qquad (5)$$

where $\zeta$ represents a distance-naive correction parameter that accounts for triplet state occupancies (Fig. 4a and Methods section 'Protocols for $\zeta$-correction of FRET efficiency calculations'). In practical terms, FRET efficiencies ($E_{true}$) more closely approximating $E_{theory}$ can be expressed as equation (6):

$$E_{true} = \frac{I_A}{I_A(1-\zeta) + \gamma I_D}, \qquad (6)$$

where $I_A$ is the acceptor fluorescence intensity after corrections for spectral crosstalk and direct excitation[18,60].

Our experiments (Fig. 2c,d) and simulations (Fig. 4b,d) indicate that $\zeta$-corrections have the potential to be beneficial for measurements of the Cy3–Cy5 FRET pair in relatively low illumination intensity regimes ($k_{ex} = 10^4 - 10^6 \text{ s}^{-1}$) (Supplementary Figs. 13–15). For the LD555–LD655 FRET pair, inclusion of $\zeta$-corrections are not necessary until confocal illumination intensities are reached (ca. >$10^6 \text{ s}^{-1}$) (Fig. 4c,e,f). Kinetic simulations suggest that implementing $\zeta$-corrections can help recover FRET efficiencies more closely approximating theory for camera-based measurements when triplet state lifetimes exceed ~1 μs, ($k_{ex} < 10^6 \text{ s}^{-1}$)

(Fig. 4g–i). Unfortunately, the precision of $\zeta$-correction procedures diminish at distances shorter than those that have been explicitly examined (Fig. 2a), particularly at elevated illumination intensities ($k_{ex} > 10^6$ s$^{-1}$) due to excited state phenomena such as singlet–singlet annihilation (SSA), singlet–triplet annihilation (STA) and reverse inter-system crossing (RISC), which tend to mitigate triplets states[31]. In the presence of strong SSA, STA and RISC processes, $\zeta$-correction param-eters correspondingly plateau (Supplementary Fig. 16). Together with challenges associated with direct excitation of the acceptor fluoro-phore, SSA, STA and RISC processes render $\zeta$ corrections distance dependent, and thus impractical to apply.

### Self-healing enables high-spatial-resolution smFRET imaging

To demonstrate the advantages of the shortened triplet states for high-spatial and -temporal resolution smFRET imaging, we imaged a mixture of six DNA duplexes site-specifically labeled at 3-bp steps (Fig. 5a) using three distinct FRET pairs (Cy3–Cy5, LD555–LD655 and ATTO550–ATTO647N). In these experiments, imaging buffers were deoxygenated to support fluorophore longevity. Excitation rates were typical of a standard single-molecule TIRF measurement (~1,000 total detected photons per 100 ms).

 In this regime, solution TSQs were required to impart sufficient photostabilities to the Cy3–Cy5- and ATTO550–ATTO647N-labeled mixtures to achieve even modestly defined FRET distributions (Supplementary Fig. 17). In contrast, the LD555–LD655 FRET-labeled mixture clearly resolved all six FRET peaks to near-baseline resolution (Fig. 5b). Analogous results were observed when donor and acceptor positions on the DNAs were reversed (Supplementary Fig. 18a). These data were congruent with stochastic FRET simulations (Fig. 5c, Supple-mentary Fig. 18b and Methods). We note in this context that the width of experimental FRET efficiency distributions remain notably broader than the theoretical limit, which considers only photon emission sta-tistics (Fig. 5c). We attribute this broadening to experimental and data analysis noise, including optical aberrations, variabilities in donor and acceptor excitation and photon collection efficiencies, camera read noise and potential photophysical complexities not included in the nine-state framework (Fig. 4a). Given improved instrument detec-tion efficiencies and reductions in experimental noise, we therefore conclude that LD555–LD655-labeled species (or other self-healing fluorophores with sub-microsecond triplet state lifetimes) have the potential to robustly resolve distinct species at Ångstrom-scale resolu-tion (Supplementary Fig. 18b).

### Discussion

Despite progress toward the development of advanced imaging plat-forms and robust data analysis solutions in the field of dynamic struc-tural biology[7,11–21,25,51,61], our findings demonstrate that fluorophore photophysics represent a critical barrier to advancing the accuracy and spatiotemporal resolution of smFRET-based measurements. Stand-ard cyanine- and rhodamine-based FRET fluorophore pairs exhibit illumination-intensity-dependent variations in FRET efficiency, even in the presence of millimolar concentrations of TSQs, which are not accounted for by standard practices in the field[11–21]. Applying proce-dures put forward by Roy et al.[12] that are commonly employed for camera-based measurements, we observe that donor triplet state occupancy deteriorates data quality while skewing empirically derived $\gamma$-correction parameters, and thus corrected FRET efficiency values. Acceptor triplet state occupancy directly interferes with energy transfer to decrease FRET efficiency. In confocal microscopy studies, standard-practice correction procedures acknowledge the influence of fluorophore dark states and other processes via empirical correc-tions to the theoretical $\gamma$ values employed[18,21,62]. Effective strategies to reduce or eliminate fluorophore triplet states are therefore expected to be broadly enabling the full potential of the field of single-molecule imaging.

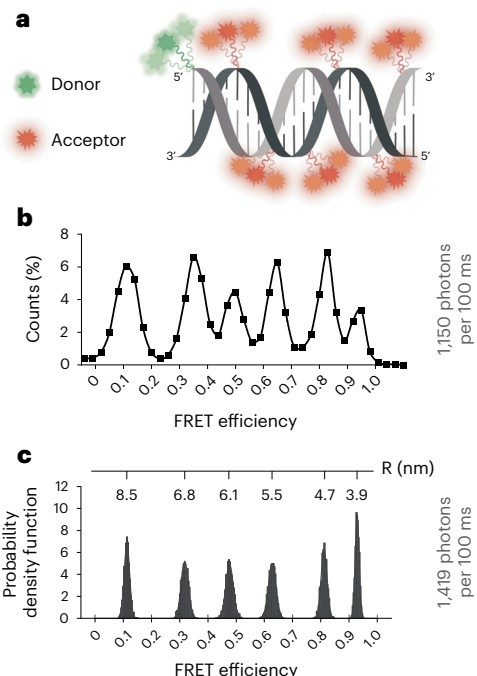

**Fig. 5 | High-spatial-resolution smFRET imaging. a**, Schematic diagram of a mixture of six DNA duplexes labeled with donor fluorophore at the 5′ terminus of one strand and acceptor fluorophore positioned at internal sites within the complementary strand separated by 5, 8, 11, 14, 17 and 20 nucleotides. **b**, Population FRET histograms generated from experiments, including all six DNA duplexes labeled with LD555–LD655 FRET pair in the absence of solution TSQs. All experiments were performed in deoxygenated imaging buffers at 100 ms time resolution using a custom-built TIRF imaging platform[51]. A single, empirically derived $\gamma$-correction parameter was used for FRET efficiency calculations. **c**, Simulated data for the LD555–LD655 FRET pair at 100 ms time resolution, which lack considerations of experimental noise. Inter-dye distance is shown with '*R*'. All imaging experiments and simulations were performed at 0.14 kW cm$^{-2}$.

Robust and accurate FRET efficiency measurements, even in rela-tively low-power regimes, require effective mitigation strategies that reduce both donor and acceptor triplet states by orders of magnitude (for example ~1 µs or shorter). Self-healing technologies demonstrate the most efficient and reliable method for reducing fluorophore triplet state lifetimes to this extent, yielding more precise and accurate FRET efficiency measurements across a range of illumination intensities without solution additives or $\zeta$-correction procedures. Further fluo-rophore engineering efforts, combined with strategies to model or control SSA, STA and RISC processes may help to further resolve true FRET efficiency values.

Robust reductions in excited state accumulation have the poten-tial to increase the spatial and temporal resolution limits of smFRET imaging substantially beyond what can be achieved today. As a result, a broader range of biological systems and scope of questions may be interrogated[4]. Progress on this front must include efforts to fur-ther idealize fluorophore performance as well as to further improve strategies to suppress instrument noise and more efficiently collect fluorescence emitted from biological samples. Achieving spatial and temporal resolution enhancements sufficient to make direct compari-sons between smFRET imaging and molecular dynamics simulations will be particularly vital to examining the molecular basis of structure–function relationships and mechanisms of biological regulation[4,7,63,64].

### Materials availability

The key materials used in this study are all commercially available.

## Reporting summary

Further information on research design is available in the Nature Portfolio Reporting Summary linked to this article.

## Online content

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

## Methods

### Calculations of steady-state populations for donor-only molecules

The rate of fluorophore excitation ($k_{ex}^D (s^{-1})$) was calculated using the following equations[58], where $I$ is the illumination intensity in W cm$^{-2}$, $h$ is Plank's constant, $f$ is the frequency of light, $\sigma_{abs}^D$ is the absorbance cross section of the donor fluorophore, $\varepsilon^D$ is the extinction coefficient of the donor fluorophore at the excitation wavelength (532 nm) and $N_A$ is Avogadro's number. For TIRF microscopy, the illumination intensity is made homogeneous in the $X,Y$ dimensions by dilation of the beam waist and we calculate the excitation rate considering that only a fraction of the beam is therefore incident on the field of view. For the $Z$ dimension, the fluorophores are expected to be at a uniform depth from the interface; the observed distribution of fluorescence intensities is consistent with this idea. We estimate that the fluorophores are roughly 28 nm from the interface and the evanescent field's penetration depth is approximately 120 nm. This suggests the error relative to a naive calculation (assuming the fluorophores are at the TIR interface) is less than 20%. For simplicity we neglect this specific effect.

$$k_{ex}^D = \frac{I}{hf}\sigma_{abs}^D$$

$$\sigma_{abs}^D = \frac{2303\varepsilon^D}{N_A}$$

Here, we demonstrate how the presence of triplet states affects the steady-state populations of the individual dyes as we vary the laser power or equivalently the excitation rate of the molecules. We assume that donor-only molecules occupy three photophysical states, the ground state, the singlet excited state and the triplet excited state, with fractional population $S_0^D$, $S_1^D$ and $T^D$, respectively. These population fractions vary over time; therefore, we can express each fraction as $S_0^D = S_0^D(t)$, $S_1^D = S_1^D(t)$, $T^D = T^D(t)$ for all times $t \geq 0$.

The temporal evolution of these population fractions is governed by the following set of equations:

$$\frac{dS_0^D}{dt} = -k_{ex}^D S_0^D + \left(k_{em}^D + k_{nr}^{S_0^D}\right)S_1^D + k_{TR}^D T^D,$$

$$\frac{dS_1^D}{dt} = k_{ex}^D S_0^D - \left(k_{em}^D + k_{nr}^{S_0^D} + k_{ISC}^D\right)S_1^D,$$

$$\frac{dT^D}{dt} = k_{ISC}^D S_1^D - k_{TR}^D T^D,$$

$$S_0^D + S_1^D + T^D = 1,$$

where $k_{ex}^D, k_{em}^D, k_{nr}^{S_0^D}, k_{TR}^D, k_{ISC}^D$ are the donor dye excitation rate, emission rate (radiative), non-radiative relaxation rate from $S_1^D$ to $S_0^D$, triplet state relaxation rate and ISC rates, respectively.

Here, the last equation guarantees the steady-state population fraction with initial populations fractions given as $\left(S_0^D(0)\ S_1^D(0)\ T^D(0)\right) = (1\ 0\ 0)$. If we gather these population fractions, $S_0^D, S_1^D, T^D$ in $\bar{P}(t) = \left(S_0^D(t)\ S_1^D(t)\ T^D(t)\right)$ and $\bar{P}_0 = \bar{P}(0) = (1\ 0\ 0)$, thereby the above set of equations can be written as: $\frac{d\bar{P}(t)}{dt} = \bar{P}(t)\overline{\overline{K}}, \bar{P}(0) = \bar{P}_0$, with the conservation law that is $\sum_{k=1}^3 P_k = 1$ and $\overline{\overline{K}} = \begin{pmatrix} -k_{ex}^D & k_{ex}^D & 0 \\ k_{fl}^{D,*} & -\left(k_{em}^D + k_{nr}^{S_0^D} + k_{ISC}^D\right) & k_{ISC}^D \\ k_{TR}^D & 0 & -k_{TR}^D \end{pmatrix}$. Then, we can write the solution to the above set of equations in closed form as follows $\bar{P}(t) = \bar{P}_0 \exp\left(\overline{\overline{K}}t\right)$, for all $t \geq 0$.

The solutions in our case can be explicitly computed and they are provided as $S_0^D = \frac{(k_{em}^D + k_{nr}^{S_0^D} + k_{ISC}^D)}{k_{ex}^D}S_1^D$, $S_1^D = \frac{1}{\frac{k_{em}^D + k_{nr}^{S_0^D} + k_{ISC}^D}{k_{ex}^D} + 1 + \frac{k_{ISC}^D}{k_{TR}^D}}$, $T^D = 1 - S_0^D - S_1^D$.

### Simulation of FRET efficiency time traces

In a prototypical smFRET experimental setup, a single molecule is attached with two fluorophores named donor and acceptor. Upon exciting the donor fluorophore, the donor fluorophore either emits photons or transfers its energy to the acceptor fluorophore. Subsequently, the acceptor fluorophore emits photons. These processes continue until both dyes photobleach. In our presentation, $S_0^D, S_1^D, T^D$ and $S_0^A, S_1^A, T^A$ denote the ground, excited and triplet states for donor and acceptor fluorophores, respectively. Photophysical parameters and Förster radius data of the dyes used in this study are tabulated in Supplementary Tables 1 and 2, respectively.

Here, we simulate the joint photophysical dynamics of the fluorophores as they visit one of the nine photophysical joint states (Fig. 4a), namely, $S_0^D S_0^A, S_1^D S_0^A, S_0^D T^A, S_1^D S_0^A, S_1^D S_1^A, S_1^D T^A, T^D S_0^A, T^D S_1^A, T^D T^A$ in continuous time as a Gillespie trajectory[65] where their joint photophysical dynamics are governed by the following generator matrix, unless otherwise specified

|  | $S_0^D S_0^A$ | $S_1^D S_0^A$ | $T^D S_0^A$ | $S_0^D S_1^A$ | $S_1^D S_1^A$ | $T^D S_1^A$ | $S_0^D T^A$ | $S_1^D T^A$ | $T^D T^A$ |
|---|---|---|---|---|---|---|---|---|---|
| $S_0^D S_0^A$ | $*$ | $k_{ex}^D$ | 0 | 0 | 0 | 0 | 0 | 0 | 0 |
| $S_1^D S_0^A$ | $k_{em}^D + k_{nr}^{S_0^D}$ | $*$ | $k_{ISC}^D$ | $k_{ET}^D$ | 0 | 0 | 0 | 0 | 0 |
| $T^D S_0^A$ | $k_{TR}^D$ | 0 | $*$ | 0 | 0 | 0 | 0 | 0 | 0 |
| $S_0^D S_1^A$ | $k_{em}^A + k_{nr}^{S_0^A}$ | 0 | 0 | $*$ | $k_{ex}^D$ | 0 | $k_{ISC}^A$ | 0 | 0 |
| $S_1^D S_1^A$ | 0 | $k_{em}^A + k_{nr}^{S_0^A}$ | 0 | $k_{em}^D + k_{nr}^{S_0^D}$ | $*$ | $k_{ISC}^D$ | 0 | $k_{ISC}^A$ | 0 |
| $T^D S_1^A$ | 0 | 0 | $k_{em}^A + k_{nr}^{S_0^A}$ | $k_{TR}^D$ | 0 | $*$ | 0 | 0 | $k_{ISC}^A$ |
| $S_0^D T^A$ | $k_{TR}^A$ | 0 | 0 | 0 | 0 | 0 | $*$ | $k_{ex}^D$ | 0 |
| $S_1^D T^A$ | 0 | $k_{TR}^A$ | 0 | 0 | 0 | 0 | $k_{em}^D + k_{nr}^{S_0^D}$ | $*$ | $k_{ISC}^D$ |
| $T^D T^A$ | 0 | 0 | $k_{TR}^A$ | 0 | 0 | 0 | $k_{TR}^D$ | 0 | $*$ |

where the $*$ notation represents the negative of the sum of each off-diagonal row element with the assumption that only the donor fluorophore undergoes the photoexcitation. Here, we note that we have not included SSA, STA or RISC events[31]. Later, we consider (1) SSA by introducing $S_1^D S_1^A \xrightarrow{k_{SSA}} S_0^D S_1^A$, (2) STA through $S_1^D T^A \xrightarrow{k_{STA}} S_0^D T^A$ and finally (3) STA ($S_1^D T^A \xrightarrow{k_{STA}} S_0^D T^A$) with RISC through $S_1^D T^A \xrightarrow{k_{STA}} S_0^D S_1^A$. In the absence of the conformational kinetics of the molecule to which the fluorophores are attached, the emitted photons are only governed by the coupled photophysical dynamics of the fluorophores.

Given the electronic states of the dyes and the rates connecting them (Supplementary Table 1), upon having high donor excitation rate ($k_{ex}^D$), we observe that, $S_1^D T^A$ gets highly populated for many excitation cycles. Thereby, this leads to a decrease in the emitted acceptor photon counts and an increase in the donor photon counts. Subsequently, we obtain a decreasing pattern in FRET efficiency. We define the excitation rates causing the decrease in FRET efficiency as the rates 'outside the weak-excitation limit'[30].

From the generator matrix, we can readily write down the theoretical FRET efficiency ($E_{theory}^{9st}$) that is the rate determining the transition from $S_1^D S_0^A$ to $S_0^D S_1^A$ given with the following equation:

$$E_{theory}^{9st} = \frac{k_{ET}^D}{k_{ET}^D + k_{S_1}^D} = \frac{k_{ET}^D}{k_{ET}^D + k_{em}^D + k_{nr}^{S_0^D} + k_{ISC}^D}.$$

In Supplementary Table 3, we summarize the difference between the theoretical FRET efficiency calculation for models with four joint states, where triplet states are removed, and nine joint states.

In our simulations we consider 20% detection efficiency, estimated for both green and red channels from the transmission specifications of the instrument's individual components and we compare the simulation with the experimental data corrected for acceptor direct excitation and crosstalk. The effect of acceptor fluorophore direct excitation by the donor laser (5%) on the theoretical FRET efficiency is also considered. Correspondingly, we correct for direct excitation by subtracting 5% of the total emission from acceptor emission. Although reports by Huang et al.[66] may suggest that excited state processes related to STA may feature strongly in the photophysical properties of Cy3–Cy5 and LD555–LD655 FRET pairs, recent reports by Zheng et al.[46,53] clarify that the triplet absorbance band of Cy5 is ~690 nm and not a mixture of ~625 nm and ~690 nm bands as Huang et al. propose[66]. We therefore consider direct excitation of acceptor triplet states negligible in our treatments when using a 630 nm laser.

The only remaining correction for the simulated FRET efficiency ($E_{sim}$) that needs to be addressed to agree with the experimental FRET efficiency ($E_{exp}$) $\left[ E_{exp} = \frac{I_A^{exp}}{I_A^{exp} + \gamma^{exp} I_D^{exp}} \right]$ is the empirical $\gamma$-correction parameter derived from an experiment[12] ($\gamma^{exp}$) accounting for the differences in fluorescence quantum yields of both fluorophores. $I_A^{exp}$ and $I_D^{exp}$ are the average number of photons experimentally detected in acceptor and donor channels, respectively upon correcting $I_A'$ for direct excitation and crosstalk such that $I_A^{exp} = I_A' - \alpha I_D' - \delta(I_D' + I_A')$ and $I_D^{exp} = I_D'$ (refs. 18,67). Here, $I_A'$ and $I_D'$ represent the baseline-subtracted donor and acceptor intensities; $\alpha$ and $\delta$ represent the spectral bleed-through from donor to acceptor channel and fraction of acceptor direct excitation with 532 nm laser wavelength. We note that in the main text (equation (3)), we dropped the superscript from $I_A^{exp}$ and $I_D^{exp}$ (written as $I_A$ and $I_D$) to avoid notational complexity as we did not introduce the notations for simulation $E_{sim}$.

In our simulations as provided in previous literature[12] in the absence of FRET, we simulate trajectories for both donor and acceptor fluorophores. Then, we compute the ratio of differences for the mean photon counts that are computed in the absence of FRET

($< I_A^{sim,No FRET} >, < I_D^{sim,No FRET} >$) across many molecule trajectories and presence of FRET ($< I_A^{sim,FRET} >, < I_D^{sim,FRET} >$) for acceptor and donor fluorophores that is $\gamma^{sim} = \frac{|< I_A^{sim,FRET} > - < I_A^{sim,No FRET} >|}{|< I_D^{sim,FRET} > - < I_D^{sim,No FRET} >|}$. Here, the '$<\cdot>$' operator represents the average across mean detected photon counts of simulated or experimental trajectories. This ratio provides us with the gamma factor ($\gamma^{sim}$) from which we compute: $E_{sim} = \frac{I_A^{sim}}{I_A^{sim} + \gamma^{sim} I_D^{sim}}$.

We then carried out corrections on the FRET efficiency after correcting for direct excitation as well as crosstalk. We followed the same photophysical trajectory generation steps for both donor and acceptor fluorophores for the simulations in Fig. 5c. We simulated ten donor and acceptor photophysical trajectories corresponding to the inter-dye distances (provided by MD simulation) at 100 ms temporal resolution (0.14 kW cm$^{-2}$ illumination intensity) with a total simulation time of 10 s. Afterwards, these trajectories were $\gamma$-corrected according to the Roy et al.[12] and FRET efficiency $\left( E_{sim} = \frac{I_A^{sim}}{I_A^{sim} + \gamma^{sim} I_D^{sim}} \right)$ trajectories were calculated. These FRET efficiency trajectories were acquired at every 100 ms and these FRET efficiency values were used to populate histograms.

In our current study, the only noise source is the shot noise of photon emission. We have not considered any noise from the detector used in the experiments. Additional improvements related to the measurement models in our simulations are left for future studies. We summarize the main quantity of interests in tables (Supplementary Tables 4–6).

## Correcting FRET efficiency outside the weak-excitation limit

Given a single donor-acceptor dye pair, the theoretical energy transfer efficiency describing the single donor excitation event will be transferred from $S_1^D S_0^A$ to $S_0^D S_1^A$ according to both four-state and nine-state photophysical models by $E_{theory}^{4st} \equiv \frac{k_{ET}^D}{k_{ET}^D + k_{S_1}^D}$ and $E_{theory}^{9st} \equiv \frac{k_{ET}^D}{k_{ET}^D + k_{S_1}^D}$. We can also write $E_{theory}^{4st,9st} \equiv \frac{k_{ET}^D}{k_{ET}^D + k_{S_1}^D} = \frac{1}{1 + \left( \frac{R}{R_0} \right)^6}$ with the help of the Förster relationship $k_{ET}^D = k_{S_1}^D \left( \frac{R_0}{R} \right)^6$. In practice, $E$ is obtained as an approximation to $E_{theory}^{4st}$ and $E_{theory}^{9st}$ from photon counting (photon counts are labeled with $I_A$ and $I_D$ in acceptor and donor channels, respectively)[68] and computed as $E_{4st} \equiv \frac{I_A}{I_A + \gamma I_D} = \frac{\rho_A}{\rho_A + \left( \frac{\varphi_A}{\varphi_D} \right) \rho_D}$ or $E_{9st} \equiv \frac{I_A}{I_A + \gamma I_D} = \frac{\rho_A}{\rho_A + \left( \frac{\varphi_A}{\varphi_D} \right) \rho_D}$ where $\rho_A$ and $\rho_D$ denote the photon emission rates, while $\varphi_A$ and $\varphi_D$ denote the quantum yields in acceptor and donor channels, respectively. Furthermore, $\gamma = \frac{\eta_A \varphi_A}{\eta_D \varphi_D}$, also known as the $\gamma$-correction parameter, accounts for differences in detection efficiencies of the donor and acceptor fluorophores ($\eta_D$ and $\eta_A$, respectively) and their quantum yields ($\varphi_D$ and $\varphi_A$, respectively).

Increased excitation rates give rise to populating other electronic states for individual dyes ($S_0^D, S_1^D, T^D$ and $S_0^A, S_1^A, T^A$), altering the occupation of states ($S_0^D S_0^A, S_0^D S_1^A, S_0^D T^A, S_1^D S_0^A, S_1^D S_1^A, S_1^D T^A, T^D S_0^A, T^D S_1^A, T^D T^A$) for the dye pairs[69,70]. Consequently, $E_{4st} \nrightarrow E_{theory}^{4st}$ and $E_{9st} \nrightarrow E_{theory}^{9st}$ ($E_{4st}$ and $E_{9st}$ does not equal or converge to $E_{theory}^{4st}$ and $E_{theory}^{9st}$, respectively). Here, we provide $E_{theory}^{4st}$ and $E_{theory}^{9st}$ for two cases, which we define without inclusion of crosstalk and direct excitation of the acceptor fluorophore considerations to simplify expressions for the $\zeta$-correction factor. Correspondingly, $\zeta$-correction, as defined, must only be applied to experimental data that that have already been normalized for crosstalk and direct acceptor fluorophore excitation (see section 'Protocols for $\zeta$-correction of FRET efficiency calculations'). As emphasized in the main text, $\zeta$-correction should also be implemented with caution, after

efforts have been made to suppress triplet states, particularly when donor and acceptor fluorophore brightness exhibit non-linear changes as a function of illumination intensity as this is a tell-tale signature of excited state accumulations that may be difficult to account for (Fig. 3d,e).

Case 1 ($E_{4st}$): both donor and acceptor fluorophores have only singlet excited and ground states and hence the system has four states for the dye pairs (Supplementary Scheme 1).

Case 2 ($E_{9st}$): both donor and acceptor fluorophore have three electronic states, including singlet excited, ground and triplet states and thereby the dye pair has nine states (Supplementary Scheme 2).

The differential equation formulation governing the system of single dye pair's states is given as:

$$\frac{dS_0^D S_0^A}{dt} = -k_{ex}^D S_0^D S_0^A + k_{S_1}^D S_1^D S_0^A + k_{S_1}^A S_0^D S_1^A,$$

$$\frac{dS_1^D S_0^A}{dt} = k_{ex}^D S_0^D S_0^A - \left(k_{S_1}^D + k_{ET}^D\right) S_1^D S_0^A + k_{S_1}^A S_1^D S_1^A,$$

$$\frac{dS_0^D S_1^A}{dt} = k_{ET}^D S_1^D S_0^A - \left(k_{ex}^D + k_{S_1}^A\right) S_0^D S_1^A + k_{S_1}^D S_1^D S_1^A,$$

$$\frac{dS_1^D S_1^A}{dt} = k_{ex}^D S_0^D S_1^A - \left(k_{S_1}^D + k_{S_1}^A\right) S_1^D S_1^A,$$

$$S_0^D S_0^A + S_1^D S_0^A + S_0^D S_1^A + S_1^D S_1^A = 1.$$

We can write these equations via the Master equation as follows $\frac{d\bar{S}(t)}{dt} = \bar{S}(t)\overline{\overline{K}}$, $\bar{S}(0) = \bar{S}_0$ with the conservation law that is $\sum_{k=1}^4 S_k = 1$, where $\bar{S} = \left(S_0^D S_0^A, S_1^D S_0^A, S_0^D S_1^A, S_1^D S_1^A\right)$ and $\overline{\overline{K}}$ is provided below:

$$\overline{\overline{K}} = \begin{array}{c|cccc} & S_0^D S_0^A & S_1^D S_0^A & S_0^D S_1^A & S_1^D S_1^A \\ \hline S_0^D S_0^A & * & k_{ex}^D & 0 & 0 \\ S_1^D S_0^A & k_{S_1}^D & * & k_{ET}^D & 0 \\ S_0^D S_1^A & k_{S_1}^A & 0 & * & k_{ex}^D \\ S_1^D S_1^A & 0 & k_{S_1}^A & k_{S_1}^D & * \end{array}$$

Subsequently, we solve the equation for steady-state probabilities $\left(\frac{d\bar{S}(t)}{dt} = \bar{S}(t)\overline{\overline{K}} = 0\right)$ that gives us the measured FRET efficiency based on $E_{4st} \equiv \frac{\rho_A}{\rho_A + \left(\frac{\varphi_A}{\varphi_D}\right)\rho_D}$ such that $\rho_D = \eta_D k_{S_1}^D \varphi_D \left(S_1^D S_0^A + S_1^D S_1^A\right)$, $\rho_A = \eta_A k_{S_1}^A \varphi_A \left(S_0^D S_1^A + S_1^D S_1^A\right)$.

When we substitute the exact expressions for $\rho_D$ and $\rho_A$ in $E_{4st} \equiv \frac{\rho_A}{\rho_A + \left(\frac{\varphi_A}{\varphi_D}\right)\rho_D}$, we find that $E_{4st} \equiv \frac{1}{1 + \frac{\varphi_A}{\varphi_D}\left(\frac{k_{S_1}^D}{k_{ET}^D} + \frac{k_{ex}^D k_{S_1}^D}{k_{S_1}^A\left(k_{ex}^D + k_{S_1}^A + k_{S_1}^D\right)}\right)}$ simplifies to

$E_{4st} \equiv \frac{1}{1 + \frac{k_{S_1}^D}{k_{ET}^D} + \frac{k_{ex}^D k_{S_1}^D}{k_{S_1}^A\left(k_{ex}^D + k_{S_1}^A + k_{S_1}^D\right)}}$ as shown by Camley et al.[30] Subsequently, upon applying the Förster formula $k_{ET}^D = k_{S_1}^D \left(\frac{R_0}{R}\right)^6$, we obtain:

$$E_{4st} \equiv \frac{1}{1 + \frac{k_{S_1}^D}{k_{S_1}^D\left(\frac{R_0}{R}\right)^6} + \frac{k_{ex}^D k_{S_1}^D}{k_{S_1}^A\left(k_{ex}^D + k_{S_1}^A + k_{S_1}^D\right)}} = \frac{1}{1 + \frac{1}{\left(\frac{R_0}{R}\right)^6} + \frac{k_{ex}^D k_{S_1}^D}{k_{S_1}^A\left(k_{ex}^D + k_{S_1}^A + k_{S_1}^D\right)}}$$

$$= \frac{1}{1 + \left(\frac{R}{R_0}\right)^6 + \frac{k_{ex}^D k_{S_1}^D}{k_{S_1}^A\left(k_{ex}^D + k_{S_1}^A + k_{S_1}^D\right)}} \neq \frac{1}{1 + \left(\frac{R}{R_0}\right)^6}.$$

$$\zeta = \frac{k_{ex}^D k_{S_1}^D}{k_{S_1}^A\left(k_{ex}^D + k_{S_1}^D + k_{S_1}^D\right)}$$

In the four-state model, $\frac{k_{ex}^D k_{S_1}^D}{k_{S_1}^D\left(k_{ex}^D + k_{S_1}^D + k_{S_1}^D\right)} \ll 1$ (reads as $\frac{k_{ex}^D k_{S_1}^D}{k_{S_1}^A\left(k_{ex}^D + k_{S_1}^A + k_{S_1}^D\right)}$ is much less than 1) is assumed due to $k_{ex}^D \ll k_{S_1}^D, k_{S_1}^A$. Thereby, in the absence of $k_{exc}^D \ll k_{S_1}^D, k_{S_1}^A$, $E_{4st}$ becomes different from $\frac{1}{1 + \left(\frac{R}{R_0}\right)^6}$ that is assumed to hold under weak excitation limit.

Similarly, we can calculate empirical $\gamma$ correction through steady-state mean detection fractions ($\rho_A$, $\rho_D$) shown as $\gamma_{theory}^{4st}$ by substituting the exact expressions for $\rho_D$ and $\rho_A$ in $\gamma_{theory}^{4st} = \frac{\rho_A - \rho_A^{NoFRET}}{\rho_D^{NoFRET} - \rho_D}$ where $\rho_A^{No\,FRET}$ and $\rho_D^{No\,FRET}$ are obtained by simply setting $k_{ET}^D = 0$. We obtain $\gamma_{theory}^{4st} = \frac{\eta_A k_{ex}^D}{\eta_D k_{em}^D}\varphi_A + \gamma$ with $\gamma = \frac{\eta_A \varphi_A}{\eta_D \varphi_D}$.

$$\frac{dS_0^D S_0^A}{dt} = -k_{ex}^D S_0^D S_0^A + \left(k_{em}^D + k_{nr}^{S_0^D}\right) S_1^D S_0^A + \left(k_{em}^A + k_{nr}^{S_0^A}\right) S_0^D S_1^A$$
$$+ k_{TR}^A S_0^D T^A + k_{TR}^D T^D S_0^A,$$

$$\frac{dS_1^D S_0^A}{dt} = k_{ex}^D S_0^D S_0^A - \left(k_{em}^D + k_{nr}^{S_0^D} + k_{ET}^D + k_{ISC}^D\right) S_1^D S_0^A$$
$$+ \left(k_{em}^A + k_{nr}^{S_0^A}\right) S_1^D S_1^A,$$

$$\frac{dT^D S_0^A}{dt} = k_{ISC}^D S_1^D S_0^A + \left(k_{em}^A + k_{nr}^{S_0^A}\right) T^D S_1^A - k_{TR}^D T^D S_0^A,$$

$$\frac{dS_0^D S_1^A}{dt} = k_{ET}^D S_1^D S_0^A - \left(k_{ex}^D + k_{em}^A + k_{nr}^{S_0^A} + k_{ISC}^A\right) S_0^D S_1^A$$
$$+ \left(k_{em}^D + k_{nr}^{S_0^D}\right) S_1^D S_1^A + k_{TR}^D T^D S_1^A,$$

$$\frac{dS_1^D S_1^A}{dt} = k_{ex}^D S_0^D S_1^A - \left(k_{ISC}^A + k_{em}^A + k_{nr}^{S_0^A} + k_{em}^D + k_{nr}^{S_0^D} + k_{ISC}^D\right) S_1^D S_1^A,$$

$$\frac{dT^D S_1^A}{dt} = k_{ISC}^D S_1^D S_1^A - \left(k_{TR}^D + k_{em}^A + k_{nr}^{S_0^A} + k_{ISC}^A\right) T^D S_1^A,$$

$$\frac{dS_0^D T^A}{dt} = k_{ISC}^A S_0^D S_1^A - \left(k_{TR}^A + k_{ex}^D\right) S_0^D T^A$$
$$+ \left(k_{em}^D + k_{nr}^{S_0^D}\right) S_1^D T^A + k_{TR}^D T^D T^A,$$

$$\frac{dS_1^D T^A}{dt} = k_{ISC}^A S_1^D S_1^A + k_{ex}^D S_0^D T^A - \left(k_{TR}^A + k_{em}^D + k_{nr}^{S_0^D} + k_{ISC}^D\right) S_1^D T^A,$$

$$\frac{dT^D T^A}{dt} = k_{ISC}^A T^D S_1^A + k_{ISC}^D S_1^D T^A - \left(k_{TR}^A + k_{TR}^D\right) T^D T^A,$$

$$S_0^D S_0^A + S_1^D S_0^A + T^D S_0^A + S_0^D S_1^A + S_1^D S_1^A + T^D S_1^A + S_0^D T^A + S_1^D T^A + T^D T^A = 1.$$

$$\overline{\overline{K}} = $$

| | $S_0^D S_0^A$ | $S_1^D S_0^A$ | $T^D S_0^A$ | $S_0^D S_1^A$ | $S_1^D S_1^A$ | $T^D S_1^A$ | $S_0^D T^A$ | $S_1^D T^A$ | $T^D T^A$ |
|---|---|---|---|---|---|---|---|---|---|
| $S_0^D S_0^A$ | $*$ | $k_{ex}^D$ | 0 | 0 | 0 | 0 | 0 | 0 | 0 |
| $S_1^D S_0^A$ | $\left(k_{em}^D + k_{nr}^{S_0^D}\right)$ | $*$ | $k_{ISC}^D$ | $k_{ET}^D$ | 0 | 0 | 0 | 0 | 0 |
| $T^D S_0^A$ | $k_{TR}^D$ | 0 | $*$ | 0 | 0 | 0 | 0 | 0 | 0 |
| $S_0^D S_1^A$ | $\left(k_{em}^A + k_{nr}^{S_0^A}\right)$ | 0 | 0 | $*$ | $k_{ex}^D$ | 0 | $k_{ISC}^A$ | 0 | 0 |
| $S_1^D S_1^A$ | 0 | $\left(k_{em}^A + k_{nr}^{S_0^A}\right)$ | 0 | $\left(k_{em}^D + k_{nr}^{S_0^D}\right)$ | $*$ | $k_{ISC}^D$ | 0 | $k_{ISC}^A$ | 0 |
| $T^D S_1^A$ | 0 | 0 | $\left(k_{em}^A + k_{nr}^{S_0^A}\right)$ | $k_{TR}^D$ | 0 | $*$ | 0 | 0 | $k_{ISC}^A$ |
| $S_0^D T^A$ | $k_{TR}^A$ | 0 | 0 | 0 | 0 | 0 | $*$ | $k_{ex}^D$ | 0 |
| $S_1^D T^A$ | 0 | $k_{TR}^A$ | 0 | 0 | 0 | 0 | $\left(k_{em}^D + k_{nr}^{S_0^D}\right)$ | $*$ | $k_{ISC}^D$ |
| $T^D T^A$ | 0 | 0 | $k_{TR}^A$ | 0 | 0 | 0 | $k_{TR}^D$ | 0 | $*$ |

Next, we proceed with solving the equation for steady-state probabilities $\left(\frac{d\bar{s}(t)}{dt} = \bar{s}(t)\overline{\overline{K}} = 0\right)$, where

$$\bar{s}(t) = \left(S_0^D S_0^A, S_1^D S_0^A, T^D S_0^A, S_0^D S_1^A, S_1^D S_1^A, T^D S_1^A, S_0^D T^A, S_1^D T^A, T^D T^A\right)$$

that gives us the measured FRET efficiency based on $E_{9st} \equiv \frac{\rho_A}{\rho_A + \left(\frac{\varphi_A}{\varphi_D}\right)\rho_D}$ such that

$$\rho_D = \eta_D k_{S_1}^D \varphi_D\left(S_1^D S_0^A + S_1^D S_1^A + S_1^D T^A\right), \quad \rho_A = \eta_A k_{S_1}^A \varphi_A\left(S_0^D S_1^A + S_1^D S_1^A + T^D S_1^A\right).$$

When substituting the exact expressions for $\rho_D$ and $\rho_A$ in $E_{9st} \equiv \frac{\rho_A}{\rho_A + \left(\frac{\varphi_A}{\varphi_D}\right)\rho_D}$, we find that

$$E_{9st} \equiv \cfrac{1}{1 + \cfrac{k_{S_1}^D}{k_{ET}^D} + \cfrac{k_{ex}^D k_{S_1}^D \left(\left(k_{S_1}^A k_{ISC}^A + \left(k_{S_1}^D + k_{TR}^A\right)\left(k_{ISC}^A + k_{TR}^A\right)\right)\left(k_{S_1}^A + k_{TR}^D\right)(k_{TR}^A + k_{TR}^D) + k_{ex}^D\left(k_{S_1}^A k_{TR}^A\left(k_{ISC}^A + k_{ISC}^D + k_{TR}^A\right) + k_{TR}^D\left(k_{ISC}^A + k_{TR}^A\right)\left(k_{ISC}^A + k_{TR}^A + k_{S_1}^D + k_{TR}^D\right)\right)\right)}{k_{S_1}^A k_{TR}^A\left(\left(\left(k_{S_1}^A + k_{S_1}^D\right)\left(k_{S_1}^A + k_{TR}^D\right) + k_{ex}^D\left(k_{S_1}^A + k_{ISC}^D + k_{TR}^D\right)\right)\left(\left(k_{S_1}^D + k_{TR}^A\right)(k_{TR}^A + k_{TR}^D) + k_{ex}^D\left(k_{ISC}^D + k_{TR}^A + k_{TR}^D\right)\right)\right)}}$$

with the Förster formula, $\left(k_{ET}^D = k_{S_1}^D \left(\frac{R_0}{R}\right)^6\right)$, $E_{9st}$ becomes

$$E_{9st} \equiv \cfrac{1}{1 + \cfrac{k_{S_1}^D}{k_{S_1}^D\left(\frac{R_0}{R}\right)^6} + \cfrac{k_{ex}^D k_{S_1}^D \left(\left(k_{S_1}^A k_{ISC}^A + (k_{S_1}^D + k_{TR}^A)(k_{ISC}^A + k_{TR}^A)\right)(k_{S_1}^A + k_{TR}^D)(k_{TR}^A + k_{TR}^D) + k_{ex}^D\left(k_{S_1}^A k_{TR}^A(k_{ISC}^A + k_{ISC}^D + k_{TR}^A) + k_{TR}^D(k_{ISC}^A + k_{TR}^A)(k_{ISC}^A + k_{TR}^A + k_{S_1}^D + k_{TR}^D)\right)\right)}{k_{S_1}^A k_{TR}^A\left(\left((k_{S_1}^A + k_{S_1}^D)(k_{S_1}^A + k_{TR}^D) + k_{ex}^D(k_{S_1}^A + k_{ISC}^D + k_{TR}^D)\right)\left((k_{S_1}^D + k_{TR}^A)(k_{TR}^A + k_{TR}^D) + k_{ex}^D(k_{ISC}^D + k_{TR}^A + k_{TR}^D)\right)\right)}}$$

and simplifies to

$$E_{9st} \equiv \cfrac{1}{k_{ex}^D k_{S_1}^D\left(\left(k_{S_1}^A k_{ISC}^A + \left(k_{S_1}^D + k_{TR}^A\right)\left(k_{ISC}^A + k_{TR}^A\right)\right)\right.}$$

(continued)

$$E_{9st} \equiv \cfrac{1}{1 + \left(\frac{R}{R_0}\right)^6 + \cfrac{\begin{array}{c} k_{ex}^D k_{S_1}^D \left(\left(k_{S_1}^A k_{ISC}^A + \left(k_{S_1}^D + k_{TR}^A\right)\left(k_{ISC}^A + k_{TR}^A\right)\right)\right. \\ \left(k_{S_1}^A + k_{TR}^D\right)(k_{TR}^A + k_{TR}^D) \\ + k_{ex}^D\left(k_{S_1}^A k_{TR}^A\left(k_{ISC}^A + k_{ISC}^D + k_{TR}^A\right) + k_{TR}^D \right. \\ \left. \left(k_{ISC}^A + k_{TR}^A\right)\left(k_{ISC}^A + k_{TR}^A + k_{S_1}^D + k_{TR}^D\right)\right)\right) \end{array}}{\begin{array}{c} k_{S_1}^A k_{TR}^A\left(\left(\left(k_{S_1}^A + k_{S_1}^D\right)\left(k_{S_1}^A + k_{TR}^D\right)\right.\right. \\ \left.+ k_{ex}^D\left(k_{S_1}^A + k_{ISC}^D + k_{TR}^D\right)\right) \\ \left.\left(\left(k_{S_1}^D + k_{TR}^A\right)(k_{TR}^A + k_{TR}^D) + k_{ex}^D\left(k_{ISC}^D + k_{TR}^A + k_{TR}^D\right)\right)\right) \end{array}}}$$

$$\neq \cfrac{1}{1 + \left(\frac{R}{R_0}\right)^6}$$

with

$$\zeta = \cfrac{\begin{array}{c} k_{ex}^D k_{S_1}^D \left(\left(k_{S_1}^A k_{ISC}^A + \left(k_{S_1}^D + k_{TR}^A\right)\left(k_{ISC}^A + k_{TR}^A\right)\right)\right. \\ \left(k_{S_1}^A + k_{TR}^D\right)(k_{TR}^A + k_{TR}^D) \\ + k_{ex}^D\left(k_{S_1}^A k_{TR}^A\left(k_{ISC}^A + k_{ISC}^D + k_{TR}^A\right) + k_{TR}^D\left(k_{ISC}^A + k_{TR}^A\right)\right. \\ \left.\left(k_{ISC}^D + k_{TR}^A + k_{S_1}^D + k_{TR}^D\right)\right)\right) \end{array}}{\begin{array}{c} k_{S_1}^A k_{TR}^A\left(\left(\left(k_{S_1}^A + k_{S_1}^D\right)\left(k_{S_1}^A + k_{TR}^D\right)\right.\right. \\ \left.+ k_{ex}^D\left(k_{S_1}^A + k_{ISC}^D + k_{TR}^D\right)\right) \\ \left.\left(\left(k_{S_1}^D + k_{TR}^A\right)(k_{TR}^A + k_{TR}^D) + k_{ex}^D\left(k_{ISC}^D + k_{TR}^A + k_{TR}^D\right)\right)\right) \end{array}}$$

## Protocols for $\zeta$-correction of FRET efficiency calculations

Mathematica notebooks, MATLAB code and sample data of the Cy3–Cy5 FRET pair are provided in Zenodo (Supplementary Additional Data 1 and 2 and Supplementary Codes 1 and 2; https://doi.org/10.5281/zenodo.10896601)[71] to enable reproduction of the results shown in Supplementary Fig. 14. All steps are described in summary here Use Mathematica v.13.1 for all our symbolic calculations. Use MATLAB 2023a or newer and make sure that SPARTAN (Single-molecule Platform for Automated Analysis) software[51] is in the MATLAB path.

Here, we provide the simplified $\zeta$-correction protocol for FRET efficiency calculations.

(1) Correct your data or the present Cy3–Cy5 data (Supplementary Additional Data 2; https://doi.org/10.5281/zenodo.10896601)[71] for crosstalk, direct excitation and γ using the γ value obtained at the lowest illumination intensity. See the manual for SPARTAN software[51] for specific commands. This will produce corrected traces files.

(2) To derive an analytical expression for $\zeta$, open the 'TestingSimplifiedZeta.nb' notebook (Supplementary Code 1; https://doi.org/10.5281/zenodo.10896601)[71] in Mathematica then run (press Shift + Enter after selecting individual cells) Part 1 and then Part 2.

(3) To derive an analytical expression for simplified $\zeta$ (without direct excitation), run Part 3 in the notebook.

(4) Review Part 4, editing relevant photophysical parameters for your FRET pair or leave the defaults that correspond to Cy3–Cy5. Run Part 4.

(5) Run Part 5a to calculate the simplified $\zeta$ values across experimental illumination intensities (kW cm$^{-2}$ and $k_{ex}$), editing the excitation rates to match your experimental condition.

(6) Run Part 5b to export the data after editing the output path as needed.

(7) Open the file 'zetacorrect.m' in MATLAB (Supplementary Code 2; https://doi.org/10.5281/zenodo.10896601)[71].
To correct the data for 100 mW (0.04 kW cm$^{-2}$), run zetacorrect.m in the Supplementary Additional Data 1 folder provided in Zenodo (https://doi.org/10.5281/zenodo.10896601)[71].

(8) Select the folder in the prompted folder that is the simplified $\zeta$ corrected file named 'Cy3Cy5_100 mW_ccorr_ade_gcorr_nzcor. traces' then it will plot Supplementary Fig. 14, second row's first panel at 100 mW (0.04 kW cm$^{-2}$).

### Preparation of donor-acceptor dye labeled DNA duplexes

A 21-nucleotide DNA, 5′-/5AmMC6/CAT GAC CAT GAC CAT GAC CAG/3BioTEG/-3′, containing a 5′ amino modifier with a six-carbon linker (5AmMC6) for fluorophore linkage and an additional 3′ biotin moiety attached via a 22-atom tetra-ethylene glycol spacer (3BioTEG) was purchased from Integrated DNA Technologies. A set of complementary DNA strands with an internal six-carbon linker-containing amino modifier (iAmMC6T) for fluorophore linkage was also purchased from Integrated DNA Technologies. Their sequences are listed below:

5′-CTG GTC ATG GTC ATG GTC A/iAmMC6T/G-3′
5′-CTG GTC ATG GTC ATG G/iAmMC6T/C ATG-3′
5′-CTG GTC ATG GTC A/iAmMC6T/G GTC ATG-3′
5′-CTG GTC ATG G/iAmMC6T/C ATG GTC ATG-3′
5′-CTG GTC A/iAmMC6T/G GTC ATG GTC ATG-3′

Then, 1 nmol of the 5′-amino modified DNA was individually labeled with tenfold excess of N-hydroxysuccinimide ester-activated donor fluorophore in a 10 µl final reaction containing 50 mM potassium borate (pH 8.1), 200 mM KCl and 10% dimethylsulfoxide. Similarly, 1 nmol of an internal amino modified complementary DNA strand was individually labeled with tenfold excess of N-hydroxysuccinimide ester-activated acceptor fluorophore in a 10 µl final reaction containing 50 mM potassium borate (pH 8.1), 200 mM KCl and 10% dimethylsulfoxide. After incubation at 23 °C for 1 h, both the reaction mixtures were subsequently quenched with 0.2 µl 1 M Tris-acetate (pH 7.5) at 23 °C for 2 min before mixing both the strands in an equimolar ratio and heated at 90 °C for 2 min followed by passive cooling to room temperature (23 °C). Then, 70 µl ddH$_2$O and 10 µl 3 M sodium acetate (pH 6) were then added followed by threefold excess of ethanol for precipitation of the duplexes overnight in a freezer. The resulting pellet from a 10-min spin at 14,000 rpm was resuspended in 1 ml buffer A (1.7 M ammonium

sulfate and 10 mM ammonium acetate, pH 5.8) and injected onto a phenyl 5PW column (FPLC, Äkta Purifier, GE Healthcare) and eluted over a 30-min gradient from buffer A to B (10% methanol and 10 mM ammonium acetate, pH 5.8). The desired peak of interest, showing absorption of both DNA and fluorophore, was collected, stored at −80 °C and used for smFRET imaging.

For single-dye-labeled DNA duplexes, only one DNA strand was labeled with a fluorophore and the complementary DNA strand was not labeled. All other protocols were followed as described above.

### Bulk fluorescence lifetime and fluorescence quantum yield measurements

Bulk fluorescence measurements were carried out in a FluoTime 300 steady-state and time-resolved spectrometer (PicoQuant). Fluorescence lifetimes of green and red dyes were measured using 532 nm (LDH-P-FA-530B) and 640 nm (LDH-D-C-640) pulsed diode lasers (PicoQuant), respectively. The instrument response function was collected using a scatterer (Ludox AS40 colloidal silica, Sigma-Aldrich). The TCSPC data from the fluorescence lifetime measurements were fitted into an exponential decay model in EasyTau software (PicoQuant). A value of $\chi^2$, in between 0.98 and 1.2, was considered as a good fit, which was further adjudged by symmetrical distribution of the residuals. Fluorescence lifetimes of green and red dyes were collected at emission wavelengths 570 and 670 nm, respectively.

Absolute fluorescence quantum yields were measured in a FluoTime 300 spectrometer using integrating sphere accessories (PicoQuant) and a 300 W xenon excitation lamp. Green and red dyes were photoexcited at 517 and 580 nm; and the detection ranges were 512–700 and 575–800 nm, respectively. Before the quantum yield measurements, absorbance of the dyes at the excitation wavelengths was adjusted to 0.02 using a Shimadzu UV-2600 spectrometer, to minimize re-absorption of emitted photons. All bulk measurements were carried out in T50 buffer (10 mM Tris HCl and 50 mM KCl, pH 7.4) at room temperature using standard 1-cm path length quartz cuvettes (Starna Cells). The fluorescence quantum yield and fluorescence lifetime values of the dyes are tabulated in Supplementary Table 1.

### Confocal single-molecule FRET measurements

Single-molecule fluorescence experiments were performed using a MicroTime 200 (PicoQuant) time-resolved confocal fluorescence microscope equipped with an inverted Olympus IX73 microscope.

DNA duplex molecules labeled with Cy3 and Cy5 or LD555 and LD655 were excited with a vertically polarized 531 nm diode laser (D-FA-530L, PicoQuant) operated in continuous wave mode. Emitted fluorescence was collected by the same microscope objective used to focus the laser beam into the sample solution (Olympus UPlanSApo ×60/1.20W), passed through a long pass filter (532 LP, Semrock) to remove the residual excitation light and was then focused onto a 100-µm pinhole before being split by a dichroic mirror (T635lpxr, Chroma). Additionally, fluorescence photons were filtered by bandpass filters (ET585/65M and H690/70, Chroma) on the donor and acceptor channels, respectively, before being focused onto two single-photon avalanche detectors (Excelitas SPCM-AQRH-TR). The arrival time of every detected photon was recorded with a HydraHarp 400M time-correlated single-photon-counting module (PicoQuant) and stored for each measurement.

FRET efficiency histograms of freely diffusing doubly labeled DNA duplex molecules were acquired on samples with concentrations between 50 and 100 pM, recorded at 10, 20, 50, 100 and 200 µW between 20 min and 3 h (full list of excitation power values in kW cm$^{-2}$ and s$^{-1}$ below). All measurements were performed in T50 buffer (10 mM Tris HCl, pH 7.4 and 50 mM KCl), in the presence of 1 mM 3,4-dihydroxybenzoic acid (PCA) and 50 nM protocatechuate 3,4-deoxygenase (PCD) (Sigma-Aldrich) as oxygen scavengers and 0.005% Tween 20 (Pierce) to minimize surface adhesion[12].

## Confocal single-molecule FRET data analysis

FRET efficiencies were obtained from $E = n_A/(n_A + n_D)$, in which $n_D$ and $n_A$ are the numbers of donor and acceptor photons, respectively, in each burst, corrected for background and acceptor direct excitation[72].

Bursts were identified from the measured photon traces following Eggeling et al.[73] and Hoffmann et al.[74] If $\Delta t_i = t_i - t_{i-1}$ is the inter-photon time of the $i^{th}$ photon, the photon is retained if $\Delta t_i \leq \Delta t_{max}$. To define an optimal $\Delta t_{max}$ for each power, inter-photon times were first computed for each measurement using a common $\Delta t_{max}$ of 150 µs; the inter-photon time distribution was then analyzed, and the power-specific $\Delta t_{max}$, reflecting the varying photon rate, was chosen to encompass the whole distribution in all cases. Final power-specific $\Delta t_{max}$ values varied between 60 µs and 20 µs for LD555–LD655 and between 70 µs and 30 µs for Cy3–Cy5, for lowest and highest powers, respectively, in both cases. To avoid a possible bias of the power-specific inter-photon time distributions due to the choice of the initial common $\Delta t_{max}$, FRET efficiency values obtained with 30% higher or lower power-specific $\Delta t_{max}$ were computed and reported in plots of power-dependent FRET efficiency in Supplementary Fig. 3. It is evident that different $\Delta t_{max}$ values slightly increase or decrease (but do not abolish) the dependence of FRET efficiency on excitation power.

When the selection algorithm detects the $n^{th}$ photon with inter-photon arrival time $\Delta t_{i+n} > \Delta t_{max}$, a burst is defined with total length $T = t_{n-1} - t_{i-1}$. The resulting string of photons is corrected for background and if the total number of photon $n_A + n_D$ exceeds the burst threshold (BT), the burst is kept[75]. The BT was set to a different value for each power to minimize inclusion of fluorescence background; variations of ±30% in the estimated optimal BT value resulted in a FRET efficiency change much smaller than that obtained when changing $\Delta t_{max}$. Finally, molecules whose acceptor bleached during the transit through the confocal spot were also filtered out[76] and the remaining photon bursts were used to construct the FRET efficiency histograms.

## Fluorescence correlation spectroscopy experiments

FCS experiments were performed using a MicroTime 200 time-resolved confocal fluorescence microscope as detailed in the previous paragraph. Cy3B and Cy5B samples were excited with continuous wave laser beams at 531 nm and 639 nm, respectively. Fluorescence emission was collected by the microscope objective (Olympus UplanApo ×60/1.20W), focused onto a 150-µm pinhole, separated into two channels with a polarizing beam splitter and after passing through a bandpass filter (H690/70, Chroma) was focused onto single-photon avalanche photodiode detectors (Excelitas SPCM-QRH-TR) for each polarization. The signal of the two detectors was then correlated between 1 ns and 1 s with logarithmically spaced time steps. The resulting fluorescence intensity auto-correlations, $G(\tau)$, were satisfactorily fitted with a model including photon anti-bunching (ab), triplet state blinking (TR) and translational diffusion (D) through a 3D Gaussian-shaped confocal volume:

$$G(\tau) = \frac{1}{N} \cdot \left(1 - A_{ab} \cdot e^{-\frac{\tau}{\tau_{ab}}}\right) \cdot \left(1 + A_{TR} \cdot e^{-\frac{\tau}{\tau_{TR}}}\right) \cdot \left(1 + \frac{\tau}{\tau_D}\right)^{-1} \cdot \left(1 + s^2 \frac{\tau}{\tau_D}\right)^{-\frac{1}{2}}$$

Where $\tau$ is the delay time, $N$ is the average number of molecules in the confocal volume, $\tau_D$ is the diffusion time in µs, $s$ is the ratio of the lateral to the axial radii of the confocal volume ($s = 0.2$), $A_{TR}$ and $\tau_{TR}$ are the amplitude and lifetime (in µs) of the triplet state population, $\tau_{ab}$ is the antibunching time in ns and $A_{ab}$ is its associated amplitude.

Because of the small size of the dyes, compounded with photobleaching, which shortens the apparent transit time of the dye through the confocal volume, the timescale for blinking and diffusion overlap to a considerable extent, complicating the estimation of an accurate triplet lifetime. The choice of a 150-µm pinhole and the low power of 10 µW (measured at the back aperture of the microscope) were chosen to extend the diffusion time and to reduce photobleaching,

thus minimizing the above-mentioned overlap. Additionally, to further reduce the uncertainty, we performed measurements also at 50 µW and obtained a more accurate triplet lifetime at zero power via linear extrapolation.

All FCS experiments were carried out with freely diffusing dyes at a concentration of ~1–5 nM in the same fluidic channels used for TIRF imaging experiments in deoxygenated conditions using 1 mM PCA and 50 nM PCD (Sigma-Aldrich) as oxygen scavengers. Triplet lifetimes reported in Supplementary Table 1 are the average of two independent experiments.

Laser power in all cases was measured at the back aperture of the microscope in µW and converted to irradiance (kW cm⁻²) assuming a 300 nm lateral radius of the confocal volume. The rate of fluorophore excitation ($k_{ex}$, s⁻¹) was calculated using the equations explained in the section 'Calculations of steady-state populations for donor-only molecules' above. For confocal experiments, both conversions are error prone because of the uncertainty in estimating the confocal volume size, which is assumed constant at different powers. Our estimate is that the illumination intensity and $k_{ex}$ values are accurate within a factor of two of the quoted values. Indeed, calculations of $k_{ex}^D$ via anti-bunching time and fluorescence lifetime from FCS experiments carried out at 10 µW and 50 µW were found to be within 1.7× of the quoted values from the naive conversion. Values of excitation powers were 10, 20, 50, 100 and 200 µW corresponding to an illumination intensity of 7, 14, 35, 71 and 141 kW cm⁻² and a $k_{ex}$ of $5 \times 10^6$, $10^7$, $2.5 \times 10^7$, $5 \times 10^7$ and $10^8$ s⁻¹.

FCS and confocal data were analyzed with Fretica, a custom WSTP software package (https://schuler.bioc.uzh.ch/programs) for Mathematica (Wolfram Research).

## smFRET imaging of DNA duplexes

smFRET imaging experiments were performed using a custom-built, prism-based TIRF microscope, as described previously[51]. Fluorophores linked to biotinylated DNA molecules were immobilized via biotin-streptavidin interactions in quartz microfluidic chambers. Fluorescence from the surface-immobilized dyes, illuminated by the evanescent wave generated by total internal reflection of laser light, was collected using a 1.27 numerical aperture, ×60 water-immersion objective (Nikon) and imaged onto scientific complementary metal-oxide semiconductor cameras (Hamamatsu ORCA-Flash 4.0 v.2) having 2,048 × 2,048 pixels with 6.5-µm pixel size and 2 × 2 binning, connected to a PC with Camera Link acquisition boards.

Experiments were performed in T50 buffer at 25 °C. The deoxygenated imaging buffers were made by using 1 mM PCA and 50 nM PCD as oxygen scavengers in T50 buffer. Whenever the impact of any additives on FRET efficiency was tested, the additives were added to the buffers and their concentrations are mentioned in the figures. The FRET experiments were performed by selective excitation of the donor fluorophore using a 532 nm laser (Opus, Laser Quantum) at 100 ms time resolution. All videos were recorded using custom software implemented in LabView (National Instruments).

## smFRET imaging data analysis

Analysis of wide-field TIRF videos was performed using SPARTAN software[51] v.3.7 implemented in MATLAB. Single molecules were detected within wide-field TIRF videos by finding peaks of fluorescence signal at least 8 × s.d. above background noise. Overlapping peaks (closer than 3.5 pixels) were automatically removed. Single-molecule traces were extracted by summing the nine most-intense pixels within the 5 × 5-pixel neighborhood around each peak of intensity, applying a scaling factor provided by the camera vendor for converting from the camera's arbitrary units to photon counts (0.49 photoelectrons per analog-to-digital units). Further, we applied a set of selection criteria: FRET above baseline >15 frames, signal-to-background noise ratio >8, number of donor-blinking of events <4 and background noise <70. FRET efficiency traces were idealized using the segmental K-means

algorithm[77]. We use the empirical $\gamma$-correction in Supplementary Figs. 5–7, 9 and 14a for each laser power.

## Data availability

Raw data for Fig. 2c, illustrating the illumination-intensity-dependent changes in FRET efficiency for the Cy3–Cy5 pair, is available for download on Zenodo (https://doi.org/10.5281/zenodo.10896601)[71]. The source data files for main text figures and supplementary figures are also available on Zenodo (https://doi.org/10.5281/zenodo.10896601)[71]. Source data are provided with this paper.

## Code availability

We used published or commercial software to collect and analyze the imaging and photon-counting data as stated in Methods. The Mathematica notebook and MATLAB code used to generate Supplementary Fig. 14 are available for download on Zenodo (https://doi.org/10.5281/zenodo.10896601)[71].

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

## Acknowledgements

This work was supported by the National Institutes of Health (R01GM098859 to S.C.B.). We thank St. Jude Children's Research Hospital for their generous support of the Single-Molecule Imaging Center. We thank M. J. Schnermann at the National Cancer Institute for providing the Cy5B dye for triplet state lifetime measurements. We also thank all Blanchard laboratory members for helpful feedback and discussions.

## Author contributions

S.C.B., A.K.P. and Z.K. conceived the project. A.K.P., M.I.M. and D.S.T. performed TIRF single-molecule fluorescence imaging experiments. M.I.M. and D.S.T. contributed equally to this work. A.B. executed, analyzed and interpreted confocal smFRET and FCS experiments. Z.K. developed the simulation methods and the analytical approaches presented. A.K.P., R.K., R.B.A. and M.I.M. prepared and purified fluorophore-labeled oligonucleotides. A.K.P., M.I.M. and S.J. performed and interpreted ensemble photophysical measurements. S.B. synthesized fluorophores. S.C.B., A.K.P., Z.K., M.I.M. and D.S.T. analyzed and interpreted single-molecule fluorescence imaging data. S.C.B. supervised the project. The manuscript was written through the contributions of all authors.

## Competing interests

S.C.B. and R.B.A. have an equity interest in Lumidyne Technologies. The remaining authors declare no competing interests.

## Additional information

**Correspondence and requests for materials** should be addressed to Scott C. Blanchard.

# Reporting Summary

## Statistics

For all statistical analyses, confirm that the following items are present in the figure legend, table legend, main text, or Methods section.

| n/a | Confirmed | |
|---|---|---|
| ☐ | ☒ | The exact sample size (*n*) for each experimental group/condition, given as a discrete number and unit of measurement |
| ☐ | ☒ | A statement on whether measurements were taken from distinct samples or whether the same sample was measured repeatedly |
| ☒ | ☐ | The statistical test(s) used AND whether they are one- or two-sided<br>*Only common tests should be described solely by name; describe more complex techniques in the Methods section.* |
| ☒ | ☐ | A description of all covariates tested |
| ☒ | ☐ | A description of any assumptions or corrections, such as tests of normality and adjustment for multiple comparisons |
| ☐ | ☒ | A full description of the statistical parameters including central tendency (e.g. means) or other basic estimates (e.g. regression coefficient) AND variation (e.g. standard deviation) or associated estimates of uncertainty (e.g. confidence intervals) |
| ☒ | ☐ | For null hypothesis testing, the test statistic (e.g. *F*, *t*, *r*) with confidence intervals, effect sizes, degrees of freedom and *P* value noted<br>*Give P values as exact values whenever suitable.* |
| ☒ | ☐ | For Bayesian analysis, information on the choice of priors and Markov chain Monte Carlo settings |
| ☒ | ☐ | For hierarchical and complex designs, identification of the appropriate level for tests and full reporting of outcomes |
| ☒ | ☐ | Estimates of effect sizes (e.g. Cohen's *d*, Pearson's *r*), indicating how they were calculated |

*Our web collection on statistics for biologists contains articles on many of the points above.*

## Software and code

Policy information about availability of computer code

| Data collection | We used published or commercial software to collect the imaging and photon-counting data as stated in Online Methods. |
|---|---|
| Data analysis | We used published or commercial software to analyze the imaging and photon-counting data as stated in Online Methods. The Mathematica notebook and MATLAB code used to generate Supplementary Fig. 14 are available for download on Zenodo (https://doi.org/10.5281/zenodo.10896601). |

For manuscripts utilizing custom algorithms or software that are central to the research but not yet described in published literature, software must be made available to editors and reviewers. We strongly encourage code deposition in a community repository (e.g. GitHub). See the Nature Portfolio guidelines for submitting code & software for further information.

## Data

Policy information about availability of data

All manuscripts must include a data availability statement. This statement should provide the following information, where applicable:
- Accession codes, unique identifiers, or web links for publicly available datasets
- A description of any restrictions on data availability
- For clinical datasets or third party data, please ensure that the statement adheres to our policy

Raw data for Fig. 2c, illustrating the illumination-intensity dependent changes in FRET efficiency for the Cy3-Cy5 pair, is available for download on Zenodo (https://

## Human research participants

Policy information about studies involving human research participants and Sex and Gender in Research.

| | |
|---|---|
| Reporting on sex and gender | Not applicable |
| Population characteristics | Not applicable |
| Recruitment | Not applicable |
| Ethics oversight | Not applicable |

Note that full information on the approval of the study protocol must also be provided in the manuscript.

# Field-specific reporting

Please select the one below that is the best fit for your research. If you are not sure, read the appropriate sections before making your selection.

☒ Life sciences          ☐ Behavioural & social sciences          ☐ Ecological, evolutionary & environmental sciences

For a reference copy of the document with all sections, see nature.com/documents/nr-reporting-summary-flat.pdf

# Life sciences study design

All studies must disclose on these points even when the disclosure is negative.

| | |
|---|---|
| Sample size | Imaging experiments were carried out on replicate samples as stated in the Online Methods and figure legends |
| Data exclusions | No data were excluded from the analyses |
| Replication | The number of experimental replicates are stated in figure legend and all attempts at replication were successful |
| Randomization | Imaging experiments were quantitative and randomization was not relevant to the study |
| Blinding | Imaging experiments were quantitative and blinding was not relevant to the study |

# Reporting for specific materials, systems and methods

We require information from authors about some types of materials, experimental systems and methods used in many studies. Here, indicate whether each material, system or method listed is relevant to your study. If you are not sure if a list item applies to your research, read the appropriate section before selecting a response.

## Materials & experimental systems

| n/a | Involved in the study |
|---|---|
| ☒ ☐ | Antibodies |
| ☒ ☐ | Eukaryotic cell lines |
| ☒ ☐ | Palaeontology and archaeology |
| ☒ ☐ | Animals and other organisms |
| ☒ ☐ | Clinical data |
| ☒ ☐ | Dual use research of concern |

## Methods

| n/a | Involved in the study |
|---|---|
| ☒ ☐ | ChIP-seq |
| ☒ ☐ | Flow cytometry |
| ☒ ☐ | MRI-based neuroimaging |

