## [Peer Review File · Nature Methods]

Peer Review Information

Manuscript Title: Recovering true FRET efficiencies from smFRET investigations requires triplet state mitigation

Corresponding author name(s): Scott Blanchard

Editorial Notes: None

Reviewer Comments & Decisions:

Decision Letter, initial version:

Dear Scott,

Your Article, "Recovering true FRET efficiencies in smFRET investigations requires triplet state mitigation", has now been seen by three reviewers. As you will see from their comments below, although the reviewers find your work of considerable potential interest, they have raised a number of concerns. We are interested in the possibility of publishing your paper in Nature Methods, but would like to consider your response to these concerns before we reach a final decision on publication.

We therefore invite you to revise your manuscript to address these concerns. We think the experimental concerns are fair and should be readily addressed (please let me know if you disagree). Our biggest concern is that referee 3 thinks the correction factor could be a "game changer" while referee 2 is skeptical that the zeta correction factor is modeled appropriately or meaningful. We hope to see a strong response to these concerns from referee 2 upon resubmission.

- * include a point-by-point response to the reviewers and to any editorial suggestions
- * please underline/highlight any additions to the text or areas with other significant changes to facilitate review of the revised manuscript
- * address the points listed described below to conform to our open science requirements

* ensure it complies with our general format requirements as set out in our guide to authors at www.nature.com/naturemethods

* resubmit all the necessary files electronically by using the link below to access your home page

[Redacted]

We hope to receive your revised paper within three months. If you cannot send it within this time, please let us know. In this event, we will still be happy to reconsider your paper at a later date so long as nothing similar has been accepted for publication at Nature Methods or published elsewhere.

OPEN SCIENCE REQUIREMENTS

REPORTING SUMMARY AND EDITORIAL POLICY CHECKLISTS

DATA AVAILABILITY

We strongly encourage you to deposit all new data associated with the paper in a persistent repository where they can be freely and enduringly accessed. We recommend submitting the data to discipline-specific and community-recognized repositories; a list of repositories is provided here:

<http://www.nature.com/sdata/policies/repositories>

All novel DNA and RNA sequencing data, protein sequences, genetic polymorphisms, linked genotype

and phenotype data, gene expression data, macromolecular structures, and proteomics data must be deposited in a publicly accessible database, and accession codes and associated hyperlinks must be provided in the "Data Availability" section.

CODE AVAILABILITY

Please include a "Code Availability" subsection in the Online Methods which details how your custom code is made available. Only in rare cases (where code is not central to the main conclusions of the paper) is the statement "available upon request" allowed (and reasons should be specified).

For more information on our code sharing policy and requirements, please see: <https://www.nature.com/nature-research/editorial-policies/reporting-standards#availability-of-computer-code>

MATERIALS AVAILABILITY

Authors reporting new chemical compounds must provide chemical structure, synthesis and

characterization details. Authors reporting mutant strains and cell lines are strongly encouraged to use established public repositories.

SUPPLEMENTARY PROTOCOL

To help facilitate reproducibility and uptake of your method (where you use the TSQ dyes to bypass intensity issues), we ask you to prepare a step-by-step Supplementary Protocol for the method described in this paper. We encourage authors to share their step-by-step experimental protocols on a protocol sharing platform of their choice and report the protocol DOI in the reference list. Nature Portfolio 's Protocol Exchange is a free-to-use and open resource for protocols; protocols deposited in Protocol Exchange are citable and can be linked from the published article. More details can found at www.nature.com/protocolexchange/about.

ORCID

Nature Methods is committed to improving transparency in authorship. As part of our efforts in this direction, we are now requesting that all authors identified as 'corresponding author' on published papers create and link their Open Researcher and Contributor Identifier (ORCID) with their account on the Manuscript Tracking System (MTS), prior to acceptance. This applies to primary research papers only. ORCID helps the scientific community achieve unambiguous attribution of all scholarly contributions. You can create and link your ORCID from the home page of the MTS by clicking on 'Modify my Springer Nature account'. For more information please visit please visit www.springernature.com/orcid.

Sincerely,
Rita

Rita Strack, Ph.D.
Senior Editor
Nature Methods

Reviewers' Comments:

Reviewer #1:
Remarks to the Author:

The manuscript by Pati et al. is a carefully conducted examination the power dependence of FRET efficiency in single molecule experiments where high laser power becomes necessary to obtain

increasing time resolution. The work clearly establishes the importance of accounting for triplet state occupancy of the fluorophores to obtain accurate measurements of absolute FRET efficiency. The authors show that this is particularly true of the acceptor triplet state, which can lead to deviations in FRET efficiency at high power.

This is of great significance to those in single molecule fluorescence and the growing field of dynamic structural biology, which require high power to obtain accurate measurements of intramolecular distances using FRET. That such a necessary correction has gone unnoticed for so long is surprising but Pati et al. have clarified the importance of this phenomena and describe methods for mitigation. As such, the work is a great fit for Nature Methods.

The manuscript is exceptionally well written and clear. The data are of high quality and the analysis is very rigorous with appropriate use of statistics throughout. The conclusions are well supported by the data. The study is well designed to illustrate the problem and provide mitigation strategies. The authors show the effect of different solution triplet state quenchers (TSQs), which all fall short of complete correction at the highest illumination power levels. The work highlights the importance of intramolecular triplet state quenchers as the only way to avoid this issue. However, the authors also provide a correction factor, which can recover the true FRET efficiency even when triplet state occupancy can't be prevented.

The manuscript is suitable for publication as is. I would suggest that it could be improved by adding a more standard TSQ cocktail in Figure 6, which is a highlight of the paper. This is a very well designed experiment that should become a standard calibration. However, the use of MV/AA as the TSQ is not optimal and not widely used in single molecule imaging. MV/AA also performs poorly in the author's own tests of TSQs (Fig S4 & S5). I get that the authors want maximum impact from their intramolecular TSQs but a more fair comparison would be to the COT/NBA/Trolox mix, which is much more effective at suppressing triplet state and preserving FRET. I would be curious to see how this mix performed in this experiment as these are the more standard TSQs in smFRET.

The manuscript also makes the contention that the power dependent effects are likely most relevant for confocal single molecule experiments, which are always performed at high power. However, the manuscript contains no experiments that are directly relevant to this contention. Is it possible to substantiate this claim with experiments. Otherwise, it is a somewhat inflammatory statement to make without support.

Reviewer #2:

Remarks to the Author:

Pati et al. investigate to what extent standard correction methods for FRET efficiency fail at elevated illumination intensities. They demonstrate this experimentally using TIRF for a set of three FRET dye-pairs commonly used in single molecule microscopy measured under many different conditions. They find that triplet-state depopulation, either achieved by adding triplet state quenchers to the imaging buffer, or by using “self-healing” fluorophores, mitigates the problematic illumination-intensity dependence of FRET data.

The possibility of using elevated illumination-intensities is highly desirable for many applications in smFRET where improved time resolution is needed. The plethora of data presented here is impressive and will surely help investigators to choose optimal dye-pairs and conditions for studies where time resolution is key. Especially, the authors demonstrate that the “self-healing” dye-pair LD555-LD655 is much better suited than its closely related pair Cy3-Cy5 which is lacking “self-healing” capacities.

Furthermore, the authors suggest a correction factor, zeta, that is meant to correct for any residual power dependence of the measured transfer efficiencies. However, for practical and for theoretical reasons I find the proposed correction procedure not convincing. The authors suggest that zeta should be calculated from a detailed photophysical model that describes the FRET process itself and in addition all mechanisms that lead to the power dependence of the measurements. Practically, however it might be quite difficult to pick the right model and to know or determine all model parameters for calculating zeta. Especially, if they depend strongly on the experimental details, as for example on the presence of triplet quenchers in the local environment of the fluorophores. On theoretical grounds, I am criticizing the photophysical model proposed by the authors, because it is neglecting quite relevant photophysical effects, namely triplet-singlet and singlet-singlet excitation annihilation (STA and SSA) as well as reverse intersystem-crossing (RISC) (see for example: van Grondelle, R. *Biochim. Biophys. Acta* 1985, 811, 147; or Hubner et al. *Phys. Rev. Lett.* 91, 093903;2003). These processes are themselves inter-dye distance depending. Including these effects into the proposed correction would hence make zeta itself distance dependent and therefore most likely useless for correction.

The proposed correction ζ zeta is not novel. It is related to the parameter Λ , which was introduced by Camley et al., simply by $\zeta = \Lambda - 1$. (Compare Eq. 3 of the manuscript to Eq.4 of Camley et al.) The introduction and discussion of Λ is the main content and result of the paper by Camley et al. The inter-dye distance dependence of Λ due to STA, SSA and RISC was later pointed out by Nettels et al. The authors of the manuscript cite Nettels et al., but surprisingly, they do not address the results of that publication.

In addition to this, I need to say that the parts of the manuscript that concern theory are not well written. Especially, the SI, that contains almost all important details on the theory behind zeta, needs a thorough proofreading and clarification of nomenclature and definitions.

In the following I present in more detail my criticism and comments:

- The authors use a nomenclature for transfer efficiencies that is confusing, if not inconsistent. Surprisingly, I find no clear definition of what the authors mean with “true transfer efficiency” in the main text, although this is a key term used throughout the manuscript. On page 12, the letter ε is introduced for denoting this quantity, “(ε ; true FRET efficiency)”. And in Eq. 2 it is used to define the correction parameter ζ . However, in my understanding, which is backed by what I read in the SI, ε is meant to be the measured (apparent) transfer efficiency, which is corrected for gamma, crosstalk, and acceptor direct excitation according to the standard procedures found

in literature. The measured ε approximates well, i.e., with high accuracy, the transfer efficiency defined by Forster theory in the limit of low excitation intensity. (Additional low shot-noise and/or averaging over many events (bins) is needed to get precise values.)

The definition of transfer efficiency according to theory can be found on page 23 of the SI and is denoted there as E_{true} . (Including k_{isc} in the denominator is correct, although for all practical uses negligible.) For this quantity Forster showed that $E_{true} = 1/(1 + (\frac{R}{R_0})^6)$. According to Eq.1 of the main text, this is equal to what the authors call E_{std} . The definition of the transfer efficiency is independent of k_{ex} . Hence, the limit " $k_{ex} \ll k_{em}^D, k_{em}^A$ " for the validity of Eq.1 is needed only in the sense that stimulated emission is neglected in the theory. But this is not the point the authors want to make here. Presumably, they want to state that the measured value ε depends on k_{ex} and approximates E_{true} only well in the limit $k_{ex} \rightarrow 0$. Alternatively, one could read Eq.1 not as a defining equation but such that E_{std} is defined as the measured and with the standard method corrected value of transfer efficiency, but then E_{std} would equal ε according to its definition in the SI. With this Eq.2. would become pointless. The authors need to clarify. (On page 30 of the SI yet another symbol is introduced, $\varepsilon^{corrected}$, which algebraically equal to E_{std} in Eq.2 of the main text.) Notation needs to be harmonized throughout both texts. Beside this I suggest to replace "true" by "accurate" throughout the text, whenever appropriate, for example in the title of the manuscript.

- The correction factor zeta introduced in the manuscript is obviously only useful if it is not itself dependent on the inter-dye distance R .

The authors show in the SI for the photophysical model depicted in Fig.4 that this is indeed the case. (To be accurate, they show it for the case that direct excitation of the acceptor is negligible.)

However, this model, which is identical to the one presented in Camley et al. (Ref. 6 in the SI), does not include well-known photophysical effects that are indeed R -dependent and might be relevant. The model assumes for example that no energy is transferred from the donor S_1 state to the acceptor dye, while the latter is in triplet state T_1 . In fact, for Cy5 it has been shown (Huang et al. *J. Phys. Chem. A* 2006, 110, 1, 45–50) that the absorbance spectrum of $T_1 \rightarrow T_n$ is similar to the absorption spectrum of $S_0 \rightarrow S_n$. As a consequence, energy transfer is expected, lifting the T_1 state of the acceptor to a higher triplet state T_n . Subsequently, this can lead to singlet-triplet annihilation (STA) or reverse inter-system crossing (RISC). In the first case, in contrast to the model prediction of Fig.4a, no donor photon is emitted, in the second case, the acceptor is in S_1 , and can immediately emit a photon. These and yet another effect, namely singlet-singlet annihilation (SSA), are inter-dye distance dependent and lead to lower deviations of the apparent transfer efficiency from the true one than predicted by the model of Fig.4a. This was previously investigated by Nettels et al. Their paper is cited in the manuscript, but possible consequences, most importantly, a possible distance dependence of zeta due to STA, RISC, and SSA processes are not discussed. (Also, a discussion of the possible role of photo-induced cis/trans isomeration of Cy5 would be desirable in this context.)

- If or if not, zeta is distance dependent for the dye-pairs and under the conditions studied here could be easily investigated by experiment. I suggest that the authors measure constructs with different labeling positions at some low-power and some high-power intensities and check if the reciprocal distances of measured transfer efficiency ($\frac{1}{\varepsilon_{highpower}} - \frac{1}{\varepsilon_{lowpower}}$) are the same for all inter-dye distances. The constructs represented in Fig. 6a would be ideal for this purpose.

- The authors might argue that their model is well justified simply by its success to correct the transfer efficiencies. However, some of the parameters (k_{TR} and k_{ISC}) were “optimized” in order to match measured and simulated efficiencies. This was done only for one inter-dye-distance R . In view of this, a model-free check like the one described above seems to be important to me.
- The authors suggest to calculate zeta based on an appropriate photo-physical model. For this essentially all model parameters (hopefully with the exception of R) need to be known. The problem with this is, from my point of view, that the photo physical mechanisms that kick in at elevated excitation powers are often not known to such an extent that a reliable quantitative calculation are feasible. Also, the fact, that kinetic parameters depend on environmental details and experimental conditions make an accurate calculation of zeta very challenging.
- Why is direct acceptor excitation neglected for the calculation of zeta in the SI? Acceptor photon emitted while the donor is in the triplet might lead to an apparent higher transfer efficiency. This might be relevant especially for the distance regime beyond R_0 .
- Joined states are represented in Fig.4a (and in corresponding figures of the SI) as sums, but in the text as products. As joined quantum states, the latter notation makes more sense to me.
- The vertical dimension of Jablonski diagrams usually represents energetic relations between the energy levels. For Fig. 4a this seems to be not the case.
- In Fig. 4a and in the corresponding figure in the SI are transitions missing which are part of the model described by Camley et al. and which are also described by the rate equations on page 28 of the SI. Why are these transitions omitted in the diagrams?
- The wavy arrows in the diagrams and their labels $k_{em}^{A,D}$ suggest that these transitions are purely radiative (quantum yields equal to one). Is this intended?
- In Eq. 3 the authors limit the validity to k_{ex} larger or equal the emission rate coefficients. This limit is never reached. At such elevated excitation rates spontaneous emission would be needed to be taken into account. On the other side, Eq. 3 is valid for very low k_{ex} values, because zeta becomes zero in this limit.
- The triplet and diffusion components in the FCS curves are hardly distinguishable. I expect the error on k_{TR} to be substantial. The authors should report the results for all fitting parameters with errors.
- Did the authors try to measure FCS at higher laser powers, where the components might be better distinguishable?
- I assume the reported 10 mW measured at the back aperture of the objective for FCS are meant to be 10 microwatts? What values does one get for k_{ex} ?
- The amplitude of the triplet component is $c_T \approx \frac{k_{ex}k_{ISC}}{k_{fl}k_{TR}}$; the expected value could be calculated from the parameters given in SI Table 1. FCS measurements at varied laser powers could help to obtain a robust and independent value for k_{ISC} .
- The first plus sign in the formula for fitting the FCS data is there by mistake. In SI Fig.10, the x-axis label should be “tau/ns” instead of “log(tau/ns).
- The authors determine the excitation rate coefficient simply from the measured light intensity of the laser beam and the extinction coefficient of the fluorophores. However, both in TIRF (evanescent field) and as well in confocal microscopy (tight laser focus) the excitation rate varies drastically across the sample volume; for TIRF especially in z-direction. How does this effect the results? Would it be better to use a spatial averaged value for k_{ex} ?

- SI on page 23, “In an attempt to keep dyes’ quantum yields unchanged, we incorporate the information that $k_{flA,*} = k_{flA} - k_{ISC A}$ (also denoted as $k_{em A}$) and $k_{flD,*} = k_{flD} - k_{ISC D}$ (similarly denoted as $k_{em D}$).”

This sentence is unclear to me. What should change the quantum yield in a theoretical calculation/simulation? What kind of information is meant. The symbols $k_{fl}^{D,A}$ were not introduced before. Why are two symbols ($k_{fl}^{D,A,*}$ and $k_{em}^{D,A}$) introduced for the same thing? Again, are these meant to be purely radiative rates? If yes, why is not the general case treated? Can it generally be excluded that zeta depends on differences in quantum yields?

- “In our simulations we consider 20% detection efficiency, ...”
The same for both colors?
- “Otherwise, the effect of the direct excitation of the acceptor fluorophore would be on the denominator of the E_{true} . However, it is known to be a very small contribution to the true FRET efficiency.”
These sentences are unclear to me. According to the equation right above this statement, E_{true} is just representing the definition of FRET efficiency according to theory for this model. How could this possibly depend on the acceptor direct excitation? Again, I do not really understand what the authors mean with “true” FRET efficiency. For my taste it would be preferable not to use this problematic term. Instead, one needs to distinguish between the theoretically defined quantity and the measured value. The latter can be more or less accurate, but not “true” in my opinion. Correction factors correct for known systematic errors. If systematic errors remain uncorrected for, one can speak about apparent transfer efficiencies.
- “We carry out the correction on the raw FRET efficiency as more complexity is getting introduced while the FRET efficiency is corrected such as quantum yield difference as well as detection efficiencies in both donor and acceptor channels.”
How is “raw FRET efficiency” defined? Please check the grammar and semantics.
- “We follow the same photophysical trajectory generation steps for both donor and acceptor fluorophores in probing high spatial resolution performance of smFRET imaging.”
In what sense do the simulation represent spatial resolution of the imaging?
- A careful proofreading of the entire SI is needed.
- The used Förster radii for the dye pairs should be reported.
- The vertical dashed lines in Fig. 6(b-d) make sense for comparing experimental (c) and theoretical (d) results for the LD555-LD655 dye pair. However, for Cy3-Cy5 (b) another Förster radius applies; a direct comparison is hence not possible in FRET efficiency space.
- “We initiated these investigations by calibrating to experimental and simulated values for Cy3B fluorophore brightness across a range of excitation rates commensurate with those used to interrogate Cy3”
Please clarify why Cy3B instead of Cy3 was used here. Given the high popularity of C3B and Cy5B, it is surprising that they were here not further investigated as a FRET pair. A direct comparison with Cy3-Cy5 would be very interesting.

Reviewer #3:
Remarks to the Author:
Dear

With much interest I have reviewed this manuscript on the demonstration of I_{exc} -dependent changes of E , and the remediation thereof via either a correction parameter or intramolecular T1-quenching strategies.

The work first is a thorough comparative analysis of normal and selfhealing dyes in TIRF/widefield single-molecule (FRET) analyses, and their dependence on TSQs and I_{exc} . The experiments are complemented with kinetic simulations that support the proposed triplet-mediated effects on E . Authors also derive a correction parameter for the triplet-affected data.

I have no comments whatsoever on the quality of the experiments and simulations. Experiments have been performed and analyzed excellently, rendering data interpretation straightforward and leaving little room for biased interpretation. I applaud the authors for their thorough work.

The ζ correction, in my opinion, is the main novelty of the work, and it can be a game changer.

I have a few suggestions to improve the manuscript.

Intro:

- Fig. 1A is clear, but somewhat obsolete as it is really just textbook info
- Fig. 1B needlessly occupies 2x space as needed, as D and A can undergo, in principle and as illustrated, identical processes. Also, explicitly indicate where the photobleaching pathways are.

"The addition of high concentrations of TSQs in solution can, however, exhibit drawbacks, and irregularities linked to their varied and limited aqueous solubilities, excited-state reactivities, and potential toxicities to the systems under investigation, particularly when investigating living cells^{36,48,49}."

- Ref 48 does not support this statement directly, and ref 36 is a review. Cite original literature proving the statement of the authors. Same remark for the next sentence, where ref 36 is cited again.

"Here, we demonstrate that the experimentally determined FRET efficiencies of a given dye pair vary as a function of illumination intensity..."

- A comparative analysis of different acceptor dyes regarding I_{exc} -dependent FRET changes was reported by Vandenberg et al 2018 J Phys Chem B., albeit without providing corrections for it. At least this publication could be cited.

Results

- The authors have limited themselves experimentally to TIRF/widefield, while it is of interest to do a comparative experimental analysis also for the MFD/confocal counterpart. Considering the vastly different (much more brief excitation per molecule, and highly inhomogeneously) excitation geometry I do not agree the performance can simply be deducted from the experiments/simulations that have been done. Especially statements such as "Solution-based methods to suppress triplet states are

therefore expected to fall short of meeting the field's goal of achieving accurate FRET efficiency estimations" are certainly valid for the current work, but unclear to what extent they are applicable (whether more or less) to MFD/confocal.

- Self-healing dyes may be spectroscopically beneficial, are they biologically equally compatible? This merits a discussion, at least.

- That acceptor blinking critically affects smFRET experiments has long been known. How does the D-A distance affect the A-caused problems as described in this paper?

Figure 2 and others: I would suggest to add a graph of the (center of mass? Median? Gaussian fitted? PDA fitted?) E values (potentially their % deviation from the value at the lowest I_{exc} as a secondary axis) to allow the reader to more easily view the deviations.

Author Rebuttal to Initial comments

Point-by-Point Responses to the Reviewers [NMETH-A52148]**Reviewer #1**

The manuscript by Pati et al. is a carefully conducted examination the power dependence of FRET efficiency in single molecule experiments where high laser power becomes necessary to obtain increasing time resolution. The work clearly establishes the importance of accounting for triplet state occupancy of the fluorophores to obtain accurate measurements of absolute FRET efficiency. The authors show that this is particularly true of the acceptor triplet state, which can lead to deviations in FRET efficiency at high power. This is of great significance to those in single molecule fluorescence and the growing field of dynamic structural biology, which require high power to obtain accurate measurements of intramolecular distances using FRET. That such a necessary correction has gone unnoticed for so long is surprising but Pati et al. have clarified the importance of this phenomena and describe methods for mitigation. As such, the work is a great fit for Nature Methods. The manuscript is exceptionally well written and clear. The data are of high quality and the analysis is very rigorous with appropriate use of statistics throughout. The conclusions are well supported by the data. The study is well designed to illustrate the problem and provide mitigation strategies. The authors show the effect of different solution triplet state quenchers (TSQs), which all fall short of complete correction at the highest illumination power levels. The work highlights the importance of intramolecular triplet state quenchers as the only way to avoid this issue. However, the authors also provide a correction factor, which can recover the true FRET efficiency even when triplet state occupancy can't be prevented. The manuscript is suitable for publication as is.

RESPONSE 1: We are grateful to the Referee for their positive feedback on the work and its recommendation for publication.

I would suggest that it could be improved by adding a more standard TSQ cocktail in Figure 6, which is a highlight of the paper. This is a very well-designed experiment that should become a standard calibration. However, the use of MV/AA as the TSQ is not optimal and not widely used in single molecule imaging. MV/AA also performs poorly in the author's own tests of TSQs (Fig S4 & S5). I get that the authors want maximum impact from their intramolecular TSQs but a more fair comparison would be to the COT/NBA/Trolox mix, which is much more effective at suppressing triplet state and preserving FRET. I would be curious to see how this mix performed in this experiment as these are the more standard TSQs in smFRET.

RESPONSE 2: We appreciate this suggestion and now include the requested experiment in the presence of the more standard cocktail of 1 mM (each) of COT, NBA, and Trolox [Fig. 5c, revised manuscript]. We wish to maintain inclusion of MV/AA as this mixture represents the popular ROXS system (Vogelsang et al., PNAS, 2005, 106, 8107-8112; Vogelsang et al., Angew. Chem. Int. Ed, 2008, 47, 5465-5469).

The manuscript also makes the contention that the power dependent effects are likely most relevant for confocal single molecule experiments, which are always performed at high power. However, the manuscript contains no experiments that are directly relevant to this contention. Is it possible to substantiate this claim with experiments. Otherwise, it is a somewhat inflammatory statement to make without support.

RESPONSE 3: Our intention with the inclusion of illumination intensities extending beyond those typically employed using wide-field imaging platforms (excitation rates $>10^6 \text{ s}^{-1}$) was only to demonstrate how a 9-state framework anticipates how corrections to FRET efficiency values would continue to trend into the confocal regime as initially posited by Camley et al, *J. Chem. Phys.*, **2009**, *131*, 104509. As we have now made clearer in our revised manuscript, we fully appreciate that counterposing complexities can arise in elevated illumination intensity regimes that lead to departures from this simplified framework, which relate to energy transfer events between donor and acceptor fluorophore excited states, as described by Nettels et al., *Phys. Chem. Chem. Phys.*, **2015**, *17*, 32304-32315.

To address this point directly, we have now included additional wide-field TIRF imaging experiments in which we have extended the illumination intensity into the confocal regime (excitation rates $\sim 3 \times 10^6 \text{ s}^{-1}$) (**Fig. 2c-e, revised manuscript**). As expected from theory, the FRET efficiencies for both the Cy3-Cy5 and LD555-LD655 FRET pairs continue to decrease with increasing illumination intensity. We have also included new data showing power-dependent changes in FRET efficiencies using confocal-based imaging platforms (**Supplementary Fig. 3; revised manuscript**). These findings support our conclusion that the first principles stipulate that triplet states contribute to power-dependent changes in FRET efficiency when experiments are performed using both wide-field TIRF and confocal imaging platforms.

We note, however, that the observed deviations in FRET efficiency obtained using confocal imaging platforms depart from those anticipated by the 9-state photophysical model (**Fig. 4a, revised manuscript**), consistent with contributions of excited state energy transfer phenomena, such as RISC, SSA and STA (Nettels et al., *Phys. Chem. Chem. Phys.*, **2015**, *17*, 32304-32315). As noted by Nettels et al., these additional complexities tend to reduce the demand for zeta corrections. These findings further demonstrate the main point of our manuscript that excited state occupancies, which are predominated by relatively long-lived triplet states, confound FRET efficiency calculations, and thus must be effectively mitigated in order to reliably, reproducibly, and predictably recover accurate FRET efficiency information from experimental data.

Reviewer #2

Pati et al. investigate to what extent standard correction methods for FRET efficiency fail at elevated illumination intensities. They demonstrate this experimentally using TIRF for a set of three FRET dye pairs commonly used in single molecule microscopy measured under many different conditions. They find that triplet-state depopulation, either achieved by adding triplet state quenchers to the imaging buffer, or by using “self-healing” fluorophores, mitigates the problematic illumination-intensity dependence of FRET data.

The possibility of using elevated illumination-intensities is highly desirable for many applications in smFRET where improved time resolution is needed. The plethora of data presented here is impressive and will surely help investigators to choose optimal dye-pairs and conditions for studies where time resolution is key. Especially, the authors demonstrate that the “self-healing” dye-pair LD555-LD655 is much better suited than its closely related pair Cy3-Cy5 which is lacking “self-healing” capacities.

RESPONSE 1: We are thankful to the Referee for their positive feedback and appreciation of the work.

Furthermore, the authors suggest a correction factor, zeta, that is meant to correct for any residual power dependence of the measured transfer efficiencies. However, for practical and for theoretical reasons I find the proposed correction procedure not convincing. The authors suggest that zeta should be calculated from a detailed photophysical model that describes the FRET process itself and in addition all mechanisms that lead to the power dependence of the measurements. Practically, however it might be quite difficult to pick the right model and to know or determine all model parameters for calculating zeta. Especially, if they depend strongly on the experimental details, as for example on the presence of triplet quenchers in the local environment of the fluorophores.

RESPONSE 2: We agree that illumination intensity-dependent changes in FRET efficiency can be an inherently difficult consideration and can present significant challenges to the effort of recovering accurate FRET efficiency information from experimental data. The main message of our manuscript is that triplet states are by far the longest-lived excited states (ca. microseconds compared to nanosecond singlet state lifetimes) and that robust and reliable triplet state quenching significantly aids this goal. Variable and uncertain triplet state quenching efficiencies should thus be avoided. Confocal measurements typically ensure this is the case by utilizing fluorophores that rapidly undergo redox chemistry, which can be efficiently quenched by either including high concentrations of BME (approximately 143 mM at ~1% vol/vol dilution in buffer) or by performing experiments under ambient oxygen conditions. In the wide-field TIRFM imaging arena, however, more biologically compatible conditions, and less redox active organic fluorophores are typically employed, where oxygen scavenging conditions are used to maximize total photon budget over the extended period of the experiment.

While experimentalists using confocal platforms have been introduced to the idea that unexplained, empirical correction factors are often necessary to align static FRET states with respect to the diagonal axis in multiparameter single-molecule FRET measurements (see discussion of “bright fraction, a_b ” in Hellenkamp et al. Nature Methods 2018 for instance), empirical corrections of this kind are not an accepted standard in using wide-field TIRF systems (see Roy et al. Nature Methods, 2008 for instance). Hence, there is a gap in knowledge in the field as to the origins of FRET efficiency deviations from expectation, particularly at elevated illumination intensities, which as the Referee points out are of increasingly broad interest as experimentalists seek to improve time resolution and to differentiate spatial resolution.

Our manuscript addresses this issue by first demonstrating the phenomenon and then providing a strategy for explicitly correcting for excited-state (principally triplet state) deviations from ideality in the form of a revised E_{theory} formula that introduces the variable zeta as a correction factor (see equation 6, in our revised manuscript). When excited state (triplet state) occupancies are negligible, this term drops out, leaving the canonical FRET efficiency relationship. When triplet states are efficiently quenched, as is the case for the LD 555/655 FRET pair, then zeta is non-zero but relatively small until excitation rates above $5 \times 10^7 \text{ s}^{-1}$ (Fig. 4f, revised manuscript). As we now include in our revised manuscript, zeta corrections are also able to correct for illumination intensity-dependent deviations in FRET efficiencies that also arise from singlet excited state accumulations, which are also expected to arise at excitation rates above 10^8 s^{-1} (Fig. 4f, revised manuscript).

The Referee correctly points out that inclusion of the zeta parameter in the calculation of FRET efficiency requires knowledge of the photophysical properties of FRET pairs employed. Most of the necessary photophysical parameters are well established and known (e.g. excitation rate and singlet state (fluorescence) lifetimes). The main variable that is often not readily known by most experimentalists are the donor and acceptor fluorophore triplet state lifetimes, which we show are the most highly populated, and thus contribute the most uncertainty to deviations in FRET efficiency at elevated excitation rates (Fig. 4d,e, revised manuscript). We appreciate that most experimentalists do not explicitly measure triplet

state lifetimes. However, in our use of the zeta correction, we estimated the triplet state lifetimes of both donor and acceptor fluorophore using fluorescence correlation spectroscopy (FCS) for Cy3B and Cy5B and laser flash photolysis (LFP) for LD555 and LD655. The values obtained approximate the triplet state lifetimes of the donor and acceptor fluorophores under the experimental conditions but are unlikely to be exact. Approximate values can also be used for the k_{isc} parameter, which is reported to be on the order of $1e7\text{ s}^{-1}$ for most organic fluorophore species within the visible spectrum (Chibisov et al., *J. Chem. Soc., Faraday Trans.*, **1996**, *92*, 4917–4925). Our findings suggest that such approximations provide reasonable estimates and we hope that our manuscript can contribute to the substantial efforts currently on-going to increase the precision and accuracy of smFRET measurements (e.g., Hellenkamp et al., *Nature Methods*, **2018**, *15*, 669–676; Lerner et al., *eLife*, **2021**, *10*:e60416; Götz et al., *Nature Communications*, **2022**, *13*, 5402; Agam et al., *Nature Methods*, **2023**, *20*, 523–535).

On theoretical grounds, I am criticizing the photophysical model proposed by the authors, because it is neglecting quite relevant photophysical effects, namely triplet-singlet and singlet-singlet excitation annihilation (STA and SSA) as well as reverse intersystem-crossing (RISC) (see for example: van Grondelle, *R. Biochim. Biophys. Acta* **1985**, *811*, 147; or Hubner et al. *Phys. Rev. Lett.* **91**, 093903;2003). These processes are themselves inter-dye distance depending. Including these effects into the proposed correction would hence make zeta itself distance dependent and therefore most likely useless for correction.

Fig. R1 (Fig. 2c,e, revised manuscript). Power-dependent smFRET imaging of (a) Cy3-Cy5 and (b) LD555-LD655 dye pairs attached to a DNA duplex as depicted in Fig. 2a, revised manuscript. The experiments were performed in deoxygenated imaging buffers in the absence of any solution-based photostabilizing agents. The data at 0.04 and 3.6 kW/cm^2 were at 100 and 5 ms time resolution, respectively.

RESPONSE 3: We appreciate the need for discussion of complexities arising from excited state accumulations (SSA, STA, and RISC). These do require consideration, particularly for the Cy3/5 FRET pair which exhibits significant excited state accumulations (mainly due to triplet states) at elevated excitation rates. We have revised our manuscript to address this important point. Van Grondelle et al. examined tetrapyrrole-based photosystems. Hubner et al. examined perylenemonoimide chromophores. Both Camley et al. and Nettels et al. considered rhodamine-based dyes. SSA, STA and RISC processes can indeed occur for these distinct dye systems. Based on theoretical considerations, Camley et al. predicted that illumination intensity-dependent deviations in FRET efficiency for the AF488 and AF594 FRET pair should come into play at elevated excitation rates, while Nettels et al. argued that the predicted deviations in FRET efficiency were largely negated by SSA, STA and RISC. While such studies may anticipate similar cancelations may exist for other dye pairs, we do not observe this to be the case for the widely employed cyanine-class organic fluorophores or other FRET pairs used in our experiments (see revised Fig. 2, summarized here in Fig. R1). These findings argue that triplet state accumulations at elevated excitation rates should be more

broadly considered in the field, which is not standard practice currently. This oversight is likely to compromise the accuracy and precision of FRET efficiency calculations when triplet states are not robustly quenched.

Fig. R2. Percentage of FRET efficiency decrease of Cy3-Cy5 and LD555-LD655 dye pairs. The experiments were performed in deoxygenated imaging buffers in the absence of any solution-based photostabilizing agents. The imaging at 3.6 kW/cm² were at 5 ms time resolution. Rest data were collected at 100 ms.

a 9-state photophysical model (Fig. 4f, revised manuscript). Such trends likely arise from the counterposing effects of STA and RISC as the Referee and prior literature (Nettels et al. Phys. Chem. Chem. Phys., 2015, 17, 32304-32315) suggest. This model was confirmed by simulations (Fig. R3a,b), which show that SSA, STA

To directly address the Referee comment, in our revised manuscript we have now included wide-field TIRF data at excitation rates commensurate to those used in confocal measurements (3.60 kW/cm², Fig. 2c,e; Fig. R1). We have also performed confocal measurements and included simulations that confirm that STA and RISC processes attenuate the magnitudes of zeta corrections required, as put forward by Nettels et al.

Below 1 kW/cm², we find good agreement between our experimental results and zeta corrections estimated using a simplified 9-state photophysical model, which does not include SSA, STA or RISC processes (Fig. 4a-c, revised manuscript). At elevated excitation rates, excited state accumulations, principally arising from donor and acceptor triplet states, do occur for both the Cy3/5 and LD555/655 pairs (Fig. 4d-f, revised manuscript), giving rise to increased probability of SSA, STA and RISC contributions. With the inclusion of these data, we can now show that FRET efficiency changes of both the LD555-LD655 pair and the Cy3-Cy5 pair plateau at elevated excitation rates (Fig. R2). Confocal microscopy experiments further confirm that both the Cy3/5 and LD555/655 FRET pairs continue to exhibit decreases in apparent FRET efficiency as a function of illumination intensity, where the rates of decrease are lower than predicted by simulations of the zeta correction values using

Fig. R3 (Supplementary Fig. 13a,b, revised manuscript). ζ correction in the presence and absence of alternative energy pathways. (a,b) Evaluated ζ correction for Cy3-Cy5 dye pairs (panel a) and LD555-LD655 dye pairs (panel b) where we assumed that $k_{SSA} = 5k_{ET}, k_{STA} = 10k_{ET}, k_{RISC} = 10^7 (s^{-1})$ (rates are chosen to show how they are affecting the zeta correction for a specific distance) such that we include $S_1^D S_1^A \xrightarrow{k_{SSA}} S_0^D S_1^A, S_1^D T^A \xrightarrow{k_{STA}} S_0^D T^A$ for the SSA and STA investigation and $S_1^D T^A \xrightarrow{k_{STA}} S_0^D T^A$ with $S_1^D T^A \xrightarrow{k_{RISC}} S_0^D S_1^A$ for both STA and RISC investigation.

and RISC annihilation process tend to reduce zeta correction values as a function of illumination intensity and that both processes are less significant for the LD555/655 FRET pair, which exhibit reduced excited state accumulations

due to their short triplet state lifetimes (revised Fig. 4d,e). These data, together with targeted 4-state photophysical model simulations (only S0 and S1 excited states for donor and acceptor fluorophores) (Fig. R3a,b) demonstrate that SSA processes do not contribute meaningfully for Cy3/5 and LD555/655 FRET pairs until excitation rates exceed 10^8 s^{-1} (Fig. 4d-f, revised manuscript). The key takeaway from these analyses is that robust and reliable triplet state quenching – as achieved through intramolecular processes for the LD555/655 FRET pair - provides a general strategy for reducing excited states accumulations, which obviates the need for relatively large, and potentially variable zeta corrections. In this regime, STA and RISC processes contribute negligibly, thus reducing the consideration of zeta correction to a 9-state photophysical model.

The proposed correction ζ zeta is not novel. It is related to the parameter Λ , which was introduced by Camley et al., simply by $\zeta = \Lambda - 1$. (Compare Eq. 3 of the manuscript to Eq.4 of Camley et al.) The introduction and discussion of Λ is the main content and result of the paper by Camley et al.

RESPONSE 4: As we attempted to acknowledge in the citations presented in our initial manuscript, we agree that Camley et al. reported that triplet states should theoretically affect FRET efficiencies and that this paper predicted the need for a correction parameter, Λ , that would take such factors into consideration. However, with the work of Nettels et al. showing that SSA, STA and RISC processes obviate the need for such corrections for the specific rhodamine dyes they examined, this concept has since been overlooked by most in the field (as evidenced by the comments from Referees 1 and 3 for instance). While we are aware of the correction value from Camley et al., it is worth noting that the information provided in that paper did not reduce to practice consideration of even the simplified 9-state photophysical model. As noted in Camley et al., " Λ_9 [a 9-state photophysical model] is exceedingly complex; we do not present the analytical expression here." In our revised manuscript, we more carefully acknowledge the work of Camley et al. as well as Nettels et al. to clarify their contributions and explicitly state the relation between ζ and Λ as well as the advances of our work.

The interdye distance dependence of Λ due to STA, SSA and RISC was later pointed out by Nettels et al. The authors of the manuscript cite Nettels et al., but surprisingly, they do not address the results of that publication.

RESPONSE 5: We thank the Referee for bringing this critical deficiency to our attention. As discussed above, in our revised manuscript we have now explicitly stated in the main text that Camley et al. put forward the idea based on theoretical grounds that specific rhodamine FRET pairs should display power-dependent FRET changes and that such changes parallel the extent of triplet state occupancy, which should increase proportionally as a function of illumination intensity and for the acceptor fluorophore as a function of FRET efficiency. We also clarify that Nettels et al. did not observe illumination intensity-dependent FRET changes for the specific system they examined, which they attributed to competing excited state energy transfer pathways (SSA, STA and RISC).

In addition to this, I need to say that the parts of the manuscript that concern theory are not well written. Especially, the SI, that contains almost all important details on the theory behind zeta, needs a thorough proofreading and clarification of nomenclature and definitions.

RESPONSE 6: We sincerely thank the Referee for this assessment. In our revised manuscript, we have made every effort to address this deficiency by making changes in our notations as well as the explanations provided in both the supplementary information as well as the main text, including the addition of supplementary tables to improve consistency and readability.

In the following I present in more detail my criticism and comments:

- The authors use a nomenclature for transfer efficiencies that is confusing, if not inconsistent. Surprisingly, I find no clear definition of what the authors mean with “true transfer efficiency” in the main text, although this is a key term used throughout the manuscript. On page 12, the letter ε is introduced for denoting this quantity, “(ε ; true FRET efficiency)”. And in Eq. 2 it is used to define the correction parameter ζ . However, in my understanding, which is backed by what I read in the SI, ε is meant to be the measured (apparent) transfer efficiency, which is corrected for gamma, crosstalk, and acceptor direct excitation according to the standard procedures found in literature. The measured ε approximates well, i.e., with high accuracy, the transfer efficiency defined by Förster theory in the limit of low excitation intensity. (Additional low shot-noise and/or averaging over many events (bins) is needed to get precise values.)

The definition of transfer efficiency according to theory can be found on page 23 of the SI and is denoted there as E_{true} . (Including k_{isc} in the denominator is correct, although for all practical uses negligible.) For this quantity Förster showed that $E_{true} = 1/(1 + (\frac{R}{R_0})^6)$. According to Eq.1 of the main text, this is equal to what the authors call E_{std} .

RESPONSE 7: We thank the Referee for bringing all these points to our attention. As mentioned above, we have now revised our main text and supplementary information documents to ensure consistency and clarity.

The definition of the transfer efficiency is independent of k_{ex} . Hence, the limit “ $k_{ex} \ll k_{em}^D, k_{em}^A$ ” for the validity of Eq.1 is needed only in the sense that stimulated emission is neglected in the theory. But this is not the point the authors want to make here. Presumably, they want to state that the measured value ε depends on k_{ex} and approximates E_{true} only well in the limit $k_{ex} \rightarrow 0$. Alternatively, one could read Eq.1 not as a defining equation but such that E_{std} is defined as the measured and with the standard method corrected value of transfer efficiency, but then E_{std} would equal ε according to its definition in the SI. With this Eq.2. would become pointless. The authors need to clarify.

RESPONSE 8: We again thank the Referee for their request for additional clarity. We have now revised our main text and supplementary information documents to explicitly clarify these points, aided by the inclusion of 4-state photophysical model simulations.

(On page 30 of the SI yet another symbol is introduced, $\varepsilon_{corrected}$, which algebraically equal to E_{std} in Eq.2 of the main text.) Notation needs to be harmonized throughout both texts. Beside this I suggest to replace “true” by “accurate” throughout the text, whenever appropriate, for example in the title of the manuscript.

RESPONSE 9: We have now changed all the notation for the physical quantities of interest as tabulated in our revised supplementary material.

- The correction factor zeta introduced in the manuscript is obviously only useful if it is not itself dependent on the inter-dye distance R . The authors show in the SI for the photophysical model depicted in Fig.4 that this is indeed the case. (To be accurate, they show it for the case that direct excitation of the acceptor is negligible.)

However, this model, which is identical to the one presented in Camley et al. (Ref. 6 in the SI), does not include well-known photophysical effects that are indeed R -dependent and might be relevant.

RESPONSE 10: We have now addressed these points in our revised manuscript as described in our responses 2–6 above.

The model assumes for example that no energy is transferred from the donor $S1$ state to the acceptor dye, while the latter is in triplet state $T1$. In fact, for Cy5 it has been shown (Huang et al. *J. Phys. Chem. A* 2006, 110, 1, 45–50) that the absorbance spectrum of $T1 \rightarrow Tn$ is similar to the absorption spectrum of $S0 \rightarrow Sn$. As a consequence, energy transfer is expected, lifting the $T1$ state of the acceptor to a higher triplet state Tn . Subsequently, this can lead to singlet-triplet annihilation (STA) or reverse inter-system crossing (RISC). In the first case, in contrast to the model prediction of Fig.4a, no donor photon is emitted, in the second case, the acceptor is in $S1$, and can immediately emit a photon. These and yet another effect, namely singlet-singlet annihilation (SSA), are inter-dye distance dependent and lead to lower deviations of the apparent transfer efficiency from the true one than predicted by the model of Fig.4a. This was previously investigated by Nettels et al. Their paper is cited in the manuscript, but possible consequences, most importantly, a possible distance dependence of zeta due to STA, RISC, and SSA processes are not discussed. (Also, a discussion of the possible role of photo-induced cis/trans isomerization of Cy5 would be desirable in this context.)

RESPONSE 11: We agree and thank the Referee for pointing out the need to explicitly address and clarify the potential complexities of excited state processes such as SSA, STA and RISC. We have addressed these points in our revised manuscript, as discussed above. We now fully acknowledge the potential for such alternative energy transfer pathways to complicate FRET efficiency measurements and that such pathways can be minimized by the mitigation of triplet excited states, which tend to accumulate at elevated excitation rates when not efficiently quenched.

- If or if not, zeta is distance dependent for the dye-pairs and under the conditions studied here could be easily investigated by experiment. I suggest that the authors measure constructs with different labeling positions at some low-power and some high-power intensities and check if the reciprocal distances of measured transfer efficiency ($\frac{1}{\epsilon_{highpower}} - \frac{1}{\epsilon_{lowpower}}$) are the same for all inter-dye distances. The constructs represented in Fig. 6a would be ideal for this purpose.

RESPONSE 12: We thank the Referee for this suggestion. During the initial stages of our investigations, we observed that FRET efficiency changes tend to be lower for samples that exhibit low ($\sim < 0.30$) and high ($\sim > 0.70$) FRET efficiency values as the Referee suggests. As shown in Fig. R4, the reciprocal of FRET efficiency differences between a high and low laser intensity is generally higher for samples that exhibit intermediate FRET efficiencies.

- The authors might argue that their model is well justified simply by its success to correct the transfer efficiencies. However, some of the parameters (k_{TR} and k_{ISC}) were “optimized” in order to match measured and simulated efficiencies. This was done only for one inter-dye distance R. In view of this, a model-free check like the one described above seems to be important to me.

RESPONSE 13: Our triplet state lifetime estimates were guided by FCS data. Our intersystem crossing rate estimates were guided by literature values reported for organic fluorophores (Chibisov et al., *J. Chem. Soc., Faraday Trans.*, 1996, 92, 4917–4925). We seeded our simulations with these information, as well as other photophysical parameters including fluorescence emission rates, to then arrive at intersystem crossing

Fig. R4. Donor-acceptor dye distance-dependent smFRET imaging. (a) Schematic diagram of a mixture of six DNA duplexes labeled with a donor dye at the terminal site (5'-end) of a DNA strand and an acceptor fluorophore at an internal position of its complementary DNA strand. The position of the acceptor fluorophore was varied in all the duplexes, in which the acceptor fluorophore was 5, 8, 11, 14, 17, and 20 nucleotides (nt) apart from the donor fluorophore. (b,c) Bar diagrams of $\left(\frac{1}{\epsilon_{\text{highpower}}} - \frac{1}{\epsilon_{\text{lowpower}}}\right)$ values for various DNA duplexes as schematized in the above scheme at various illumination intensities. Error bars represent standard deviations.

(hopefully with the exception of R) need to be known. The problem with this is, from my point of view, that the photo physical mechanisms that kick in at elevated excitation powers are often not known to such an extent that a reliable quantitative calculation are feasible. Also, the fact, that kinetic parameters depend on environmental details and experimental conditions make an accurate calculation of zeta very challenging.

RESPONSE 14: The Referee is correct in their assessment that unknown and/or varying triplet excited state accumulations present significant challenges to estimating the zeta correction factor values needed to accurately and precisely determine FRET efficiencies, particularly at elevated excitation rates. This consideration, however, emphasizes a main point of our manuscript, which is that FRET efficiency calculations require that efforts be made to robustly mitigate triplet state occupancy to prevent excited state accumulation. As we show for the LD555/655 FRET pair, when excited states are sufficiently depopulated, SSA, STA and RISC processes become negligible and zeta corrections can be estimated from a 9-state photophysical model. Our findings thus motivate researchers to minimize deviations in FRET efficiency arising from triplet excited state accumulation and a mathematical framework for correcting for such impacts.

• Why is direct acceptor excitation neglected for the calculation of zeta in the SI? Acceptor photon emitted while the donor is in the triplet might lead to an apparent higher transfer efficiency. This might be relevant especially for the distance regime beyond R_0 .

rates consistent with the data, which was within ~50% of the seed value. We allowed this variable to float because we felt it was the most uncertain. As we show in (Fig. 5e, revised manuscript the k_{ISC} value that we arrived at using this approach recapitulated the experimental data for different FRET efficiency values.

• The authors suggest to calculate zeta based on an appropriate photophysical model. For this essentially all model parameters

RESPONSE 15: We are grateful to the Referee for their careful assessment of our manuscript and catching this oversight. We have now included direct acceptor excitation considerations in our analyses. We note that this correction does not affect the main message of our manuscript.

- Joined states are represented in Fig.4a (and in corresponding figures of the SI) as sums, but in the text as products. As joined quantum states, the latter notation makes more sense to me.

RESPONSE 16: We thank the Referee again for catching this mistake. We have now corrected the issue both in the main manuscript and supplementary information files.

- The vertical dimension of Jablonski diagrams usually represents energetic relations between the energy levels. For Fig. 4a this seems to be not the case.

RESPONSE 17: The author is correct. Jablonski diagrams, such as schematized in Fig.1b, are typically drawn to represent energetic relations between excited states. In our revised manuscript, we now clarify in the figure legend for the 9-state photophysical model that “for simplicity, the presented schematic does not maintain energetic relations between excited states in the vertical dimension”.

- In Fig. 4a and in the corresponding figure in the SI are transitions missing which are part of the model described by Camley et al. and which are also described by the rate equations on page 28 of the SI. Why are these transitions omitted in the diagrams?

RESPONSE 18: We thank the Referee for pointing out the missing connections in our pictorial representation of the 9-state photophysical model. We have now revised Fig. 4a to address this deficiency. We note that all connections were included in our presented analyses yet were missing from our schematic (see revised Fig. 4a, as well as the schematic provided in the supplementary material).

- The wavy arrows in the diagrams and their labels $k_{em}^{A,D}$ suggest that these transitions are purely radiative (quantum yields equal to one). Is this intended?

RESPONSE 19: Yes. The wavy arrows represent radiative processes. We have described all the notations in our revised **Supplementary Table 1**.

- In Eq. 3 the authors limit the validity to k_{ex} larger or equal the emission rate coefficients. This limit is never reached. At such elevated excitation rates spontaneous emission would be needed to be taken into account. On the other side, Eq. 3 is valid for very low k_{ex} values, because zeta becomes zero in this limit.

RESPONSE 20: We agree with this point. To avoid confusion, we have now excluded the boundary conditions that were initially presented.

- The triplet and diffusion components in the FCS curves are hardly distinguishable. I expect the error on k_{TR} to be substantial. The authors should report the results for all fitting parameters with errors.

RESPONSE 21: The fit itself is very good and the fitting error of the resulting parameters (1-2%) is relatively small. See tables below (**Table R1**) from the fit of Cy3B (left table, green) and Cy5B (right table, magenta) correlation curves at 10 μ W presented in **Supplementary Fig. 8** (which now includes the same two tables beside the fitted correlations), where N is the average number of molecules in the confocal volume, τ_D is the diffusion time in microseconds (μ s), s is the ratio of the lateral to the axial radii of the confocal volume;

A_{TR} and τ_{TR} are the amplitude and lifetime (in μs) of the triplet state population; τ_{ab} is the antibunching time in ns and A_{ab} is its associated amplitude.

Table R1: Fitting parameters of FCS curve of Cy3B.

	Estimate	Standard Error		Estimate	Standard Error
N	5.51	0.031	N	0.98	0.0059
τ_D	226	2.47	τ_D	173	1.72
S	0.14	----	S	0.14	----
A_{TR}	0.46	0.0082	A_{TR}	0.28	0.0077
τ_{TR}	33.4	0.50	τ_{TR}	40.8	0.79
A_{ab}	-0.92	0.016	A_{ab}	-0.88	0.013
τ_{ab}	1.97	0.038	τ_{ab}	1.86	0.029

We have varied the starting parameters for the FCS fit 400% in both directions, resulting in nearly identical fits in all cases (data not shown).

A bootstrapping analysis of the Cy3B data set (Fig. R5) plotted in Supplementary Fig. 8 (10^4 cycles maintaining the same sample size), resulted in a distribution of values with a virtually identical mean value to the fitted triplet lifetime (33402 ns vs. 33379 ns) and standard deviation similar to the fitting error (581 ns vs. 487 ns), further indicating that the fit is robust and well converged.

Fig. R5. The graph of Cy3B bootstrap analysis.

Thus, we chose a different approach to obtain a more meaningful estimation of the uncertainty through power-dependent measurements (see next point).

- Did the authors try to measure FCS at higher laser powers, where the components might be better distinguishable?

RESPONSE 22: The Referee makes a very good point. FCS experiments at higher power had indeed been performed for both Cy3B and Cy5B (with a 100 μm pinhole). Unfortunately, the expected increase in triplet amplitude at laser powers > 10 μW was accompanied by significant photobleaching, which shortens the diffusion time and causes an ever greater “coupling” between triplet blinking and diffusion (see plots below for Cy3B showing a progressively shorter diffusion time and a concomitant shortening of the triplet lifetime and reduction of its amplitude, Fig. R6).

Fig. R6. Changes of diffusion time, triplet lifetime, and amplitude of triplet component with increasing the laser power.

This is why the results that we present in the paper have been obtained with an experimental setup that was optimized to keep conditions between TIRF and confocal experiments as close as possible (same microfluidics channel and buffer composition), while trying to have the maximum separation between blinking and diffusion: 150 μm pinhole to extend the diffusion time and 10 μW (measured at the back aperture of the microscope) to minimize photobleaching, which would shorten the apparent transit time of the dye through the confocal volume.

To better capture the intrinsic uncertainty in triplet time estimation, we now use the triplet lifetimes estimates obtained from measurements at two low laser powers using a 150 μm pinhole to extrapolate mean lifetime and associated uncertainty at zero power (below in Fig. R7, left panel in green for Cy3B and right panel magenta for Cy5B; values in **Supplementary Table 1**). The linear extrapolation remains an approximation, but we believe that the error introduced by assuming linearity over a very small power range is probably negligible for our purposes. This approach yielded values of $31 \pm 5 \mu\text{s}$ and $51 \pm 8 \mu\text{s}$ for Cy3B and Cy5B, respectively, compared to the average and uncertainty obtained at 10 μW of $30 \pm 4 \mu\text{s}$ and $47 \pm 9 \mu\text{s}$; note that as part of the power dependent estimates of triplet state lifetimes, in addition to new measurements at 50 μW , one new measurement has been recorded also at 10 μW and included in the average for both dyes.

Fig. R7. Changes of triplet lifetimes at two different laser powers.

- I assume the reported 10 mW measured at the back aperture of the objective for FCS are meant to be 10 microwatts? What values does one get for k_{ex} ?

RESPONSE 23: The referee is correct: 10 mW should have been stated as 10 μ W. The power is the one measured at the back aperture of the microscope.

At this illumination intensity k_{ex} is $\sim 5 \times 10^6 \text{ s}^{-1}$ using a naïve conversion formula (Sakhapov et al., *J. Phys. Chem. Lett.*, **2022**, *13*, 4823–4830), and $\sim 4 \times 10^6 \text{ s}^{-1}$ from measured lifetime and antibunching time for Cy3B according to $\tau_{ab} = \frac{1}{\tau_{fl} + k_{ex}}$.

For Cy5B, the value calculated with the same formula is $\sim 6 \times 10^6 \text{ s}^{-1}$ and the one obtained from the measured lifetime and antibunching is $\sim 10^7 \text{ s}^{-1}$.

- The amplitude of the triplet component is $c_T \approx \frac{k_{ex} k_{ISC}}{k_{fl} k_{TR}}$, the expected value could be calculated from the parameters given in SI Table 1. FCS measurements at varied laser powers could help to obtain a robust and independent value for k_{ISC} .

RESPONSE 24: As discussed above, given the similarity of timescales between triplet state and translational diffusion, the analysis suggested by the referee, although possible in principle, is regrettably not attainable in practice.

- The first plus sign in the formula for fitting the FCS data is there by mistake. In SI Fig.10, the x-axis label should be “tau/ns” instead of “log(tau/ns).

RESPONSE 25: We thank the Referee for spotting these errors, which have been rectified in our revised manuscript.

- The authors determine the excitation rate coefficient simply from the measured light intensity of the laser beam and the extinction coefficient of the fluorophores. However, both in TIRF (evanescent field) and as well in confocal microscopy (tight laser focus) the excitation rate varies drastically across the sample volume; for TIRF especially in z-direction. How does this effect the results? Would it be better to use a spatial averaged value for k_{ex} ?

RESPONSE 26: We appreciate the opportunity to clarify these points. In short, we do not expect these considerations to affect our results in discernable ways and the calculated k_{ex} is indeed a spatial average. For TIRF microscopy, the illumination intensity is fairly homogeneous in the X,Y-dimensions and we calculate the excitation rate considering that much of the beam is outside the field of view. For the Z dimension, the fluorophores are expected to be at a uniform depth from the interface; the observed distribution of fluorescence intensities is consistent with this idea. We estimate the fluorophores are roughly 28 nm from the interface and the penetration depth is 120 nm. This suggests the error relative to a naïve calculation (assuming the fluorophores are at the TIR interface) is less than 20%. For simplicity we neglect this specific effect, but this approximation is now explicitly described in the methods for clarity.

For confocal illumination of diffusing particles, the illumination scheme is indeed much more complex. We do not explicitly treat this complexity in the manuscript and instead only consider the regime of intensities that are expected for confocal imaging of diffusing particles.

• SI on page 23, “In an attempt to keep dyes’ quantum yields unchanged, we incorporate the information that $k_{fl A,*} = k_{fl A} - k_{ISC A}$ (also denoted as $k_{em A}$) and $k_{fl D,*} = k_{fl D} - k_{ISC D}$ (similarly denoted as $k_{em D}$). “This sentence is unclear to me. What should change the quantum yield in a theoretical calculation/simulation? What kind of information is meant. The symbols $k_{fl D,A}$ were not introduced before. Why are two symbols ($k_{fl D,A,*}$ and $k_{em D,A}$) introduced for the same thing? Again, are these meant to be purely radiative rates? If yes, why is not the general case treated? Can it generally be excluded that zeta depends on differences in quantum yields?”

RESPONSE 27: We thank the Referee for giving us the opportunity to clarify these points. To avoid confusion, we have now removed this discussion and we have now provided $k_{S_1}^D$ instead of k_{fl}^D and included its specific definition (Table R2).

Table R2: Description of $k_{S_1}^D$.

4 state model	9 state model
$k_{S_1}^D = k_{em}^D + k_{nr}^{S_0^D}$	$k_{S_1}^D = k_{em}^D + k_{nr}^{S_0^D} + k_{ISC}^D$

• “In our simulations we consider 20% detection efficiency, ...” The same for both colors?

RESPONSE 28: Yes, the donor and acceptor channels have similar overall collection efficiency in our optical setup. We have clarified this point in the revised supplementary information as follows: “In our simulations we consider 20% detection efficiency, estimated for both green and red channels from the transmission specifications of the instrument’s individual components”.

• “Otherwise, the effect of the direct excitation of the acceptor fluorophore would be on the denominator of the E_{true} . However, it is known to be a very small contribution to the true FRET efficiency.” These sentences are unclear to me. According to the equation right above this statement, E_{true} is just representing the definition of FRET efficiency according to theory for this model. How could this possibly depend on the acceptor direct excitation? Again, I do not really understand what the authors mean with “true” FRET efficiency. For my taste it would be preferable not to use this problematic term. Instead, one needs to distinguish between the theoretically defined quantity and the measured value. The latter can be more or less accurate, but not “true” in my opinion. Correction factors correct for known systematic errors. If systematic errors remain uncorrected for, one can speak about apparent transfer efficiencies.

RESPONSE 29: We apologize for introducing potentially confusing language in our initial manuscript. We now utilize the terms E_{theory}^{4st} and E_{theory}^{9st} corresponding to the associated 4 state and 9 state photophysical models, respectively, throughout the main text and supplementary information.

• “We carry out the correction on the raw FRET efficiency as more complexity is getting introduced while the FRET efficiency is corrected such as quantum yield difference as well as detection efficiencies in both donor and acceptor channels.” How is “raw FRET efficiency” defined? Please check the grammar and semantics.

RESPONSE 30: We again thank the Referee for pointing out confusing terminology. We have revised our manuscript to address these shortcomings.

- “We follow the same photophysical trajectory generation steps for both donor and acceptor fluorophores in probing high spatial resolution performance of smFRET imaging.”
In what sense do the simulation represent spatial resolution of the imaging?

RESPONSE 31: We have removed this wording from our revised manuscript.

- A careful proofreading of the entire SI is needed.

RESPONSE 32: We appreciate the opportunity to revise and clarify our supplementary information file to maximize clarity.

- The used Forster radii for the dye pairs should be reported.

RESPONSE 33: This information is now provided in SI Table2.

- The vertical dashed lines in Fig.6(b-d) make sense for comparing experimental (c) and theoretical (d) results for the LD555-LD655 dye pair. However, for Cy3-Cy5 (b) another Forster radius applies; a direct comparison is hence not possible in FRET efficiency space.

RESPONSE 34: The Referee is correct. We have now rectified the issue according to their suggestion.

- “We initiated these investigations by calibrating to experimental and simulated values for Cy3B fluorophore brightness across a range of excitation rates commensurate with those used to interrogate Cy3”. Please clarify why Cy3B instead of Cy3 was used here. Given the high popularity of C3B and Cy5B, it is surprising that they were here not further investigated as a FRET pair. A direct comparison with Cy3-Cy5 would be very interesting.

RESPONSE 35: As mentioned in the legend of **Supplementary Fig. 8**, the rigidified Cy3 (Cy3B) and Cy5 (Cy5B) fluorophores simplified the fluorescence correlation measurements so that triplet state lifetimes could be robustly recovered. We have not used Cy3B-Cy5B FRET pair in our TIRF imaging studies as their rigid chemical structures cause aggregation/stacking phenomena on DNA resulting in poorly behaved single-molecule FRET traces.

Reviewer #3

With much interest I have reviewed this manuscript on the demonstration of I_{exc} -dependent changes of E , and the remediation thereof via either a correction parameter or intramolecular T1-quenching strategies.

The work first is a thorough comparative analysis of normal and self-healing dyes in TIRF/widefield single-molecule (FRET) analyses, and their dependence on TSQs and I_{exc} . The experiments are complemented with kinetic simulations that support the proposed triplet-mediated effects on E . Authors also derive a correction parameter for the triplet-affected data.

I have no comments whatsoever on the quality of the experiments and simulations. Experiments have been performed and analyzed excellently, rendering data interpretation straightforward and leaving little room for biased interpretation. I applaud the authors for their thorough work.

The dzeta correction, in my opinion, is the main novelty of the work, and it can be a game changer. I have a few suggestions to improve the manuscript.

RESPONSE 1: We thank the Referee for their positive feedback and appreciating the work.

Intro:- Fig. 1A is clear, but somewhat obsolete as it is really just textbook info- Fig. 1B needlessly occupies 2x space as needed, as D and A can undergo, in principle and as illustrated, identical processes. Also, explicitly indicate where the photobleaching pathways are.

RESPONSE: We understand the Referee's desire for figure compression, and we have revised our manuscript to address this point by combining all theoretical data into a single figure (Fig. 4, revised manuscript). In the opening figure, we believe the general reader will benefit from keeping Fig. 1 in its presented form as it highlights the type of information smFRET seeks to attain as well as the photophysical complexities associated with the distinct donor and acceptor fluorophore excited states.

We have revised Fig. 1 as recommended to show the photobleaching step.

"The addition of high concentrations of TSQs in solution can, however, exhibit drawbacks, and irregularities linked to their varied and limited aqueous solubilities, excited-state reactivities, and potential toxicities to the systems under investigation, particularly when investigating living cells^{36,48,49}."

- Ref 48 does not support this statement directly, and ref 36 is a review. Cite original literature proving the statement of the authors. Same remark for the next sentence, where ref 36 is cited again.

RESPONSE 2: We thank the Referee for pointing out these referencing issues. We have now appropriately cited original literature in our revised manuscript.

"Here, we demonstrate that the experimentally determined FRET efficiencies of a given dye pair vary as a function of illumination intensity..."

- A comparative analysis of different acceptor dyes regarding k_{exc}-dependent FRET changes was reported by Vandenberg et al 2018 J Phys Chem B., albeit without providing corrections for it. At least this publication could be cited.

RESPONSE 3: We have added the reference suggested by the Referee in our revised manuscript (ref. #17, revised manuscript).

Results- The authors have limited themselves experimentally to TIRF/widefield, while it is of interest to do a comparative experimental analysis also for the MFD/confocal counterpart. Considering the vastly different (much more brief excitation per molecule, and highly inhomogeneously) excitation geometry I do not agree the performance can simply be deducted from the experiments/simulations that have been done. Especially statements such as "Solution-based methods to suppress triplet states are therefore expected to fall short of meeting the field's goal of achieving accurate FRET efficiency estimations" are certainly valid for the current work, but unclear to what extent they are applicable (whether more or less) to MFD/confocal.

RESPONSE 4: We understand and appreciate this comment. To address the point directly, we have now included additional wide-field TIRF imaging experiments in which we have extended the illumination intensity into the confocal regime (excitation rates $\sim 3 \times 10^6 \text{ s}^{-1}$) (Fig. 2c-e, revised manuscript). As expected from theory, the FRET efficiencies for both for the Cy3-Cy5 and LD555-LD655 FRET pairs continue to decrease with increasing illumination intensity. We have also included new data showing power-dependent changes in FRET efficiencies using confocal-based imaging platforms (Supplementary Fig. 3; revised manuscript). These findings support our conclusion that first principles stipulate that triplet states

contribute to power-dependent changes in FRET efficiency when experiments are performed using both wide-field TIRF and confocal imaging platforms.

- Self-healing dyes may be spectroscopically beneficial, are they biologically equally compatible? This merits a discussion, at least.

RESPONSE 5: Our collaborative team has successfully employed self-healing organic fluorophores across a diverse array of complex biological systems at experimental settings, including live-cell imaging applications (see examples below)¹⁻²¹. Others have also utilized them successfully their biological applications.²²⁻²⁹ We have now included some of these citations in our revised manuscript (refs. # 63-72) to clarify the practical advantages of the self-healing fluorophore technologies that our group is developing.

1. Holm, M. *et al.* mRNA decoding in human is kinetically and structurally distinct from bacteria. *Nature* **617**, 200–207 (2023).
2. Levring, J. *et al.* CFTR function, pathology and pharmacology at single-molecule resolution. *Nature* **616**, 606–614 (2023).
3. Wieland, M. *et al.* The cyclic octapeptide antibiotic argyrisin B inhibits translation by trapping EF-G on the ribosome during translocation. *Proc. Natl. Acad. Sci. USA* **119**, e2114214119 (2022).
4. Asher, W. B. *et al.* GPCR-mediated β -arrestin activation deconvoluted with single-molecule precision. *Cell* **185**, 1661–1675.e16 (2022).
5. Rundlet, E. J. *et al.* Structural basis of early translocation events on the ribosome. *Nature* **595**, 741–745 (2021).
6. Asher, W. B. *et al.* Single-molecule FRET imaging of GPCR dimers in living cells. *Nat. Methods* **18**, 397–405 (2021).
7. Ciftci, D. *et al.* Single-molecule transport kinetics of a glutamate transporter homolog shows static disorder. *Sci. Adv.* **6**, eaaz1949 (2020).
8. Morse, J. C. *et al.* Elongation factor-Tu can repetitively engage aminoacyl-tRNA within the ribosome during the proofreading stage of tRNA selection. *Proc. Natl. Acad. Sci. USA* **117**, 3610–3620 (2020).
9. Henderson, R. *et al.* Disruption of the HIV-1 Envelope allosteric network blocks CD4-induced rearrangements. *Nat. Commun.* **11**, 520 (2020).
10. Lu, M. *et al.* Shedding-Resistant HIV-1 Envelope Glycoproteins Adopt Downstream Conformations That Remain Responsive to Conformation-Preferring Ligands. *J. Virol.* **94**, (2020).
11. Fitzgerald, G. A. *et al.* Quantifying secondary transport at single-molecule resolution. *Nature* **575**, 528–534 (2019).
12. Elshenawy, M. M. *et al.* Cargo adaptors regulate stepping and force generation of mammalian dynein-dynactin. *Nat. Chem. Biol.* **15**, 1093–1101 (2019).

13. Lu, M. *et al.* Associating HIV-1 envelope glycoprotein structures with states on the virus observed by smFRET. *Nature* **568**, 415–419 (2019).
14. LeVine, M. V. *et al.* The allosteric mechanism of substrate-specific transport in SLC6 is mediated by a volumetric sensor. *Proc. Natl. Acad. Sci. USA* **116**, 15947–15956 (2019).
15. Gutzeit, V. A. *et al.* Conformational dynamics between transmembrane domains and allosteric modulation of a metabotropic glutamate receptor. *Elife* **8**, (2019).
16. Terry, D. S. *et al.* A partially-open inward-facing intermediate conformation of LeuT is associated with Na⁺ release and substrate transport. *Nat. Commun.* **9**, 230 (2018).
17. Dyla, M. *et al.* Dynamics of P-type ATPase transport revealed by single-molecule FRET. *Nature* **551**, 346–351 (2017).
18. Gregorio, G. G. *et al.* Single-molecule analysis of ligand efficacy in β 2AR-G-protein activation. *Nature* **547**, 68–73 (2017).
19. Lv, Z. *et al.* Direct Detection of α -Synuclein Dimerization Dynamics: Single-Molecule Fluorescence Analysis. *Biophys. J.* **108**, 2038–2047 (2015).
20. Akyuz, N. *et al.* Transport domain unlocking sets the uptake rate of an aspartate transporter. *Nature* **518**, 68–73 (2015).
21. Munro, J. B. *et al.* Conformational dynamics of single HIV-1 envelope trimers on the surface of native virions. *Science* **346**, 759–763 (2014).
22. Vyklicky, V., Stanley, C., Habrian, C. & Isacoff, E. Y. Conformational rearrangement of the NMDA receptor amino-terminal domain during activation and allosteric modulation. *Nat. Commun.* **12**, 2694 (2021).
23. Selvakumar, P. *et al.* Structural and compositional diversity in the kainate receptor family. *Cell Rep.* **37**, 109891 (2021).
24. Jack, A. *et al.* Compartmentalization of telomeres through DNA-scaffolded phase separation. *Dev. Cell* **57**, 277–290.e9 (2022).
25. Thibado, J. K. *et al.* Differences in interactions between transmembrane domains tune the activation of metabotropic glutamate receptors. *Elife* **10**, (2021).
26. Yang, Z. *et al.* SARS-CoV-2 Variants Increase Kinetic Stability of Open Spike Conformations as an Evolutionary Strategy. *MBio* **13**, e0322721 (2022).
27. Kusacki, E. *et al.* Lis1 binding regulates force-induced detachment of cytoplasmic dynein from microtubules. *BioRxiv* (2022). doi:10.1101/2022.06.02.494578
28. Ao, Y. *et al.* An intact amber-free HIV-1 system for in-virus protein bioorthogonal click labeling that delineates envelope conformational dynamics. *BioRxiv* (2023). doi:10.1101/2023.02.28.530526
29. Krishna Kumar, K. *et al.* Negative allosteric modulation of the glucagon receptor by RAMP2. *BioRxiv* (2022). doi:10.1101/2022.08.30.505955

-That acceptor blinking critically affects smFRET experiments has long been known. How does the D-A distance affect the A-caused problems as described in this paper?

RESPONSE 6: We thank the Referee for the opportunity to provide additional clarification of this point in our revised manuscript.

Figure 2 and others: I would suggest to add a graph of the (center of mass? Median? Gaussian fitted? PDA fitted?) Values (potentially their % deviation from the value at the lowest lexo as a secondary axis) to allow the reader to more easily view the deviations.

RESPONSE 7: We thank the Referee for the comment. We have now shown the % of FRET efficiency decrease values in **Fig. 2**, **Supplementary Fig. 2**, and **Supplementary Fig. 9** of our revised manuscript.

Decision Letter, first revision:

Dear Scott,

Thank you for your letter detailing how you would respond to the reviewer concerns regarding your Article, "Recovering true FRET efficiencies from smFRET investigations requires triplet state mitigation". We have decided to invite you to revise your manuscript as you have outlined, before we reach a final decision on publication.

[Redacted]

We hope to receive your revised paper within one month. If you cannot send it within this time, please let us know. In this event, we will still be happy to reconsider your paper at a later date so long as nothing similar has been accepted for publication at Nature Methods or published elsewhere.

OPEN SCIENCE REQUIREMENTS

REPORTING SUMMARY AND EDITORIAL POLICY CHECKLISTS

DATA AVAILABILITY

CODE AVAILABILITY

Please include a "Code Availability" subsection in the Online Methods which details how your custom code is made available. Only in rare cases (where code is not central to the main conclusions of the paper) is the statement "available upon request" allowed (and reasons should be specified).

For more information on our code sharing policy and requirements, please see: <https://www.nature.com/nature-research/editorial-policies/reporting-standards#availability-of-computer-code>

MATERIALS AVAILABILITY

SUPPLEMENTARY PROTOCOL

To help facilitate reproducibility and uptake of your method, we ask you to prepare a step-by-step Supplementary Protocol for the method described in this paper. We encourage authors to share their step-by-step experimental protocols on a protocol sharing platform of their choice and report the protocol DOI in the reference list. Nature Portfolio 's Protocol Exchange is a free-to-use and open resource for protocols; protocols deposited in Protocol Exchange are citable and can be linked from the published article. More details can found at www.nature.com/protocolexchange/about.

ORCID

Nature Methods is committed to improving transparency in authorship. As part of our efforts in this direction, we are now requesting that all authors identified as 'corresponding author' on published papers create and link their Open Researcher and Contributor Identifier (ORCID) with their account on the Manuscript Tracking System (MTS), prior to acceptance. This applies to primary research papers only. ORCID helps the scientific community achieve unambiguous attribution of all scholarly contributions. You can create and link your ORCID from the home page of the MTS by clicking on 'Modify my Springer Nature account'. For more information please visit please visit www.springernature.com/orcid.

Sincerely,
Rita

Rita Strack, Ph.D.
Senior Editor
Nature Methods

Reviewers' Comments:

Reviewer #1:

Remarks to the Author:

The manuscript by Pati et al. is a carefully conducted examination the power dependence of FRET

efficiency in single molecule experiments where high laser power becomes necessary to obtain increasing time resolution. The work clearly establishes the importance of accounting for triplet state occupancy of the fluorophores to obtain accurate measurements of absolute FRET efficiency. The authors show that this is particularly true of the acceptor triplet state, which can lead to deviations in FRET efficiency at high power.

This is of great significance to those in single molecule fluorescence and the growing field of dynamic structural biology, which require high power to obtain accurate measurements of intramolecular distances using FRET. That such a necessary correction has gone unnoticed for so long is surprising but Pati et al. have clarified the importance of this phenomena and describe methods for mitigation. As such, the work is a great fit for Nature Methods.

The manuscript is exceptionally well written and clear. The data are of high quality and the analysis is very rigorous with appropriate use of statistics throughout. The conclusions are well supported by the data. The study is well designed to illustrate the problem and provide mitigation strategies. The authors show the effect of different solution triplet state quenchers (TSQs), which all fall short of complete correction at the highest illumination power levels. The work highlights the importance of intramolecular triplet state quenchers as the only way to avoid this issue. However, the authors also provide a correction factor, which can recover the true FRET efficiency even when triplet state occupancy can't be prevented.

The manuscript is suitable for publication as is. I would suggest that it could be improved by adding a more standard TSQ cocktail in Figure 6, which is a highlight of the paper. This is a very well designed experiment that should become a standard calibration. However, the use of MV/AA as the TSQ is not optimal and not widely used in single molecule imaging. MV/AA also performs poorly in the author's own tests of TSQs (Fig S4 & S5). I get that the authors want maximum impact from their intramolecular TSQs but a more fair comparison would be to the COT/NBA/Trolox mix, which is much more effective at suppressing triplet state and preserving FRET. I would be curious to see how this mix performed in this experiment as these are the more standard TSQs in smFRET.

The manuscript also makes the contention that the power dependent effects are likely most relevant for confocal single molecule experiments, which are always performed at high power. However, the manuscript contains no experiments that are directly relevant to this contention. Is it possible to substantiate this claim with experiments. Otherwise, it is a somewhat inflammatory statement to make without support.

Reviewer #2:

Remarks to the Author:

The authors made a commendable effort to address many of the reviewers' concerns, but unfortunately not all of them. Most importantly, my main criticism of the correction method proposed by Pati et al. was not satisfactorily addressed or resolved by the authors. On the contrary, new experimental results presented by the authors in the rebuttal (but not in the manuscript) clearly confirm my concerns. The authors propose to further correct the radiometric transfer efficiency values ϵ , which are previously corrected only for differences in quantum yields, direct excitation, etc., by applying:

$$\frac{1}{E} = \frac{1}{\epsilon} - \zeta$$

where the correction factor zeta was previously introduced (with slightly different notation) by Camley et al. The authors calculate zeta according to a 9-state photophysical model depicted in Fig. 4A of the manuscript, which has also been presented previously by Camley et al.^{*} My criticism is that the model neglects relevant processes like acceptor direct excitation, SSA, STA and RISC. Taking such processes into account leads in general to an inter-dye distance dependence of zeta. However, if zeta itself depends on the distance R , then the correction method proposed here becomes rather useless, since R is usually the unknown parameter for which FRET measurements are performed in the first place.

I suggested that the authors test experimentally whether or not zeta is distance dependent. RESPONSE 12 and Fig. R4 of the rebuttal show the results of such measurements on the Cy3-Cy5 dye pair. Indeed, the data show that zeta is strongly distance dependent. To see this, note that according to the equation above, we have:

$$\frac{1}{\epsilon_{\text{highpower}}} - \frac{1}{\epsilon_{\text{lowpower}}} = \zeta_{\text{highpower}} - \zeta_{\text{lowpower}} \approx \zeta_{\text{highpower}}$$

(Consult Fig. 4f for seeing that $\zeta_{\text{lowpower}} \approx 0$ at 0.04 kW/cm².)

The results show that zeta varies by more than a factor of ten with inter-dye distance. This clearly contradicts the 9-state model of Fig.4a used by the authors, according to which zeta should be independent of R .

The authors have chosen not to discuss the problem of R-dependence of zeta in either the manuscript or the rebuttal, which I find problematic.[†] Potential readers of this paper who want to adopt the method for their own research may be impressed that the correction seems to work well, but may not realize that it is only demonstrated for a specific inter-dye distance. I need to reiterate here that the k_{TR} and k_{ISC} parameters were "optimized" to match the measured and simulated efficiencies[‡]. In this light, it would

^{*} In the revised manuscript, this is still not clearly stated. Instead the authors write: "While excited state contributions to FRET efficiency were initially hypothesized by Camley et al²³, the ζ -correction parameter presented here represents a complete analytical expression of the term given the 9-state photophysical framework". Camley et al. did not just "initially hypothesize", they fully treated the problem, with mathematical rigor and elegance. They introduced $\Lambda = 1 - \zeta$ and discussed with erudition its properties and limitations. Even the possible failure of the theory due to unaccounted for annihilation processes is briefly discussed. As stated in their work, they did not print the analytical expressions for Λ because they are "unwieldy", and can for practical use easily be obtained from any modern computer algebra software.

[†] The authors added to the revised SI some calculations regarding SSA, STA, and RISC. However, they only studied the power dependence, not the distance dependence of zeta.

[‡] The manuscript states that the values for the simulations were "derived from literature". Only in the footnotes of Table 1 in the SI does the more careful reader learn that for Cy3 and Cy5 " k_{TR} values are based on simulations that showed a close matching between simulated FRET efficiency and experimental FRET efficiency values" and " k_{ISC}

seem that other values for k_{TR} and k_{ISC} would be needed for other inter-dye distances. However, there is, of course, no justification for doing so in the context of the 9-state model used in the paper.

Furthermore, the authors state in the rebuttal that they appreciate the need to discuss annihilation processes. However, in the revised manuscript, the corresponding discussion is quite limited and attempts to avoid the relevance for the topic. The authors write: "Although it has been argued that excited state processes negate the need for illumination intensity-related corrections for a specific FRET pair²⁴, this is not the case for Cy3-Cy5 and LD555-LD655 FRET pairs (**Supplementary Fig. 13c**)." This phrasing may be misunderstood by many readers to mean that these "excited state processes" are irrelevant for these dye pairs. The opposite is true. The long lifetime of the T_1 state is the reason why the correction is needed, in this respect, I agree with the authors, but for the same reason STA and RISC cannot be ignored for the correction. As I pointed out in my review, energy transfer to Cy5 while it is in the triplet is certainly expected. Regrettably, the authors chose not to discuss and cite Huang et al. *J. Phys. Chem. A* 2006, 110, 1, 45–50, where this is clearly shown.

Finally, I would like to note that I could not find any calculations of zeta involving direct excitation of the acceptor in the revised manuscript, although such were announced in RESPONSE 15 of the rebuttal. I performed calculations of zeta as a function of inter-dye distance using the Cy3-Cy5 parameters from SI Table 1 for different excitation rates, and found, as expected, that zeta depends strongly on R for values above R_0 , even in the absence of STA and RISC, only due to 5% direct excitation:

In RESPONSE 15 is written: "We note that this correction does not affect the main message of our manuscript."

Based on these important concerns, I cannot support publication of the manuscript.

(intersystem crossing rate constant) values were optimized through simulations." Confronted with this, the authors write in RESPONSE 13 "Our triplet state lifetime estimates were guided by FCS data". This aspect remains unclear.

They further write in RESPONSE 13: "As we show in Fig. 5e, revised manuscript the k_{ISC} value that we arrived at using this approach recapitulated the experimental data for different FRET efficiency values." However, it seems to me that this does not prove anything, because the data in Fig. 5e are simulated for the LD555-LD655 FRET pair at 0.14 kW/cm² excitation intensity. However, according to Fig. 4c, this corresponds to an excitation rate in the range of 10⁵ to 10⁶ per second, where zeta is essentially zero for this dye pair (see Fig. 4f) and no correction is needed. Therefore, the quality of the zeta correction cannot be judged from Fig. 5e.

Reviewer #3:
None

Author Rebuttal, first revision:

Point-by-Point Responses to the Reviewers [NMETH-A52148A]**Reviewer #1**

The manuscript by Pati et al. is a carefully conducted examination the power dependence of FRET efficiency in single molecule experiments where high laser power becomes necessary to obtain increasing time resolution. The work clearly establishes the importance of accounting for triplet state occupancy of the fluorophores to obtain accurate measurements of absolute FRET efficiency. The authors show that this is particularly true of the acceptor triplet state, which can lead to deviations in FRET efficiency at high power. This is of great significance to those in single molecule fluorescence and the growing field of dynamic structural biology, which require high power to obtain accurate measurements of intramolecular distances using FRET. That such a necessary correction has gone unnoticed for so long is surprising, but Pati et al. have clarified the importance of this phenomena and describe methods for mitigation. As such, the work is a great fit for Nature Methods. The manuscript is exceptionally well written and clear. The data are of high quality and the analysis is very rigorous with appropriate use of statistics throughout. The conclusions are well supported by the data. The study is well designed to illustrate the problem and provide mitigation strategies. The authors show the effect of different solution triplet state quenchers (TSQs), which all fall short of complete correction at the highest illumination power levels. The work highlights the importance of intramolecular triplet state quenchers as the only way to avoid this issue. However, the authors also provide a correction factor, which can recover the true FRET efficiency even when triplet state occupancy can't be prevented. The manuscript is suitable for publication as is.

I would suggest that it could be improved by adding a more standard TSQ cocktail in Figure 6, which is a highlight of the paper. This is a very well-designed experiment that should become a standard calibration. However, the use of MV/AA as the TSQ is not optimal and not widely used in single molecule imaging. MV/AA also performs poorly in the author's own tests of TSQs (Fig S4 & S5). I get that the authors want maximum impact from their intramolecular TSQs but a more fair comparison would be to the COT/NBA/Trolox mix, which is much more effective at suppressing triplet state and preserving FRET. I would be curious to see how this mix performed in this experiment as these are the more standard TSQs in smFRET.

The manuscript also makes the contention that the power dependent effects are likely most relevant for confocal single molecule experiments, which are always performed at high power. However, the manuscript contains no experiments that are directly relevant to this contention. Is it possible to substantiate this claim with experiments. Otherwise, it is a somewhat inflammatory statement to make without support.

RESPONSE 1: The request for the experiment comparing to standard buffer conditions was addressed in our first revised manuscript (Fig. 5c). The comment related to the confocal measurements was also addressed in our first revised manuscript (**Supplementary Fig. 3**).

Reviewer #2

The authors made a commendable effort to address many of the reviewers' concerns, but unfortunately not all of them. Most importantly, my main criticism of the correction method proposed by Pati et al. was not satisfactorily addressed or resolved by the authors. On the contrary, new experimental results presented by the authors in the rebuttal (but not in the manuscript) clearly confirm my concerns. The authors propose to further correct the ratiometric transfer efficiency values ϵ , which are previously corrected only for differences in quantum yields, direct excitation, etc., by applying:

$$1/E = 1/\epsilon - \zeta$$

where the correction factor zeta was previously introduced (with slightly different notation) by Camley et al. The authors calculate zeta according to a 9-state photophysical model depicted in Fig. 4A of the manuscript, which has also been presented previously by Camley et al.* My criticism is that the model neglects relevant processes like acceptor direct excitation, SSA, STA and RISC. Taking such processes into account leads in general to an inter-dye distance dependence of zeta. However, if zeta itself depends on the distance R, then the correction method proposed here becomes rather useless, since R is usually the unknown parameter for which FRET measurements are performed in the first place.

I suggested that the authors test experimentally whether or not zeta is distance dependent. RESPONSE 12 and Fig. R4 of the rebuttal show the results of such measurements on the Cy3-Cy5 dye pair. Indeed, the data show that zeta is strongly distance dependent. To see this, note that according to the equation above, we have:

$$1/E_{\text{highpower}} - 1/E_{\text{lowpower}} = \zeta_{\text{highpower}} - \zeta_{\text{lowpower}} \approx \zeta_{\text{highpower}}$$

(Consult Fig. 4f for seeing that $\zeta_{\text{lowpower}} \approx 0$ at 0.04 kW/cm².)

The results show that zeta varies by more than a factor of ten with inter-dye distance. This clearly contradicts the 9-state model of Fig. 4a used by the authors, according to which zeta should be independent of R.

The authors have chosen not to discuss the problem of R-dependence of zeta in either the manuscript or the rebuttal, which I find problematic.† Potential readers of this paper who want to adopt the method for their own research may be impressed that the correction seems to work well, but may not realize that it is only demonstrated for a specific inter-dye distance.

[*In the revised manuscript, this is still not clearly stated. Instead the authors write: “While excited state contributions to FRET efficiency were initially hypothesized by Camley et al²³, the ζ -correction parameter presented here represents a complete analytical expression of the term given the 9-state photophysical framework”. Camley et al. did not just “initially hypothesize”, they fully treated the problem, with mathematical rigor and elegance. They introduced $\Lambda = 1 - \zeta$ and discussed with erudition its properties and limitations. Even the possible failure of the theory due to unaccounted for annihilation processes is briefly discussed. As stated in their work, they did not print the analytical expressions for Λ because they are “unwieldy”, and can for practical use easily be obtained from any modern computer algebra software.]

[†The authors added to the revised SI some calculations regarding SSA, STA, and RISC. However, they only studied the power dependence, not the distance dependence of zeta.]

RESPONSE 1: Our manuscript experimentally demonstrates for the first time that long-lived triplet states are the principal reason behind changes in FRET efficiency that arise at elevated illumination intensity. In cases where triplet states persist despite efforts to eliminate them, or when highly elevated illumination intensities are required, we recommend the implementation of ζ -corrections to recover more accurate and precise FRET efficiency values. Referee 2, however, rightly points out that distance is an important consideration when seeking to achieve “true” FRET efficiencies fully corrected for all experimental and photophysical idiosyncrasies. Camley et al and Nettels et al. were the first to put forward and test the idea that FRET values should be impacted due to the accumulation of excited states species for both donor and acceptor fluorophores at elevated illumination intensities. Camley et al. were first to put forward the idea strictly on theoretical grounds; Nettels et al. were the first to experimentally examine this theory, and they showed that FRET efficiency values don’t in fact change at elevated intensities for the specific dye pair they examined due to canceling effects of other excited state species (eg. SSA, STA and RISC). Consequently, the field has largely overlooked this important topic. To address the Referee concern that we have not sufficiently acknowledged this point, we have further revised our manuscript to more explicitly clarify the undertakings, findings and contributions of these bodies of work.

We agree with the position of Nettels et al. and Referee 2 that precise corrections for illumination intensity-dependent deviations in FRET efficiency require foreknowledge of the distance between donor and acceptor dyes, which cannot be known in advance. In cases where strong SSA, STA and RISC process are present, as in the case for the Alexa 488/Alexa 594 pair, one must also know the rates of these processes and their distance dependence. We understand that consideration of direct excitation, SSA, STA and RISC processes must be given 1] when employing very high illumination intensity regimes (ca. the upper end of intensities employed in confocal imaging studies); 2] in situations where triplet states are relatively long and 3] for the Alexa 488/Alexa 594 pair, where spectral overlaps are present that render these processes relatively efficient. Explained further in Points 1] and 2] below.

Point 1] The “simplified” ζ -correction factor described in our manuscript, which can be implemented by experimentalists to correct for triplet state occupancies manifesting at elevated illumination intensities, is indeed independent of inter-dye distances. This independence, a consequence of the simplified photophysical framework from which it was derived, is a distinguishing advantage that offers an approach for recovering FRET efficiency values more closely approximating “true” FRET efficiencies when triplet states are ineffectively quenched (see **Supplementary Fig. 14, revised manuscript**). We realize that the “simplified” ζ -correction is only an approximation.

Point 2] To address this point directly, we have clarified in our revised manuscript that the “simplified” ζ -correction strategy provides a first approximation to the corrections needed to account for triplet state impacts on FRET efficiency. We therefore clarify in our revised manuscript that FRET efficiency values derived from simplified ζ -corrections can be used to define distance from the experimental data. Hence, the data output from “simplified” ζ -correction procedures can be input into a second-round correction procedure, where the distance information derived from the first round of corrections can input into the theoretical model to yield an “iterative” ζ -correction that even more closely approximate “true” FRET efficiency. Although we extensively studied the effects of SSA, STA and RISC across illumination intensities in our revised Supplemental Information

file, we explain in RESPONSE 3 below why we have not included SSA, STA and RISC in photophysical model used to derive the “simplified” ζ -correction factor.

As shown in **Figure R1**, both the simplified and iterative ζ -correction procedures proposed robustly correct for triplet state impacts on FRET efficiency across the full range of FRET-relevant distances. The significance of these

corrections are most profound for the Cy3/Cy5 pair (**Figure R1b**), where both donor and acceptor fluorophore exhibit relatively long (ca. 30 ns) triplet states.

Figure R1c corroborates our finding that ζ -correction procedures are generally not required for the LD555/LD655 FRET pair, where both donor and acceptor fluorophore possess inherently short triplet state lifetimes (ca. <1 μ s). Please note that the y-axis values for the LD555-LD655 pair are 60-fold smaller than for the Cy3/Cy5 pair.

In our revised manuscript, we have now clarified that experimentalists should quench triplet states to the best of their ability using judiciously selected strategies of their choice and verify that triplet states are remain an issue for their measurements by examining the illumination-intensity dependence of donor and acceptor brightness before employing procedures ζ -correction. We have also clarified that the simplified ζ -correction procedure's utility is based on its distance independence and that ζ -corrections should be implemented with consideration that its precision diminishes when inter-dye distances greater than the R_0 value of the FRET pair are examined. The reason for this is because at relatively long distances the magnitude of the acceptor fluorescence intensity due to FRET diminishes so as to become comparable to that arising from direct excitation by the donor laser.

I need to reiterate here that the k_{TR} and k_{ISC} parameters were "optimized" to match the measured and simulated efficiencies[†]. In this light, it would seem that other values for k_{TR} and k_{ISC} would be needed for other inter-dye distances. However, there is, of course, no justification for doing so in the context of the 9-state model used in the paper.

[[†]The manuscript states that the values for the simulations were "derived from literature". Only in the footnotes of Table 1 in the SI does the more careful reader learn that for Cy3 and Cy5 " k_{TR} values are based on simulations that showed a close matching between simulated FRET efficiency and experimental FRET efficiency values" and " k_{ISC} (intersystem crossing rate constant) values were optimized through simulations." Confronted with this, the authors write in RESPONSE 13 "Our triplet state lifetime estimates were guided by FCS data". This aspect remains unclear. They further write in RESPONSE 13: "As we show in Fig. 5e, revised manuscript the k_{ISC} value that we arrived at using this approach recapitulated the experimental data for different FRET efficiency values." However, it seems to me that this does not prove anything, because the data in Fig. 5e are simulated for the LD555-LD655 FRET pair at 0.14 kW/cm² excitation intensity. However, according to Fig. 4c, this corresponds to an excitation rate in the range of 10⁵ to 10⁶ per second, where zeta is essentially zero for this dye pair (see Fig. 4f) and no correction is needed. Therefore, the quality of the zeta correction cannot be judged from Fig. 5e.

RESPONSE 2: To address this Referee concern, we have carefully revised our text to make it more explicitly clear that literature values for k_{TR} and k_{ISC} were used as initial boundary conditions for our simulations, where we established k_{TR} and k_{ISC} parameters that enabled us to more closely recapitulate experimental FRET efficiency values across different distances (**Figure 5d,e**). As the literature values for k_{TR} and k_{ISC} were obtained from distinct reaction conditions and solvent, slightly different values are to be anticipated. Throughout the manuscript and supplementary documents, we have clarified that k_{TR} and k_{ISC} parameters were optimized by simulations using literature values as initial boundary conditions as well as estimations of k_{TR} in our aqueous solutions obtained from FCS data presented in our work.

As described above, the quality of the zeta correction is now apparent in **Figure R1 (Supplementary Fig. 14)**. We have now also provided two additional supplementary figures correcting Cy3/Cy5 FRET pairs data using zeta correction protocol and comparing them with the standard methods (**Supplementary Figs. 15,16**).

Furthermore, the authors state in the rebuttal that they appreciate the need to discuss annihilation processes. However, in the revised manuscript, the corresponding discussion is quite limited and attempts to avoid the relevance for the topic. The authors write: "Although it has been argued that excited state processes negate the need for illumination intensity-related corrections for a specific FRET pair²⁴, this is not the case for Cy3-Cy5 and LD555-LD655 FRET pairs (Supplementary Fig. 13c)." This phrasing may be misunderstood by many readers to mean that these "excited state processes" are irrelevant for these dye pairs. The opposite is true. The long lifetime of the T₁ state is the reason why the correction is needed, in this respect, I agree with the authors, but for the same reason STA and RISC cannot be ignored for the correction. As I pointed out in my review, energy transfer to Cy5 while it is in the triplet is certainly expected. Regrettably, the authors chose not to discuss and cite Huang et al. *J. Phys. Chem. A* 2006, 110, 1, 45–50, where this is clearly shown.

RESPONSE 3: We have removed the concerned words "excited state processes" from our revised manuscript to avoid unnecessary confusion.

Nettels et al. report that "Transient absorption spectra of the T₁ state have been measured with flash photolysis and radiolysis experiments for several organic dyes, including rhodamines.^{28–32} In these cases, the absorption spectra of the T₁ state are blue-shifted with respect to S₁ absorption, and the T₁ absorption cross-section is of similar magnitude, often even greater, than the S₁ absorption. The resulting larger overlap of the donor S₁ emission and the acceptor T₁ absorption suggests that the Forster radius for the S₁T₁-S₀T_n energy transfer tends to be larger than the one of the S₁S₀-S₀S₁ transfer." We believe that Referee 2 understands that the same to be true for cyanine-class dyes and therefore finds it 'regrettable' that we did not cite Huang et al., *J. Phys. Chem. A* 2006, as they report that Cy5 has a triplet absorption band at both ~625 nm (blue-shifted with respect to its S₀-S₁ absorbance max of 650 nm) and ~690 nm (red-shifted with respect to its S₀-S₁ absorbance max of 650 nm). We are aware of this publication, and its potential relevance to the STA/RISC pathways in our photophysical model, however, our team has published (Zheng et al., *Chem. Sci.*, 2017; Zheng et al., *J. Phys. Chem. Lett.*, 2012) that only the ~700 nm feature is the experimentally relevant triplet state. Huang et al. only see the 625 nm feature with the addition of ethyl iodide, which induces heavy-atom effects that increase triplet state occupation by increasing k_{isc} . Huang et al. also did not show that either state's lifetime was sensitive to molecular oxygen, which would have identified its triplet character. Since the 625 nm feature only appears with additives that dramatically alter k_{isc} , it should only be considered in systems with these heavy-atom inducers (iodide, bromide, 2nd/3rd row transition metals). As these additives, to our knowledge, are not used for smFRET investigations, we omit consideration of STA in our model. Consistent with the 625 nm triplet assignment of Huang et al. being irrelevant for our studies, we do not observe evidence of STA and RISC processes for cyanine-class fluorophores in typical TIRF power regimes, as judged by both the strong power dependence of FRET efficiencies and our capacity to correct for triplet state impacts using ζ -corrections that do not include SSA, STA or RISC considerations. For the LD555/LD655 FRET pair this conclusion extends into the confocal power regimes as well. This conclusion is supported by our simulations (Fig. 4), where we show that the prerequisite states for such processes are poorly occupied for both Cy3/Cy5 and LD555/LD655 FRET pairs in typical TIRF illumination regimes. These and related simulations that we present show, however, that confocal microscopy illumination intensities begin to sufficiently populate Cy3/Cy5 FRET pair excited states for STA/RISC processes to occur. Each consideration leads us to conclude that 1] triplet states indeed complicate smFRET measurements and reduce the experimental "accuracy and precision"; 2] robust triplet state quenching (such as intra-molecular triplet state quenching present in the LD555/LD655 pair) greatly reduces issues associated with excited state entanglements. When triplet states are inefficiently mitigated, we show that implementation of the distance-independent ζ -correction parameter into FRET calculations provides a robust approach to recovering FRET efficiency values that more closely approximate "true" FRET efficiencies. As discussed above, we realize that this approach is only approximate but that it does provide the means to make distance-independent corrections.

Finally, I would like to note that I could not find any calculations of zeta involving direct excitation of the acceptor in the revised manuscript, although such were announced in RESPONSE 15 of the rebuttal. I performed calculations of zeta as a function of inter-dye distance using the Cy3-Cy5 parameters from SI Table 1 for different excitation rates, and found, as expected, that zeta depends strongly on R for values above R_0 , even in the absence of STA and RISC, only due to 5% direct excitation:

In RESPONSE 15 is written: “We note that this correction does not affect the main message of our manuscript.” Based on these important concerns, I cannot support publication of the manuscript.

RESPONSE 4: As stated above in RESPONSE 1 above, we have now explicitly considered the direct excitation issue and included an iterative ζ -correction approach in our revised manuscript (Figure R1, Supplementary Figs. 14 - 16).

Decision Letter, second revision:

Dear Scott,

Thank you for submitting your revised manuscript "Recovering true FRET efficiencies from smFRET investigations requires triplet state mitigation" (NMETH-A52148B). It has now been seen by the original referees and their comments are below. The reviewers find that the paper has improved in revision, and therefore we'll be happy in principle to publish it in Nature Methods, pending minor revisions to satisfy the referees' final requests and to comply with our editorial and formatting guidelines.

Regarding the remaining referee comments, we ask that you clearly state that the zeta correction factor is an approximation and mention photophysical nuances that may lead to deviations from your assumptions. You can emphasize that the best strategy involves getting better data from that start using either lower illumination intensities or the self-healing dyes.

You will see that reviewer 2 asks that the reviews be made public. We have this as an option for all authors, and see how the reviews could benefit the community in this case. Still, we leave this entirely at your discretion.

TRANSPARENT PEER REVIEW

Please note: we allow redactions to authors' rebuttal and reviewer comments in the interest of confidentiality. If you are concerned about the release of confidential data, please let us know specifically what information you would like to have removed. Please note that we cannot incorporate redactions for any other reasons. Reviewer names will be published in the peer review files if the reviewer signed the comments to authors, or if reviewers explicitly agree to release their name. For more information, please refer to our FAQ page.

ORCID

Sincerely,
Rita

Rita Strack, Ph.D.
Senior Editor
Nature Methods

Reviewer #1 (Remarks to the Author):

The authors have done a truly commendable job of adding additional references, experimental details, new experiments and new simulations, which have increased the scope of the manuscript. The authors have also clarified, in specific details, their contribution to this problem relative to the works of Camley et al. and Nettels et al. in that the authors are the first to describe FRET pairs that experimentally show zeta dependence and implement practical approaches (experimental and mathematical) to solve the problem. The initial draft stressed the importance of mitigating dark state occupancy through dye selection to obtain accurate FRET efficiency, which I felt was underappreciated in the field and worthy of publication for this reason alone. They also provided a simplified correction that significantly improved FRET accuracy but, as stressed by Reviewer 2, may have fallen short of "true" FRET efficiency. The revised manuscript now provides additional experimental details and an expanded model of the underlying photophysical transitions responsible for the magnitude of this phenomenon involving both new simulations and new experiments. Additionally, the authors have expanded the development of a practical strategy for mathematically correcting for dark state occupancy to obtain "true" FRET efficiency, which may be more cumbersome than experimentally solving the problem if possible. Nonetheless, the authors have now explicitly demonstrated the feasibility of applying their iterative methods for determining the relevant zeta value for systems with unknown inter-dye distance. They also show that the original simplified correction performed reasonably well such that the additional work resulted in minimal further improvement. Depending on the nature of the FRET experiment, this additional level of performance may be useful but did not substantially change the conclusions of the previous draft. In summary, the current manuscript draws focus to a previously known phenomena by experimentally demonstrating the magnitude of the problem for the most commonly used dyes in single molecule FRET. They demonstrated how the problem could be solved through the use of self-healing dyes and now provide a workable solution for improving FRET accuracy when experimental limitations cannot be overcome. While there may still be disagreements about the sufficiency of theory underlying these phenomena, the practical application shows that the two strategies (experimental and mathematical) do work in experiments. For this reason, I feel that this expanded manuscript should be published.

Reviewer #2 (Remarks to the Author):

See attachment.

Pati et al. have made substantial changes to the manuscript as a result of my strong criticism. They now clearly acknowledge that the 9-state photophysical model and the $\Lambda=1$ -zeta parameter they use were introduced earlier by Camely et al. They also acknowledge that the model is, in their own words, "naive" with respect to a possible inter-dye distance dependence of zeta. Such a dependence is expected due to processes (not included in the model) like direct acceptor excitation, SSA, STA and RISC as previously demonstrated and discussed theoretically and experimentally for a pair of rhodamine dyes by Nettels et al. The authors of the manuscript now admit that direct acceptor excitation makes zeta distance dependent, but they are still convinced that STA and RISC are negligible, not in general, but in the case of the Cy3/ Cy5 dye pair. According to their interpretation of self-conducted transient absorption measurements, presented in Zheng et al. (Chem. Sci., 2017) and Zheng et al. (J. Phys. Chem. Let., 2012), does the triplet absorption of Cy5 not overlap with the singlet emission of Cy3. This is in contrast to earlier results by Huang et al. (J. Phys. Chem. A 2006), which suggested a rather strong overlap. Zheng et al. and Huang et al. used different strategies to populate the triplet state. I find it difficult to interpret these apparently contradictory results.

In any case, if it is true that the STA and RISC pathways are not relevant for cyanine-class fluorophores, as the authors claim in their newest rebuttal, then I would have expected them to come up with some other explanation for the measured distance dependence of zeta that the authors provided in their first rebuttal after I had asked for such measurements:

FIG. 1 (taken from first rebuttal)

The bars in these plots (b and c) essentially represent measured values of zeta as I pointed out in my response to the first rebuttal. The observed distance dependence is clearly not explained by including the effect of direct acceptor excitation alone. (See the above-mentioned calculation in my last response, reproduced in Fig. 3 below.) Instead of providing an explanation for the cause of the distance dependence, the authors now admit, that the naïve zeta correction is only an approximation. However, by introducing the new Supplementary Fig. 14 (Figure R1 in the second rebuttal), they now claim that this approximation does still serve well to improve the accuracy of transfer efficiency determination. For later comparison, I reproduce panel (b) of this figure here:

FIG.2 (Deviations of simulated data from E_{true} , taken from Supplementary Fig. 14)

According to the figure caption in the SI, these results showing the deviation $|E_{true} - E_{app}|$ are based on synthetic data, simulated for the same Cy3/Cy5 labeled constructs shown in Fig1a above. The naïve 9-state model is used with an excitation rate of $10^6 s^{-1}$, including direct excitation as the only source of distance dependence of zeta. The small deviations obtained after naïve zeta correction (green bars) seem to support the proposed method; all deviations are below 0.05. However, this does not prove much, as can be seen from my earlier calculations of the distance dependence of zeta:

FIG. 3 (taken from my response to the first rebuttal)

Since zeta is essentially constant for the first five distances probed (see black points, $R_0=5.69$ nm) assuming $k_{ex} = 10^6 s^{-1}$, a good approximation is, of course, expected from the distance-naïve zeta correction. Only for the sixth point with the largest dye spacing (8 nm), zeta is decreased. Interestingly, this is exactly the case where the naïve zeta correction does not improve the accuracy, but it degrades it. (See at 8 nm in Fig. 2.)

Much more conclusive is when the same analysis as in Fig. 2 is applied not to simulations but to the real FRET measurements shown in FIG. 1c above. The result is quite different:

FIG. 4 (Deviations of experimental data from E_{true} , back-calculated from the data in Fig 1.c)

Here I used the dye spacings reported in FIG. 2 to calculate the "true" transfer efficiencies using $R_0=5.69$ nm (see SI). Since the original data were not available, I back-calculated the measured values from $E_{app} = 1 / (\frac{1}{E_{true}} + \zeta)$ using the values for zeta given in FIG.1c.

Note that the resulting experimental deviations (orange bars) are much smaller than assumed for the simulation above (in Fig. 2) and do not decrease monotonically with dye spacing, as it would be expected for constant zeta. This means that the simulations presented above are quite far from reality. Next, I applied the naive zeta correction (green bars) to the experimental data exactly as it was done for the simulated data by using the zeta value obtained for the construct with the fourth-longest interdye distance (5.98 nm).

Given the discrepancy between the measured (Fig.1) and simulated (Fig.3) dye spacing dependencies of zeta, it is not surprising, that applying the naive zeta correction method proposed by Pati et al. to leads to large deviations from the "true" values, even larger than for the correction without zeta. The deterioration is drastic. (Note that for comparison with FIG. 2, absolute values are shown, but the green deviations are in all cases negative (or zero). This leads for the shortest dye spacing to an unphysical transfer efficiency value of 1.18 when the naive zeta correction is applied). This clearly shows on the one hand that optimizing kisc and kT of the 9-state model for one construct and using the resulting zeta value for all constructs does not lead to the goal of improving the accuracy of transfer efficiency determination when applied to the experimental data measured on these Cy3/Cy5 constructs. On the other hand, it is clear that the simulated data shown in Figure 2 are unrealistic, because they are built on an incomplete model. This result calls into question not only the validity of the proposed correction method, but potentially all other quantitative results presented in the paper that are based on this model.

In summary, application of the proposed zeta correction to the authors' self-measured data of Fig.1 (which, unfortunately, although highly relevant, never made it into the manuscript) reveals the inappropriateness of the method.

Certainly, a more complete understanding of the photophysics of the Cy3/Cy5 pair would be required for a full quantitative understanding of the experimental data of Fig 1. Until this is achieved, I do not see a simple method for correcting FRET data measured with this dye pair in the high excitation regime. I certainly agree with the authors that the use of effective triplet state quenching techniques is currently the best way to go.

In conclusion, I still do not support publication of the manuscript, because potential thoughtless use of the here presented distance-naïve correction method based on the 9-state model without careful consideration of the pitfalls by the user would not serve the FRET community well. If, despite of my well-

founded criticism, the editor comes to a different conclusion, I assume, in the interest of transparency, that the entire review process will be published along with the manuscript.

Author Rebuttal, second revision:

Point-by-Point Responses to the Reviewers [NMETH-A52148B]**Reviewer #1**

The authors have done a truly commendable job of adding additional references, experimental details, new experiments and new simulations, which have increased the scope of the manuscript. The authors have also clarified, in specific details, their contribution to this problem relative to the works of Camley et al. and Nettels et al in that the authors are the first to describe FRET pairs that experimentally show zeta dependence and implement practical approaches (experimental and mathematical) to solve the problem. The initial draft stressed the importance of mitigating dark state occupancy through dye selection to obtain accurate FRET efficiency, which I felt was underappreciated in the field and worthy of publication for this reason alone. They also provided a simplified correction that significantly improved FRET efficiency but, as stressed by Reviewer 2, may have fallen short of "true" FRET efficiency. The revised manuscript now provides additional experimental details and an expanded model of the underlying photophysical transitions responsible for the magnitude of this phenomenon involving both new simulations and new experiments. Additionally, the authors have expanded the development of a practical strategy for mathematically correcting for dark state occupancy to obtain "true" FRET efficiency, which may be more cumbersome than experimentally solving the problem if possible. Nonetheless, the authors have now explicitly demonstrated the feasibility of applying their iterative methods for determining the relevant zeta value for systems with unknown interdye distance. They also show that the original simplified correction performed reasonably well such that the additional work resulted in minimal further improvement. Depending on the nature of the FRET experiment, this additional level of performance may be useful but did not substantially change the conclusions of the previous draft. In summary, the current manuscript draws focus to a previously known phenomena by experimentally demonstrating the magnitude of the problem for the most commonly used dyes in single molecule FRET. They demonstrated how the problem could be solved through the use of self-healing dyes and now provide a workable solution for improving FRET accuracy when experimental limitations cannot be overcome. While there may still be disagreements about the sufficiency of theory underlying these phenomena, the practical application shows that the two strategies (experimental and mathematical) do work in experiments. For this reason, I feel that this expanded manuscript should be published.

RESPONSE 1: We thank the Referee for their positive feedback and acceptance of the manuscript.

Reviewer #2

Pati et al. have made substantial changes to the manuscript as a result of my strong criticism. They now clearly acknowledge that the 9-state photophysical model and the $\Lambda=1-\zeta$ parameter they use were introduced earlier by Camely et al. They also acknowledge that the model is, in their own words, "naive" with respect to a possible inter-dye distance dependence of zeta. Such a dependence is expected due to processes (not included in the model) like direct acceptor excitation, SSA, STA and RISC as previously demonstrated and discussed theoretically and experimentally for a pair of rhodamine dyes by Nettels et al. The authors of the manuscript now admit that direct acceptor excitation makes zeta distance dependent, but they are still convinced that STA and RISC are negligible, not in general, but in the case of the Cy3/ Cy5 dye pair. According to their interpretation of self-conducted transient absorption measurements, presented in Zheng et al. (Chem. Sci., 2017) and Zheng et al. (J. Phys. Chem. Let., 2012), does the triplet absorption of Cy5 not overlap with the singlet emission of Cy3. This is in contrast to earlier results by Huang et al. (J. Phys. Chem. A 2006), which suggested a rather strong overlap. Zheng et al. and Huang et al. used different strategies to populate the triplet state. I find it difficult to interpret these apparently contradictory results.

In any case, if it is true that the STA and RISC pathways are not relevant for cyanine-class fluorophores, as the authors claim in their newest rebuttal, then I would have expected them to come up with some other explanation for the measured distance dependence of zeta that the authors provided in their first rebuttal after I had asked for such measurements:

FIG. 1 (taken from first rebuttal)

The bars in these plots (b and c) essentially represent measured values of zeta as I pointed out in my response to the first rebuttal. The observed distance dependence is clearly not explained by including the effect of direct acceptor excitation alone. (See the above-mentioned calculation in my last response, reproduced in Fig. 3 below.) Instead of providing an explanation for the cause of the distance dependence, the authors now admit, that the naïve zeta correction is only an approximation. However, by introducing the new Supplementary Fig. 14 (Figure R1 in the second rebuttal), they now claim that this approximation does still serve well to improve the accuracy of transfer efficiency determination. For later comparison, I reproduce panel (b) of this figure here:

FIG.2 (Deviations of simulated data from E_{true} , taken from Supplementary Fig. 14)

According to the figure caption in the SI, these results showing the deviation $|E_{true} - E_{app}|$ are based on synthetic data, simulated for the same Cy3/Cy5 labeled constructs shown in Fig1a above. The naïve 9- state model is used with an excitation rate of $10^6 s^{-1}$, including direct excitation as the only source of distance dependence of zeta. The small deviations obtained after naïve zeta correction (green bars) seem to support the proposed method; all deviations are below 0.05. However, this does not prove much, as can be seen from my earlier calculations of the distance dependence of zeta:

FIG. 3 (taken from my response to the first rebuttal)

Since zeta is essentially constant for the first five distances probed (see black points, $R_0=5.69$ nm) assuming $k_{ex} = 10^6 s^{-1}$, a good approximation is, of course, expected from the distance-naïve zeta correction. Only for the sixth point with the largest dye spacing (8 nm), zeta is decreased. Interestingly, this is exactly the case where the naïve zeta correction does not improve the accuracy, but it degrades it. (See at 8 nm in Fig. 2.)

Much more conclusive is when the same analysis as in Fig. 2 is applied not to simulations but to the real FRET measurements shown in FIG. 1c above. The result is quite different FIG. 4.

(Deviations of experimental data from E_{true} , back-calculated from the data in Fig 1.c):

Here I used the dye spacings reported in FIG. 2 to calculate the "true" transfer efficiencies using $R_0=5.69$ nm (see SI). Since the original data were not available, I back-calculated the measured values from

$$E_{app} = 1 / \left(\frac{1}{E_{true}} + \zeta \right) \text{ using the values for zeta given in FIG.1c.}$$

Note that the resulting experimental deviations (orange bars) are much smaller than assumed for the simulation above (in Fig. 2) and do not decrease monotonically with dye spacing, as it would be expected for constant zeta. This means that the simulations presented above are quite far from reality. Next, I applied the naive zeta correction (green bars) to the experimental data exactly as it was done for the simulated data by using the zeta value obtained for the construct with the fourth-longest interdye distance (5.98 nm).

Given the discrepancy between the measured (Fig.1) and simulated (Fig.3) dye spacing dependencies of zeta, it is not surprising, that applying the naive zeta correction method proposed by Pati et al. to leads to large deviations from the "true" values, even larger than for the correction without zeta. The deterioration is drastic. (Note that for comparison with FIG. 2, absolute values are shown, but the green deviations are in all cases negative (or zero). This leads for the shortest dye spacing to an unphysical transfer efficiency value of 1.18 when the naive zeta correction is applied). This clearly shows on the one hand that optimizing k_{isc} and k_T of the 9-state model for one construct and using the resulting zeta value for all constructs does not lead to the goal of improving the accuracy of transfer efficiency determination when applied to the experimental data measured on these Cy3/Cy5 constructs. On the other hand, it is clear that the simulated data shown in Figure 2 are unrealistic, because they are built on an incomplete model. This result calls into question not only the validity of the proposed correction method, but potentially all other quantitative results presented in the paper that are based on this model.

In summary, application of the proposed zeta correction to the authors' self-measured data of Fig.1 (which, unfortunately, although highly relevant, never made it into the manuscript) reveals the inappropriateness of the method.

Certainly, a more complete understanding of the photophysics of the Cy3/Cy5 pair would be required for a full quantitative understanding of the experimental data of Fig 1. Until this is achieved, I do not see a simple method for correcting FRET data measured with this dye pair in the high excitation regime. I certainly agree with the authors that the use of effective triplet state quenching techniques is currently the best way to go.

In conclusion, I still do not support publication of the manuscript, because potential thoughtless use of the here presented distance-naïve correction method based on the 9-state model without careful consideration of the pitfalls by the user would not serve the FRET community well. If, despite of my well-founded criticism, the editor comes to a different conclusion, I assume, in the interest of transparency, that the entire review process will be published along with the manuscript.

RESPONSE 1: We thank the Referee for their valuable feedback and comments.

We fully appreciate and understand the Referee's concerns. As a result of the valuable input received during the review process, we have decided to restrict significantly all discussions surrounding the use of ζ -correction procedures to avoid any potential misuse by practitioners in the field. As our primary goal is to make the scientific community aware of the contributions of triplet states to FRET efficiency, in our revised manuscript, we only define how we successfully implemented distance-naïve ζ -correction procedures to recover more accurate FRET efficiency values for the specific system examined. We further acknowledge the distance dependency of ζ -corrections and emphasize that users should strive to robustly suppress triplet states as their principal goal rather than trying to correct data that is skewed by triplet state and associated excited-state accumulations. We also emphasize the diminishing performance of ζ -correction procedures at elevated excitation rates and short distances, where singlet excited states contribute more substantially to FRET efficiency. Finally, we explicitly clarify

that inappropriate implementation of ζ -correction procedures can lead to over-corrections when excited-state accumulations do contribute substantially to the signals detect, which lead to FRET efficiency calculations that are further from the truth than if the data were not put through ζ -correction procedures.

Final Decision Letter:

Dear Scott,

I am pleased to inform you that your Article, "Recovering true FRET efficiencies from smFRET investigations requires triplet state mitigation", has now been accepted for publication in Nature Methods. The received and accepted dates will be March 29, 2023 and April 25, 2024. This note is intended to let you know what to expect from us over the next month or so, and to let you know where to address any further questions.

Over the next few weeks, your paper will be copyedited to ensure that it conforms to Nature Methods style. Once your paper is typeset, you will receive an email with a link to choose the appropriate publishing options for your paper and our Author Services team will be in touch regarding any additional information that may be required. It is extremely important that you let us know now whether you will be difficult to contact over the next month. If this is the case, we ask that you send us the contact information (email, phone and fax) of someone who will be able to check the proofs and deal with any last-minute problems.

Please note that *Nature Methods* is a Transformative Journal (TJ). Authors may publish their research with us through the traditional subscription access route or make their paper immediately open access through payment of an article-processing charge (APC). Authors will not be required to make a final decision about access to their article until it has been accepted. Find out more about Transformative Journals

You may wish to make your media relations office aware of your accepted publication, in case they consider it appropriate to organize some internal or external publicity. Once your paper has been scheduled you will receive an email confirming the publication details. This is normally 3-4 working days in advance of publication. If you need additional notice of the date and time of publication,

please let the production team know when you receive the proof of your article to ensure there is sufficient time to coordinate. Further information on our embargo policies can be found here: <https://www.nature.com/authors/policies/embargo.html>

If you are active on Twitter/X, please e-mail me your and your coauthors' handles so that we may tag you when the paper is published.

Best regards,
Rita

Rita Strack, Ph.D.
Senior Editor
Nature Methods